# DIFFUSION MODELS MEET CONTEXTUAL BANDITS

## ABSTRACT

Efficient exploration in contextual bandits is crucial due to their large action space, where uninformed exploration can lead to computational and statistical inefficiencies. However, the rewards of actions are often correlated, which can be leveraged for more efficient exploration. In this work, we use pre-trained diffusion model priors to capture these correlations and develop diffusion Thompson sampling (`dTS`). We establish both theoretical and algorithmic foundations for `dTS`. Specifically, we derive efficient posterior approximations (required by `dTS`) under a diffusion model prior, which are of independent interest beyond bandits and reinforcement learning. We analyze `dTS` in linear instances and provide a Bayes regret bound. Our experiments validate our theory and demonstrate `dTS`'s favorable performance.

## 1 INTRODUCTION

A *contextual bandit* is a popular and practical framework for online learning under uncertainty (Li et al., 2010). In each round, an agent observes a *context*, takes an *action*, and receives a *reward* based on the context and action. The goal is to maximize the expected cumulative reward over $n$ rounds, striking a balance between exploiting actions with high estimated rewards from available data and exploring other actions to improve current estimates. This trade-off is often addressed using either *upper confidence bound (UCB)* (Auer et al., 2002) or *Thompson sampling (TS)* (Scott, 2010).

The action space in contextual bandits is often large, resulting in less-than-optimal performance with standard exploration strategies. Luckily, actions usually exhibit correlations, making efficient exploration possible as one action may inform the agent about other actions. In particular, Thompson sampling offers remarkable flexibility, allowing its integration with informative priors (Hong et al., 2022b) that capture these correlations. Inspired by the achievements of diffusion models (Sohl-Dickstein et al., 2015; Ho et al., 2020), which effectively approximate complex distributions (Dhariwal & Nichol, 2021; Rombach et al., 2022), this work captures action correlations by employing diffusion models as priors in contextual Thompson sampling.

We illustrate the idea using video streaming. The objective is to optimize watch time for a user $j$ by selecting a video $i$ from a catalog of $K$ videos. Users $j$ and videos $i$ are associated with context vectors $x_j$ and unknown video parameters $\theta_i$, respectively. User $j$'s expected watch time for video $i$ is linear as $x_j^\top \theta_i$. Then, a natural strategy is to independently learn video parameters $\theta_i$ using `LinTS` or `LinUCB` (Agrawal & Goyal, 2013a; Abbasi-Yadkori et al., 2011), but this proves statistically inefficient for larger $K$. Fortunately, the reward when recommending a movie can provide informative insights into other movies. To capture this, we leverage offline estimates of video parameters denoted by $\hat{\theta}_i$ and build a diffusion model on them. This diffusion model approximates the video parameter distribution, capturing their dependencies. This model enriches contextual Thompson sampling as a prior, effectively capturing complex video dependencies while ensuring computational efficiency.

We introduce a framework for contextual bandits with diffusion model priors, upon which we develop diffusion Thompson sampling (`dTS`) that is both computationally and statistically efficient. `dTS` requires *fast updates of the posterior* and *fast sampling from the posterior*, both of which are achieved through our novel efficient posterior approximations. These approximations become exact when both the diffusion model and likelihood are linear. We establish a bound on `dTS`'s Bayes regret for this specific case, highlighting the advantages of using diffusion models as priors. Our empirical evaluations validate our theory and demonstrate `dTS`'s strong performance across various settings.

Diffusion models were applied in offline decision-making (Ajay et al., 2022; Janner et al., 2022; Wang et al., 2022), but their use in online learning was only recently explored by Hsieh et al. (2023),

who focused on *multi-armed bandits without theoretical guarantees*. Our work extends Hsieh et al. (2023) in two ways. First, we apply the concept to the broader contextual bandit, which is more practical and realistic. Second, we show that with diffusion models parametrized by linear link functions and linear rewards, we can derive exact closed-form posteriors without approximations. These exact posteriors are valuable as they enable theoretical analysis (unlike Hsieh et al. (2023), who did not provide theoretical guarantees) and motivate efficient approximations for non-linear link functions in contextual bandits, addressing gaps in Hsieh et al. (2023)'s focus on multi-armed bandits.

A key contribution, beyond applying diffusion models in contextual bandits, is the efficient *computation* and *sampling* of the posterior distribution of a $d$-dimensional parameter $\theta \mid H_t$, with $H_t$ representing the data, when using a diffusion model prior on $\theta$. This is relevant not only to bandits and RL but also to a broader range of applications (Chung et al., 2022). Our approximations are motivated with exact closed-form solutions obtained in cases where both the link functions of the diffusion model and the likelihood are linear. These solutions form the basis for our approximations for non-linear link functions, demonstrating both strong empirical performance and computational efficiency. Our approach avoids the computational burden of heavy approximate sampling algorithms required for each latent parameter. For a detailed related work discussion, see Appendix A, where we discuss diffusion models in decision-making, structured bandits, approximate posteriors, etc.

## 2 SETTING

The agent interacts with a *contextual bandit* over $n$ rounds. In round $t \in [n]$, the agent observes a *context* $X_t \in \mathcal{X}$, where $\mathcal{X} \subseteq \mathbb{R}^d$ is a *context space*, it takes an *action* $A_t \in [K]$, and then receives a stochastic reward $Y_t \in \mathbb{R}$ that depends on both the context $X_t$ and the taken action $A_t$. Each action $i \in [K]$ is associated with an *unknown action parameter* $\theta_{*,i} \in \mathbb{R}^d$, so that the reward received in round $t$ is $Y_t \sim P(\cdot \mid X_t; \theta_{*,A_t})$, where $P(\cdot \mid x; \theta_{*,i})$ is the reward distribution of action $i$ in context $x$. Throughout the paper, we assume that the reward distribution is parametrized as a generalized linear model (GLM) (McCullagh & Nelder, 1989). That is, for any $x \in \mathcal{X}$, $P(\cdot \mid x; \theta_{*,i})$ is an exponential-family distribution with mean $g(x^\top \theta_{*,i})$, where $g$ is the mean function. For example, we recover linear bandits when $P(\cdot \mid x; \theta_{*,i}) = \mathcal{N}(\cdot; x^\top \theta_{*,i}, \sigma^2)$ where $\sigma > 0$ is the observation noise. Similarly, we recover logistic bandits (Filippi et al., 2010) if we let $g(u) = (1 + \exp(-u))^{-1}$ and $P(\cdot \mid x; \theta_{*,i}) = \mathrm{Ber}(g(x^\top \theta_{*,i}))$, where $\mathrm{Ber}(p)$ be the Bernoulli distribution with mean $p$.

We consider the *Bayesian* bandit setting (Russo & Van Roy, 2014; Hong et al., 2022b; Neu et al., 2022; Gouverneur et al., 2023), where the action parameters $\theta_{*,i}$ are assumed to be sampled from a *known* prior distribution. We proceed to define this prior distribution using a diffusion model. The correlations between the action parameters $\theta_{*,i}$ are captured through a diffusion model, where they share a set of $L$ consecutive *unknown latent parameters* $\psi_{*,\ell} \in \mathbb{R}^d$ for $\ell \in [L]$. Precisely, the action parameter $\theta_{*,i}$ depends on the $L$-th latent parameter $\psi_{*,L}$ as $\theta_{*,i} \mid \psi_{*,1} \sim \mathcal{N}(f_1(\psi_{*,1}), \Sigma_1)$, where the *link function* $f_1$ and covariance $\Sigma_1$ are *known*. Also, the $\ell - 1$-th latent parameter $\psi_{*,\ell-1}$ depends on the $\ell$-th latent parameter $\psi_{*,\ell}$ as $\psi_{*,\ell-1} \mid \psi_{*,\ell} \sim \mathcal{N}(f_\ell(\psi_{*,\ell}), \Sigma_\ell)$, where $f_\ell$ and $\Sigma_\ell$ are *known*. Finally, the $L$-th latent parameter $\psi_{*,L}$ is sampled as $\psi_{*,L} \sim \mathcal{N}(0, \Sigma_{L+1})$, where $\Sigma_{L+1}$ is *known*. We summarize this model in Eq. (1) and its graph in Fig. 1.

$$
\begin{aligned}
\psi_{*,L} &\sim \mathcal{N}(0, \Sigma_{L+1}), \\
\psi_{*,\ell-1} \mid \psi_{*,\ell} &\sim \mathcal{N}(f_\ell(\psi_{*,\ell}), \Sigma_\ell), \quad \forall \ell \in [L]/\{1\}, \\
\theta_{*,i} \mid \psi_{*,1} &\sim \mathcal{N}(f_1(\psi_{*,1}), \Sigma_1), \quad \forall i \in [K], \\
Y_t \mid X_t, \theta_{*,A_t} &\sim P(\cdot \mid X_t; \theta_{*,A_t}), \quad \forall t \in [n].
\end{aligned}
\tag{1}
$$

Figure 1: Graphical model of Eq. (1).

The model in Eq. (1) represents a Bayesian bandit, where the agent interacts with a bandit instance defined by $\theta_{*,i}$ over $n$ rounds (4-th line in Eq. (1)). These action parameters $\theta_{*,i}$ are drawn from the generative process in the first 3 lines of Eq. (1). In practice, Eq. (1) can be built by pre-training a diffusion model on offline estimates of the action parameters $\theta_{*,i}$ (Hsieh et al., 2023).

A natural goal for the agent in this Bayesian framework is to minimize its *Bayes regret* (Russo & Van Roy, 2014) that measures the expected performance across multiple bandit instances $\theta_* = (\theta_{*,i})_{i \in [K]}$,

$$
\mathcal{BR}(n) = \mathbb{E}\left[ \sum_{t=1}^n r(X_t, A_{t,*}; \theta_*) - r(X_t, A_t; \theta_*) \right],
\tag{2}
$$

---

**Algorithm 1** dTS: **d**iffusion **T**hompson **S**ampling

---

**Input:** Prior: $f_\ell, \ell \in [L]$, $\Sigma_\ell, \ell \in [L+1]$, and $P$.

**for** $t = 1, \dots, n$ **do**

     Sample $\psi_{t,L} \sim Q_{t,L}$ (requires fast approximate posterior update and sampling)

     **for** $\ell = L, \dots, 2$ **do**

         Sample $\psi_{t,\ell-1} \sim Q_{t,\ell-1}(\cdot \mid \psi_{t,\ell})$ (requires fast approximate posterior update and sampling)

     **for** $i = 1, \dots, K$ **do**

         Sample $\theta_{t,i} \sim P_{t,i}(\cdot \mid \psi_{t,1})$ (requires fast approximate posterior update and sampling)

     Take action $A_t = \operatorname{argmax}_{i \in [K]} r(X_t, i; \theta_t)$, where $\theta_t = (\theta_{t,i})_{i \in [K]}$

     Receive reward $Y_t \sim P(\cdot \mid X_t; \theta_{*,A_t})$ and update posteriors $Q_{t+1,\ell}$ and $P_{t+1,i}$.

---

where the expectation in Eq. (2) is taken over all random variables in Eq. (1). Here $r(x, i; \theta_*) = \mathbb{E}_{Y \sim P(\cdot | x; \theta_{*,i})}[Y]$ is the expected reward of action $i$ in context $x$ and $A_{t,*} = \arg\max_{i \in [K]} r(X_t, i; \theta_*)$ is the optimal action in round $t$. The Bayes regret is known to capture the benefits of using informative priors (Hong et al., 2022b;a; Aouali et al., 2023b), and hence it is suitable for our problem.

# 3 DIFFUSION CONTEXTUAL THOMPSON SAMPLING

We design a Thompson sampling algorithm that samples the latent and action parameters hierarchically (Lindley & Smith, 1972). Precisely, let $H_t = (X_k, A_k, Y_k)_{k \in [t-1]}$ be the history of all interactions up to round $t$ and let $H_{t,i} = (X_k, A_k, Y_k)_{\{k \in [t-1]; A_k = i\}}$ be the history of interactions *with action* $i$ up to round $t$. To motivate our algorithm, we decompose the posterior $\mathbb{P}(\theta_{*,i} = \theta \mid H_t)$ recursively as

$$\mathbb{P}(\theta_{*,i} = \theta \mid H_t) = \int_{\psi_{1:L}} Q_{t,L}(\psi_L) \prod_{\ell=2}^{L} Q_{t,\ell-1}(\psi_{\ell-1} \mid \psi_\ell) P_{t,i}(\theta \mid \psi_1) \, d\psi_{1:L}, \quad \text{where} \quad (3)$$

$Q_{t,L}(\psi_L) = \mathbb{P}(\psi_{*,L} = \psi_L \mid H_t)$ is the *latent-posterior* density of $\psi_{*,L} \mid H_t$. Moreover, for any $\ell \in [2 : L]$, $Q_{t,\ell-1}(\psi_{\ell-1} \mid \psi_\ell) = \mathbb{P}(\psi_{*,\ell-1} = \psi_{\ell-1} \mid H_t, \psi_{*,\ell} = \psi_\ell)$ is the *conditional latent-posterior* density of $\psi_{*,\ell-1} \mid H_t, \psi_{*,\ell} = \psi_\ell$. Finally, for any action $i \in [K]$, $P_{t,i}(\theta \mid \psi_1) = \mathbb{P}(\theta_{*,i} = \theta \mid H_{t,i}, \psi_{*,1} = \psi_1)$ is the *conditional action-posterior* density of $\theta_{*,i} \mid H_{t,i}, \psi_{*,1} = \psi_1$.

The decomposition in Eq. (3) inspires hierarchical sampling. In round $t$, we initially sample the $L$-th latent parameter as $\psi_{t,L} \sim Q_{t,L}(\cdot)$. Then, for $\ell \in [L]/\{1\}$, we sample the $\ell-1$-th latent parameter given that $\psi_{*,\ell} = \psi_{t,\ell}$, as $\psi_{t,\ell-1} \sim Q_{t,\ell-1}(\cdot \mid \psi_{t,\ell})$. Lastly, given that $\psi_{*,1} = \psi_{t,1}$, each action parameter is sampled *individually* as $\theta_{t,i} \sim P_{t,i}(\theta \mid \psi_{t,1})$. This is possible because action parameters $\theta_{*,i}$ are conditionally independent given $\psi_{*,1}$. This leads to Algorithm 1, named **d**iffusion **T**hompson **S**ampling (dTS). dTS requires sampling from the $K + L$ posteriors $P_{t,i}$ and $Q_{t,\ell}$. Thus we start by providing an efficient recursive scheme to express these posteriors using known quantities. We note that these expressions do not necessarily lead to closed-form posteriors and approximation might be needed. First, the conditional action-posterior $P_{t,i}(\cdot \mid \psi_1)$ can be written as

$$P_{t,i}(\theta \mid \psi_1) \propto \prod_{k \in S_{t,i}} P(Y_k \mid X_k; \theta) \mathcal{N}(\theta; f_1(\psi_1), \Sigma_1), \quad (4)$$

where $S_{t,i} = \{\ell \in [t-1], A_\ell = i\}$ are the rounds where the agent takes action $i$ up to round $t$. Moreover, let $\mathcal{L}_\ell(\psi_\ell) = \mathbb{P}(H_t \mid \psi_{*,\ell} = \psi_\ell)$ be the likelihood of observations up to round $t$ given that $\psi_{*,\ell} = \psi_\ell$. Then, for any $\ell \in [L]/\{1\}$, the $\ell - 1$-th conditional latent-posterior $Q_{t,\ell-1}(\cdot \mid \psi_\ell)$ is

$$Q_{t,\ell-1}(\psi_{\ell-1} \mid \psi_\ell) \propto \mathcal{L}_{\ell-1}(\psi_{\ell-1}) \mathcal{N}(\psi_{\ell-1}, f_\ell(\psi_\ell), \Sigma_\ell), \quad (5)$$

and $Q_{t,L}(\psi_L) \propto \mathcal{L}_L(\psi_L) \mathcal{N}(\psi_L, 0, \Sigma_{L+1})$. All the terms above are known, except the likelihoods $\mathcal{L}_\ell(\psi_\ell)$ for $\ell \in [L]$. These are computed recursively as follows. First, the basis of the recursion is

$$\mathcal{L}_1(\psi_1) = \prod_{i=1}^{K} \int_{\theta_i} \prod_{k \in S_{t,i}} P(Y_k \mid X_k; \theta_i) \mathcal{N}(\theta_i; f_1(\psi_1), \Sigma_1) \, d\theta_i. \quad (6)$$

Then for $\ell \in [L]/\{1\}$, the recursive step is $\mathcal{L}_\ell(\psi_\ell) = \int_{\psi_{\ell-1}} \mathcal{L}_{\ell-1}(\psi_{\ell-1}) \mathcal{N}(\psi_{\ell-1}; f_\ell(\psi_\ell), \Sigma_\ell) \, d\psi_{\ell-1}$.

All posterior expressions above use known quantities $(f_\ell, \Sigma_\ell, P(y \mid x; \theta))$. However, these expressions typically need to be approximated, except when the link functions $f_\ell$ are linear and the reward

distribution $P(\cdot \mid x; \theta)$ is linear-Gaussian, where closed-form solutions can be obtained with careful derivations. These approximations are not trivial, and prior studies often rely on computationally intensive approximate sampling algorithms. In the following sections, we explain how we derive our efficient approximations which are motivated by the closed-form solutions of linear instances.

### 3.1 POSTERIOR APPROXIMATION

The reward distribution is parameterized as a generalized linear model (GLM) (McCullagh & Nelder, 1989), allowing for non-linear rewards, which necessitates an approximation. We adopt an approach similar to the Laplace approximation, where a Gaussian density approximates the likelihood. Specifically, the reward distribution $P(\cdot \mid x; \theta)$ belongs to the exponential family with a mean function $g$. Then we approximate the likelihood as $\prod_{k \in S_{t,i}} P(Y_k \mid X_k; \theta) \approx \mathcal{N}(\theta; \hat{B}_{t,i}, \hat{G}_{t,i}^{-1})$, where $\hat{B}_{t,i}$ is the maximum likelihood estimate (MLE) and $\hat{G}_{t,i}$ is the Hessian of the negative log-likelihood:

$$\hat{B}_{t,i} = \arg\max_{\theta \in \mathbb{R}^d} \log \prod_{k \in S_{t,i}} P(Y_k \mid X_k; \theta), \quad \hat{G}_{t,i} = \sum_{k \in S_{t,i}} \dot{g}(X_k^\top \hat{B}_{t,i}) X_k X_k^\top . \quad (7)$$

where $S_{t,i} = \{\ell \in [t-1] : A_\ell = i\}$ represents the rounds where the agent selects action $i$ up to round $t$. Unlike Laplace, which approximates the entire posterior with a Gaussian, we only approximate the likelihood, allowing the approximate posterior to remain more complex (a diffusion model with updated parameters) than a Gaussian, as described next. After this initial approximation, we plug it in the action and latent posteriors, $P_{t,i}$ and $Q_{t,\ell}$, in Eqs. (4) and (5). This removes the non-linearity of the reward but still doesn't yield a closed-form solution due to the non-linearity in the link functions $f_\ell$. Thus, we apply another approximation inspired by the linear diffusion case where the link functions $f_\ell$ are linear, such as $f_\ell(\psi_\ell) = W_\ell \psi_\ell$ for $\ell \in [L]$, with $W_\ell \in \mathbb{R}^{d \times d}$ (see Appendix B.1). In that case, closed-form solutions can be derived (Appendix B.2), and we use these to construct efficient approximations by replacing the linear terms $W_\ell \psi_\ell$ with the more general term $f_\ell(\psi_\ell)$, resulting in highly efficient approximations (see Appendix C for details). Specifically, we approximate $P_{t,i}(\cdot \mid \psi_1) \approx \mathcal{N}(\cdot; \hat{\mu}_{t,i}, \hat{\Sigma}_{t,i})$, where

$$\hat{\Sigma}_{t,i}^{-1} = \Sigma_1^{-1} + \hat{G}_{t,i} \qquad \hat{\mu}_{t,i} = \hat{\Sigma}_{t,i}(\Sigma_1^{-1} f_1(\psi_1) + \hat{G}_{t,i} \hat{B}_{t,i}). \quad (8)$$

In the absence of samples, $G_{t,i} = 0_{d \times d}$. Thus, the approximate action posterior in Eq. (8) matches precisely the term $\mathcal{N}(f_1(\psi_1), \Sigma_1)$ in the diffusion prior in Eq. (1). Moreover, as more data is accumulated, $G_{t,i}$ increases, and the influence of the prior diminishes as $\hat{G}_{t,i} \hat{B}_{t,i}$ will dominate the prior term $\Sigma_1^{-1} f_1(\psi_1)$. Similarly, for $\ell \in [L]/\{1\}$, the $\ell - 1$-th conditional latent-posterior is approximated by a Gaussian distribution as $Q_{t,\ell-1}(\cdot \mid \psi_\ell) \approx \mathcal{N}(\bar{\mu}_{t,\ell-1}, \bar{\Sigma}_{t,\ell-1})$, where

$$\bar{\Sigma}_{t,\ell-1}^{-1} = \Sigma_\ell^{-1} + \bar{G}_{t,\ell-1} , \qquad \bar{\mu}_{t,\ell-1} = \bar{\Sigma}_{t,\ell-1}(\Sigma_\ell^{-1} f_\ell(\psi_\ell) + \bar{B}_{t,\ell-1}) , \quad (9)$$

and the $L$-th latent-posterior is $Q_{t,L}(\cdot) = \mathcal{N}(\bar{\mu}_{t,L}, \bar{\Sigma}_{t,L})$,

$$\bar{\Sigma}_{t,L}^{-1} = \Sigma_{L+1}^{-1} + \bar{G}_{t,L} , \qquad \bar{\mu}_{t,L} = \bar{\Sigma}_{t,L} \bar{B}_{t,L} . \quad (10)$$

Here, $\bar{G}_{t,\ell}$ and $\bar{B}_{t,\ell}$ for $\ell \in [L]$ are computed recursively. The basis of the recursion are

$$\bar{G}_{t,1} = \sum_{i=1}^{K} (\Sigma_1^{-1} - \Sigma_1^{-1} \hat{\Sigma}_{t,i} \Sigma_1^{-1}) , \qquad \bar{B}_{t,1} = \Sigma_1^{-1} \sum_{i=1}^{K} \hat{\Sigma}_{t,i} \hat{G}_{t,i} \hat{B}_{t,i} . \quad (11)$$

Then, the recursive step for $\ell \in [L]/\{1\}$ is,

$$\bar{G}_{t,\ell} = \Sigma_\ell^{-1} - \Sigma_\ell^{-1} \bar{\Sigma}_{t,\ell-1} \Sigma_\ell^{-1} , \qquad \bar{B}_{t,\ell} = \Sigma_\ell^{-1} \bar{\Sigma}_{t,\ell-1} \bar{B}_{t,\ell-1} . \quad (12)$$

Similarly, in the absence of samples, $Q_{t,\ell-1}$ in Eq. (9) precisely matches the term $\mathcal{N}(f_\ell(\psi_\ell), \Sigma_\ell)$ in the diffusion prior in Eq. (1). As more data is accumulated, the influence of this prior diminishes. Therefore, this approximation retains a key attribute of exact posteriors: they match the prior when there is no data, and the prior's effect diminishes as data accumulates.

Note that this approximate posterior is also a diffusion model with updated means and covariances. For instance, the latent-posterior means can be viewed as *updated link functions* $\hat{f}_{t,\ell}(\psi_\ell) = \bar{\mu}_{t,\ell-1} = \bar{\Sigma}_{t,\ell-1}(\Sigma_\ell^{-1} f_\ell(\psi_\ell) + \bar{B}_{t,\ell-1})$, and similarly for the *updated covariances* $\bar{\Sigma}_{t,\ell}$. Thus, this approximation results in a complex posterior (a diffusion model with updated parameters) without requiring heavy computations, and it is different from the Laplace approximation,

which approximates the entire posterior with a Gaussian distribution. Other approximations can be used, but they can be costly. We need fast updates and sampling from the posterior, both of which our approximation achieves. These two requirements may not be met by other methods. For example, optimizing a variational bound using the re-parameterization trick and Monte Carlo estimation would introduce a complex optimization problem into a bandit algorithm that needs to be updated in each interaction round. Appendix E.3 provides an experiment demonstrating that this approximation closely matches the exact posterior in that setting.

# 4 ANALYSIS

We analyze `dTS` asusming that: **(A1)** The rewards are linear $P(\cdot \mid x; \theta_{*,a}) = \mathcal{N}(\cdot; x^\top \theta_{*,a}, \sigma^2)$. **(A2)** The link functions $f_\ell$ are linear such as $f_\ell(\psi_{*,\ell}) = W_\ell \psi_{*,\ell}$ for $\ell \in [L]$, where $W_\ell \in \mathbb{R}^{d \times d}$ are *known mixing matrices*. This leads to a structure with $L$ layers of linear Gaussian relationships detailed in Appendix B.1. In particular, this leads to closed-form posteriors given in Appendix B.2 that inspired our approximation and enable theory similar to linear bandits (Agrawal & Goyal, 2013a). However, proofs are not the same, and technical challenges remain (explained in Appendix D).

Although our result holds for milder assumptions, we make additional simplifications for clarity and interpretability. We assume that **(A3)** Contexts satisfy $\|X_t\|_2^2 = 1$ for any $t \in [n]$. Note that **(A3)** can be relaxed to any contexts $X_t$ with bounded norms $\|X_t\|_2$. **(A4)** Mixing matrices and covariances satisfy $\lambda_1(W_\ell^\top W_\ell) = 1$ for any $\ell \in [L]$ and $\Sigma_\ell = \sigma_\ell^2 I_d$ for any $\ell \in [L+1]$. **(A4)** can be relaxed to positive definite covariances $\Sigma_\ell$ and arbitrary mixing matrices $W_\ell$. In particular, this is satisfied once we use a diffusion model parametrized with linear functions. In this section, we write $\tilde{\mathcal{O}}$ for the big-O notation up to polylogarithmic factors. We start by stating our bound for `dTS`.

**Theorem 4.1.** *Let* $\sigma_{\text{MAX}}^2 = \max_{\ell \in [L+1]} 1 + \frac{\sigma_\ell^2}{\sigma^2}$. *For any* $\delta \in (0,1)$, *the Bayes regret of* `dTS` *under* **(A1)**, **(A2)**, **(A3)** *and* **(A4)** *is bounded as*

$$\mathcal{BR}(n) \le \sqrt{2n\left(\mathcal{R}^{\text{ACT}}(n) + \sum_{\ell=1}^L \mathcal{R}_\ell^{\text{LAT}}\right)\log(1/\delta)} + cn\delta \,, \text{ with } c > 0 \text{ is constant and,} \quad (13)$$

$$\mathcal{R}^{\text{ACT}}(n) = c_0 dK \log\left(1 + \frac{n\sigma_1^2}{d}\right), \; c_0 = \frac{\sigma_1^2}{\log(1+\sigma_1^2)} \,, \quad \mathcal{R}_\ell^{\text{LAT}} = c_\ell d \log\left(1 + \frac{\sigma_{\ell+1}^2}{\sigma_\ell^2}\right), c_\ell = \frac{\sigma_{\ell+1}^2 \sigma_{\text{MAX}}^{2\ell}}{\log(1+\sigma_{\ell+1}^2)},$$

Eq. (13) holds for any $\delta \in (0,1)$. In particular, the term $cn\delta$ is constant when $\delta = 1/n$. Then, the bound is $\tilde{\mathcal{O}}\left(\sqrt{n(dK\sigma_1^2 + d\sum_{\ell=1}^L \sigma_{\ell+1}^2 \sigma_{\text{MAX}}^{2\ell})}\right)$, and this dependence on the horizon $n$ aligns with prior Bayes regret bounds. The bound comprises $L+1$ main terms, $\mathcal{R}^{\text{ACT}}(n)$ and $\mathcal{R}_\ell^{\text{LAT}}$ for $\ell \in [L]$. First, $\mathcal{R}^{\text{ACT}}(n)$ relates to action parameters learning, conforming to a standard form (Lu & Van Roy, 2019). Similarly, $\mathcal{R}_\ell^{\text{LAT}}$ is associated with learning the $\ell$-th latent parameter.

To include more structure, we propose the *sparsity* assumption **(A5)** $W_\ell = (\bar{W}_\ell, 0_{d,d-d_\ell})$, where $\bar{W}_\ell \in \mathbb{R}^{d \times d_\ell}$ for any $\ell \in [L]$. Note that **(A5)** is not an assumption when $d_\ell = d$ for any $\ell \in [L]$. Notably, **(A5)** incorporates a plausible structural characteristic that a diffusion model could capture.

**Proposition 4.2** (Sparsity). *Let* $\sigma_{\text{MAX}}^2 = \max_{\ell \in [L+1]} 1 + \frac{\sigma_\ell^2}{\sigma^2}$. *For any* $\delta \in (0,1)$, *the Bayes regret of* `dTS` *under* **(A1)**, **(A2)**, **(A3)**, **(A4)** *and* **(A5)** *is bounded as*

$$\mathcal{BR}(n) \le \sqrt{2n\left(\mathcal{R}^{\text{ACT}}(n) + \sum_{\ell=1}^L \tilde{\mathcal{R}}_\ell^{\text{LAT}}\right)\log(1/\delta)} + cn\delta \,, \text{ with } c > 0 \text{ is constant,} \quad (14)$$

$$\mathcal{R}^{\text{ACT}}(n) = c_0 dK \log\left(1 + \frac{n\sigma_1^2}{d}\right), c_0 = \frac{\sigma_1^2}{\log(1+\sigma_1^2)} \,, \quad \tilde{\mathcal{R}}_\ell^{\text{LAT}} = c_\ell d_\ell \log\left(1 + \frac{\sigma_{\ell+1}^2}{\sigma_\ell^2}\right), c_\ell = \frac{\sigma_{\ell+1}^2 \sigma_{\text{MAX}}^{2\ell}}{\log(1+\sigma_{\ell+1}^2)}.$$

From Proposition 4.2, our bounds scales as

$$\mathcal{BR}(n) = \tilde{\mathcal{O}}\left(\sqrt{n(dK\sigma_1^2 + \sum_{\ell=1}^L d_\ell \sigma_{\ell+1}^2 \sigma_{\text{MAX}}^{2\ell})}\right). \quad (15)$$

The Bayes regret bound has a clear interpretation: if the true environment parameters are drawn from the prior, then the expected regret of an algorithm stays below that bound. Consequently, a less informative prior (such as high variance) leads to a more challenging problem and thus a higher

bound. Then, smaller values of $K$, $L$, $d$ or $d_\ell$ translate to fewer parameters to learn, leading to lower regret. The regret also decreases when the initial variances $\sigma_\ell^2$ decrease. These dependencies are common in Bayesian analysis, and empirical results match them. The reader might question the dependence of our bound on both $L$ and $K$. We will address this next.

**Why the bound increases with $K$?** This arises due to our conditional learning of $\theta_{*,i}$ given $\psi_{*,1}$. Rather than assuming deterministic linearity, $\theta_{*,i} = W_1\psi_{*,1}$, we account for stochasticity by modeling $\theta_{*,i} \sim \mathcal{N}(W_1\psi_{*,1}, \sigma_1^2 I_d)$. This makes dTS robust to misspecification scenarios where $\theta_{*,i}$ is not perfectly linear with respect to $\psi_{*,1}$, at the cost of additional learning of $\theta_{*,i} \mid \psi_{*,1}$. If we were to assume deterministic linearity ($\sigma_1 = 0$), our regret bound would scale with $L$ only.

**Why the bound increases with $L$?** This is because increasing the number of layers $L$ adds more initial uncertainty due to the additional covariance introduced by the extra layers. However, this does not imply that we should always use $L = 1$ (the minimum possible $L$). Precisely, the theoretical results predict that regret increases with $L$ when the true prior distribution matches a diffusion model of depth $L$, as increasing $L$ reflects a more complex action parameter distribution and hence a more complex bandit problem. However, in practice, when $L$ is small, the pre-trained diffusion model may be too simple to capture the true prior distribution, violating the assumptions of our theory. Increasing $L$ improves the pre-trained model's quality, reducing regret. Once $L$ is large enough and the pre-trained model adequately captures the true prior distribution, the theoretical assumptions hold, and regret begins to increase with $L$, as predicted. This is validated empirically in Fig. 4b.

**Technical contributions.** dTS uses hierarchical sampling. Thus the marginal posterior distribution of $\theta_{*,i} \mid H_t$ is not explicitly defined. The first contribution is deriving $\theta_{*,i} \mid H_t$ using the total covariance decomposition combined with an induction proof, as our posteriors were derived recursively. Unlike standard analyses where the posterior distribution of $\theta_{*,i} \mid H_t$ is predetermined due to the absence of latent parameters, our method necessitates this recursive total covariance decomposition. Moreover, in standard proofs, we need to quantify the increase in posterior precision for the action taken $A_t$ in each round $t \in [n]$. However, in dTS, our analysis extends beyond this. We not only quantify the posterior information gain for the taken action but also for every latent parameter, since they are also learned. To elaborate, we use our recursive posteriors that connect the posterior covariance of each latent parameter $\psi_{*,\ell}$ with the covariance of the posterior action parameters $\theta_{*,i}$. This allows us to propagate the information gain associated with the action taken in round $A_t$ to all latent parameters $\psi_{*,\ell}$, for $\ell \in [L]$ by induction. More technical details are provided in Appendix D.

## 4.1 Discussion

**Computational benefits.** Action correlations prompt an intuitive approach: marginalize all latent parameters and maintain a joint posterior of $(\theta_{*,i})_{i\in[K]} \mid H_t$. Unfortunately, this is computationally inefficient for large action spaces. To illustrate, suppose that all posteriors are multivariate Gaussians. Then maintaining the joint posterior $(\theta_{*,i})_{i\in[K]} \mid H_t$ necessitates converting and storing its $dK \times dK$-dimensional covariance matrix, leading to $\mathcal{O}(K^3 d^3)$ and $\mathcal{O}(K^2 d^2)$ time and space complexities. In contrast, the time and space complexities of dTS are $\mathcal{O}((L + K)d^3)$ and $\mathcal{O}((L + K)d^2)$. This is because dTS requires converting and storing $L + K$ covariance matrices, each being $d \times d$-dimensional. The improvement is huge when $K \gg L$, which is common in practice. Certainly, a more straightforward way to enhance computational efficiency is to discard latent parameters and maintain $K$ individual posteriors, each relating to an action parameter $\theta_{*,i} \in \mathbb{R}^d$ (LinTS). This improves time and space complexity to $\mathcal{O}(Kd^3)$ and $\mathcal{O}(Kd^2)$. However, LinTS maintains independent posteriors and fails to capture the correlations among actions; it only models $\theta_{*,i} \mid H_{t,i}$ rather than $\theta_{*,i} \mid H_t$ as done by dTS. Consequently, LinTS incurs higher regret due to the information loss caused by unused interactions of similar actions. Our regret bound and empirical results reflect this aspect.

**Statistical benefits.** We do not provide a matching lower bound. The only Bayesian lower bound that we know of is $\Omega(\log^2(n))$ for a much simpler $K$-armed bandit (Lai, 1987, Theorem 3). All seminal works on Bayesian bandits do not match it and providing such lower bounds on Bayes regret is still relatively unexplored (even in standard settings) compared to the frequentist one. Also, a min-max lower bound of $\Omega(d\sqrt{n})$ was given by Dani et al. (2008). In this work, we argue that our bound reflects the overall structure of the problem by comparing dTS to algorithms that only partially use the structure or do not use it at all as follows.

When the link functions are linear, we can transform the diffusion prior into a Bayesian linear model (`LinTS`) by marginalizing out the latent parameters; in which case the prior on action parameters becomes $\theta_{*,i} \sim \mathcal{N}(0, \Sigma)$, with the $\theta_{*,i}$ being not necessarily independent, and $\Sigma$ is the marginal initial covariance of action parameters and it writes $\Sigma = \sigma_1^2 I_d + \sum_{\ell=1}^{L} \sigma_{\ell+1}^2 B_\ell B_\ell^\top$ with $B_\ell = \prod_{k=1}^{\ell} W_k$. Then, it is tempting to directly apply `LinTS` to solve our problem. This approach will induce higher regret because the additional uncertainty of the latent parameters is accounted for in $\Sigma$ despite integrating them. This causes the *marginal* action uncertainty $\Sigma$ to be much higher than the *conditional* action uncertainty $\sigma_1^2 I_d$, since we have $\Sigma = \sigma_1^2 I_d + \sum_{\ell=1}^{L} \sigma_{\ell+1}^2 B_\ell B_\ell^\top \succcurlyeq \sigma_1^2 I_d$. This discrepancy leads to higher regret, especially when $K$ is large. This is due to `LinTS` needing to learn $K$ independent $d$-dimensional parameters, each with a considerably higher initial covariance $\Sigma$. This is also reflected by our regret bound. To simply comparisons, suppose that $\sigma \geq \max_{\ell \in [L+1]} \sigma_\ell$ so that $\sigma_{\text{MAX}}^2 \leq 2$. Then the regret bounds of `dTS` (where we bound $\sigma_{\text{MAX}}^{2\ell}$ by $2^\ell$) and `LinTS` read

$$\texttt{dTS} : \tilde{\mathcal{O}}\big(\sqrt{n(dK\sigma_1^2 + \sum_{\ell=1}^{L} d_\ell \sigma_{\ell+1}^2 2^\ell)}\big), \qquad \texttt{LinTS} : \tilde{\mathcal{O}}\big(\sqrt{ndK(\sigma_1^2 + \sum_{\ell=1}^{L} \sigma_{\ell+1}^2)}\big).$$

Then regret improvements are captured by the variances $\sigma_\ell$ and the sparsity dimensions $d_\ell$, and we proceed to illustrate this through the following scenarios.

**(I) Decreasing variances.** Assume that $\sigma_\ell = 2^\ell$ for any $\ell \in [L+1]$. Then, the regrets become

$$\texttt{dTS} : \tilde{\mathcal{O}}\big(\sqrt{n(dK + \sum_{\ell=1}^{L} d_\ell 4^\ell)}\big), \qquad \texttt{LinTS} : \tilde{\mathcal{O}}\big(\sqrt{ndK2^L}\big)$$

Now to see the order of gain, assume the problem is high-dimensional ($d \gg 1$), and set $L = \log_2(d)$ and $d_\ell = \lfloor \frac{d}{2^\ell} \rfloor$. Then the regret of `dTS` becomes $\tilde{\mathcal{O}}\big(\sqrt{nd(K + L)}\big)$, and hence the multiplicative factor $2^L$ in `LinTS` is removed and replaced with a smaller additive factor $L$.

**(II) Constant variances.** Assume that $\sigma_\ell = 1$ for any $\ell \in [L+1]$. Then, the regrets become

$$\texttt{dTS} : \tilde{\mathcal{O}}\big(\sqrt{n(dK + \sum_{\ell=1}^{L} d_\ell 2^\ell)}\big), \qquad \texttt{LinTS} : \tilde{\mathcal{O}}\big(\sqrt{ndKL}\big)$$

Similarly, let $L = \log_2(d)$, and $d_\ell = \lfloor \frac{d}{2^\ell} \rfloor$. Then `dTS`'s regret is $\tilde{\mathcal{O}}\big(\sqrt{nd(K + L)}\big)$. Thus the multiplicative factor $L$ in `LinTS` is removed and replaced with the additive factor $L$. By comparing this to **(I)**, the gain with decreasing variances is greater than with constant ones. In general, diffusion models use decreasing variances (Ho et al., 2020) and hence we expect great gains in practice. All observed improvements in this section could become even more pronounced when employing non-linear diffusion models. In our theory, we used linear diffusion models, and yet we can already discern substantial differences. Moreover, under non-linear diffusion Eq. (1), the latent parameters cannot be analytically marginalized, making `LinTS` with exact marginalization inapplicable. Finally, Appendix D.7 provide an additional comparison and connection to hierarchies with two levels.

**Large action space aspect and regret independent of $K$?** `dTS`'s regret bound scales with $K\sigma_1^2$ instead of $K \sum_\ell \sigma_\ell^2$, which is particularly beneficial when $\sigma_1$ is small, as often seen in diffusion models. Both our regret bound and experiments demonstrate that `dTS` outperforms `LinTS` more significantly as the action space grows. Previous studies (Foster et al., 2020; Xu & Zeevi, 2020; Zhu et al., 2022) proposed bandit algorithms whose regret do not scale with $K$, but our setting is fundamentally different, explaining our inherent dependence on $K$ when $\sigma_1 > 0$. Specifically, they assume a reward function $r(x, i; \theta_*) = \phi(x, i)^\top \theta_*$, with a shared $\theta_* \in \mathbb{R}^d$ and a known mapping $\phi$. In contrast, we consider $r(x, i; \theta_*) = x^\top \theta_{*,i}$, where $\theta_* = (\theta_{*,i})_{i \in [K]} \in \mathbb{R}^{dK}$, requiring the learning of $K$ separate $d$-dimensional action parameters. Using our proof techniques, we can show that `dTS`'s regret is independent of $K$ in their setting, assuming the availability of $\phi$. Our setting reflects practical scenarios like recommendation systems where each product is represented by a unique embedding.

## 5 EXPERIMENTS

We evaluate `dTS` using both synthetic and MovieLens problems. In our experiments, we run 50 random simulations and plot the average regret with its standard error.

### 5.1 WHEN THE TRUE PRIOR IS A DIFFUSION MODEL

Synthetic bandit problems are generated from the diffusion model in Eq. (1) with both linear and non-linear rewards. Linear rewards follow $P(\cdot \mid x; \theta_{*,a}) = \mathcal{N}(x^\top \theta_{*,a}, 1)$, while non-linear rewards

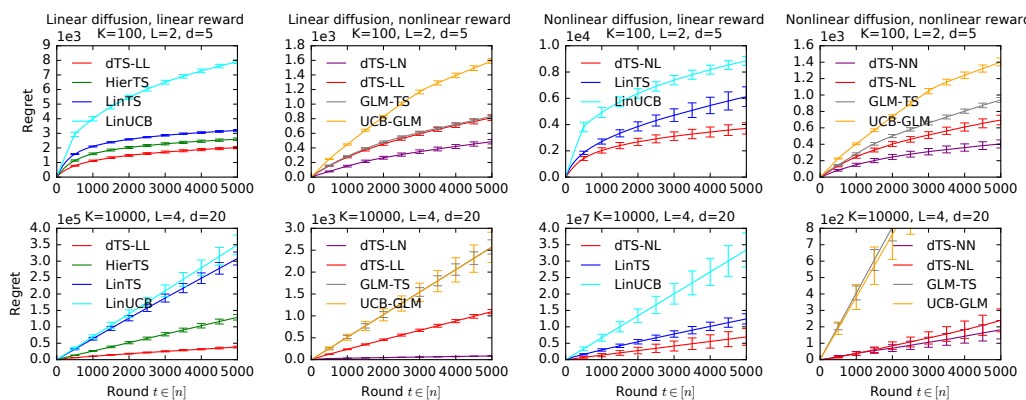

Figure 2: Regret of dTS with varying diffusion and reward models and varying parameters $d, K, L$.

are binary from $P(\cdot \mid x; \theta_{*,a}) = \text{Ber}(g(x^\top \theta_{*,a}))$, with $g$ as the sigmoid function. Covariances are $\Sigma_\ell = I_d$, and contexts $X_t$ are uniformly drawn from $[-1, 1]^d$. We vary $d \in \{5, 20\}$, $L \in \{2, 4\}$, $K \in \{10^2, 10^4\}$, and set the horizon to $n = 5000$, considering both linear and non-linear models.

**Linear diffusion.** We consider Eq. (1) with $f_\ell(\psi) = W_\ell \psi$, where $W_\ell$ uniformly drawn from $[-1, 1]^{d \times d}$. Sparsity is introduced by zeroing the last $d_\ell$ columns of $W_\ell$ as $W_\ell = (\bar{W}_\ell, 0_{d,d-d_\ell})$. For $d = 5$ and $L = 2$, $(d_1, d_2) = (5, 2)$; for $d = 20$ and $L = 4$, $(d_1, d_2, d_3, d_4) = (20, 10, 5, 2)$.

**Non-linear diffusion.** We consider Eq. (1) where $f_\ell$ are 2-layer neural networks with random weights in $[-1, 1]$, ReLU activation, and hidden layers of size $h = 20$ for $d = 5$, and $h = 60$ for $d = 20$.

**Baselines.** For linear rewards, we use LinUCB (Abbasi-Yadkori et al., 2011), LinTS (Agrawal & Goyal, 2013a), and HierTS (Hong et al., 2022b), marginalizing out all latent parameters except $\psi_{*,L}$, which corresponds to HierTS-1 in Appendix D.7. For non-linear rewards, we include UCB-GLM (Li et al., 2017) and GLM-TS (Chapelle & Li, 2012). We exclude GLM-UCB (Filippi et al., 2010) due to high regret and HierTS as it's designed for linear rewards. We name dTS as dTS-dr, where d refers to diffusion type (L for linear, N for non-linear) and r indicates reward type (L for linear, N for non-linear). For example, dTS-LL signifies dTS in linear diffusion with linear rewards.

**Results and interpretations.** Results are shown in Fig. 2 and we make the following observations:

**1) dTS demonstrates superior performance (Fig. 2).** dTS consistently outperforms the baselines across all settings, including the four combinations of linear/non-linear diffusion and reward (columns in Fig. 2) and both bandit settings with varying $K$, $L$, and $d$ (rows in Fig. 2).

**2) Latent diffusion structure may be more important than the reward distribution.** When rewards are non-linear (second and fourth columns in Fig. 2), we include variants of dTS that use the correct diffusion prior but the wrong reward distribution, applying linear-Gaussian instead of logistic-Bernoulli (dTS-LL in the second column and dTS-NL in the fourth). Despite the reward misspecification, these variants outperform models using the correct reward distribution but ignoring the latent diffusion structure, such as GLM-TS and UCB-GLM. This highlights the importance of accounting for latent structure, which can be more critical than an accurate reward distribution.

**3) Performance gap between dTS and LinTS widens as $K$ increases (Fig. 3a).** To show dTS's improved scalability, we evaluate its performance with varying values of $K \in [10, 5 \times 10^4]$, in the linear diffusion and rewards setting. Fig. 3a shows the final cumulative regret for varying $K$ values for both dTS-LL and LinTS, revealing a widening performance gap as $K$ increases.

**4) Regret scaling with $K$, $d$ and $L$ matches our theory (Fig. 3b).** We assess the effect of the number of actions $K$, context dimension $d$, and diffusion depth $L$ on dTS's regret. Using the linear diffusion and rewards setting, for which we have derived a Bayes regret upper bound, we plot dTS-LL's regret across varying values of $K \in \{10, 100, 500, 1000\}$, $d \in \{5, 10, 15, 20\}$, and $L \in \{2, 4, 5, 6\}$ in Fig. 3b. As predicted by our theory, the empirical regret increases with larger values of $K$, $d$, or $L$, as these make the learning problem more challenging, leading to higher regret.

**5) Diffusion prior misspecification (Fig. 3c).** Here, dTS's diffusion prior parameters differ from the true diffusion prior. In the linear diffusion and reward setting, we replace the true parameters $W_\ell$

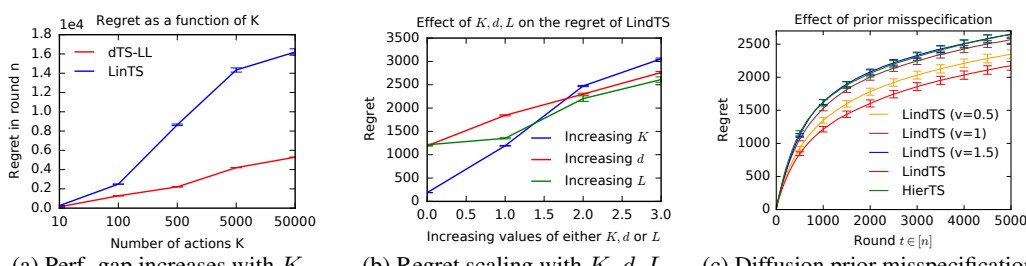

Figure 3: Effect of various factors on `dTS`'s performance.

(a) Perf. gap increases with $K$. (b) Regret scaling with $K$, $d$, $L$. (c) Diffusion prior misspecification.

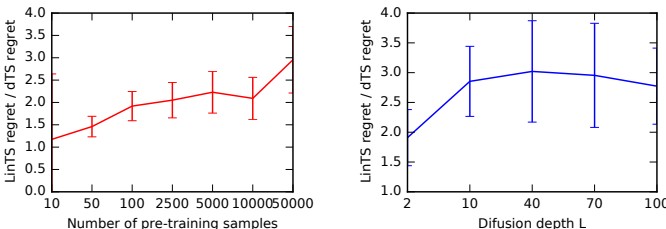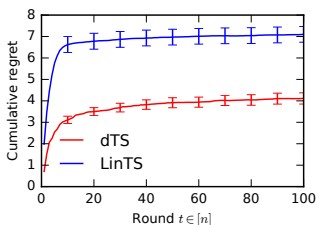

(a) Ratio of `LinTS`/`dTS` cumulative regret in the last round with varying pre-training sample size in $[10, 5 \times 10^4]$. **Higher values mean a bigger performance gap**.

(b) Ratio of `LinTS`/`dTS` cumulative regret in the last round with varying diffusion depth $L$ in $[2, 100]$. **Higher values mean a bigger performance gap**.

(c) Regret of `dTS` in **MovieLens**. The diffusion model with $L = 40$ is pre-trained on embeddings obtained by low-rank factorization of MovieLens rating matrix.

Figure 4: **(a) and (b):** Impact of pre-training sample size and diffusion depth $L$ for the **Swiss roll data**. **(c):** Regret of `dTS` in **MovieLens**.

and $\Sigma_\ell$ with misspecified ones, $W_\ell + \epsilon_1$ and $\Sigma_\ell + \epsilon_2$, where $\epsilon_1$ and $\epsilon_2$ are uniformly sampled from $[v, v + 0.5]^{d \times d}$, with $v$ controlling the misspecification level. We vary $v \in \{0.5, 1, 1.5\}$ and assess `dTS`'s performance, comparing it to the well-specified `dTS-LL` and the strongest baseline in this fully-linear setting, `HierTS`. As shown in Fig. 3c, `dTS`'s performance decreases with increasing misspecification but remains superior to the baseline, except at $v = 1.5$, where their performances are comparable. Additional misspecification experiments are presented in Section 5.2, where the bandit environment is not sampled from a diffusion model.

## 5.2 EFFECT OF PRE-TRAINING WHEN THE TRUE PRIOR IS NOT A DIFFUSION MODEL

**Swiss roll data.** Unlike previous experiments, the true action parameters are now sampled from the Swiss roll distribution (see Fig. 5 in Appendix E.1), rather than from a diffusion model. The diffusion model used by `dTS` is pre-trained on samples from this distribution, with the offline pre-training procedure described in Appendix E.2. Fig. 4a shows that larger sample sizes increase the performance gap between `dTS` and `LinTS`. More samples improve the estimation of the diffusion prior (see Fig. 5 in Appendix E.1), leading to better `dTS` performance. Notably, comparable performance was achieved with as few as 10 samples, and `dTS` outperformed `LinTS` by a factor of 1.5 with just 50 samples. While more samples may be required for more complex problems, `LinTS` would also struggle in such cases. Therefore, we expect these gains to be even more significant in more challenging settings.

We studied the effect of the pre-trained diffusion model depth $L$ and found that $L \approx 40$ yields the best performance, with a drop beyond that point (Fig. 4b). While our theory doesn't apply directly here, as it assumes a linear diffusion model, it still offers some intuition on the decreased performance for $L > 40$. The theorem shows `dTS`'s regret bound increases with $L$ when the true distribution is a diffusion model. For small $L$, the pre-trained model doesn't fully capture the true distribution, making the theorem inapplicable, but at $L \approx 40$, the distribution is nearly captured, and further increases in $L$ lead to higher regret, consistent with our theory.

**MovieLens data.** We also evaluate `dTS` using the standard MovieLens setting. In this semi-synthetic experiment, a user is sampled from the rating matrix in each interaction round, and the reward is the rating the user gives to a movie (see Clavier et al. (2023), Section 5) for details about this setting).

Here, the true distribution of action parameters is unknown and not a diffusion model. The diffusion model is pre-trained on offline estimates of action parameters obtained through low-rank factorization of the rating matrix. Fig. 4c demonstrates that `dTS` outperforms `LinTS` in this setting.

## 6 CONCLUSION

We use a pre-trained diffusion model as a strong and flexible prior for `dTS`. Diffusion model pre-training relies on offline data which is often widely available. This diffusion model is then sequentially refined through online interactions using our posterior approximation. This approximation allows fast sampling and updating of the posterior while performing very well empirically. `dTS` regret is bounded in a simple linear instance. Limitations and future research, broader impact and computational resources used are discussed in Appendices F to H, respectively.

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

## SUPPLEMENTARY MATERIALS

**Notation.** For any positive integer $n$, we define $[n] = \{1, 2, ..., n\}$. Let $v_1, \ldots, v_n \in \mathbb{R}^d$ be $n$ vectors, $(v_i)_{i \in [n]} \in \mathbb{R}^{nd}$ is the $nd$-dimensional vector obtained by concatenating $v_1, \ldots, v_n$. For any matrix $A \in \mathbb{R}^{d \times d}$, $\lambda_1(A)$ and $\lambda_d(A)$ denote the maximum and minimum eigenvalues of $A$, respectively. Finally, we write $\tilde{\mathcal{O}}$ for the big-O notation up to polylogarithmic factors.

**Table of notations.**

Table 1: Notation.

| Symbol | Definition |
|---|---|
| $n$ | Learning horizon |
| $\mathcal{X}$ | Context space |
| $K$ | Number of actions |
| $[K]$ | Set of actions |
| $d$ | Dimension of contexts and action parameters $d$ |
| $\theta_{*,i}$ | $d$-dimensional parameter of action $i \in [K]$ |
| $P(\cdot \mid x; \theta_{*,a})$ | Reward distribution of context $x$ and action $a$ |
| $r(x, a; \theta_*)$ | Reward function of context $x$ and action $a$ |
| $\mathcal{BR}(n)$ | Bayes regret after $n$ interactions |
| $\mathcal{N}(\mu, \Sigma)$ | Multivariate Gaussian distribution of parameters $\mu$ and $\Sigma$ |
| $\mathcal{N}(\cdot; \mu, \Sigma)$ | Multivariate Gaussian density of parameters $\mu$ and $\Sigma$ |
| $L$ | Diffusion model depth |
| $\psi_{*,\ell}$ | $\ell$-th $d$-dimensional latent parameter |
| $f_\ell$ | Link functions of the diffusion model |
| $\Sigma_\ell$ | Covariances of the link function |
| $H_t$ | History of interactions |
| $P_{t,i}$ | action-posterior density of $\theta_{*,i} \mid H_t$ |
| $Q_{t,\ell-1}$ | Latent-posterior density of $\psi_{*,\ell-1} \mid \psi_{*,\ell}, H_t$ |

## A  EXTENDED RELATED WORK

**Thompson sampling (TS)** operates within the Bayesian framework and it involves specifying a prior/likelihood model. In each round, the agent samples unknown model parameters from the current posterior distribution. The chosen action is the one that maximizes the resulting reward. TS is naturally randomized, particularly simple to implement, and has highly competitive empirical performance in both simulated and real-world problems (Russo & Van Roy, 2014; Chapelle & Li, 2012). Regret guarantees for the TS heuristic remained open for decades even for simple models. Recently, however, significant progress has been made. For standard multi-armed bandits, TS is optimal in the Beta-Bernoulli model (Kaufmann et al., 2012; Agrawal & Goyal, 2013b), Gaussian-Gaussian model (Agrawal & Goyal, 2013b), and in the exponential family using Jeffrey's prior (Korda et al., 2013). For linear bandits, TS is nearly-optimal (Russo & Van Roy, 2014; Agrawal & Goyal, 2017; Abeille & Lazaric, 2017). In this work, we build TS upon complex diffusion priors and analyze the resulting Bayes regret (Russo & Van Roy, 2014; Neu et al., 2022; Gouverneur et al., 2023) in the linear contextual bandit setting.

**Decision-making with diffusion models** gained attention recently, especially in offline learning (Ajay et al., 2022; Janner et al., 2022; Wang et al., 2022). However, their application in online learning was only examined by Hsieh et al. (2023), which focused on meta-learning in multi-armed bandits without theoretical guarantees. In this work, we expand the scope of Hsieh et al. (2023) to encompass the broader contextual bandit framework. In particular, we provide theoretical analysis for linear instances, effectively capturing the advantages of using diffusion models as priors in contextual Thompson sampling. These linear cases are particularly captivating due to closed-form posteriors, enabling both theoretical analysis and computational efficiency; an important practical consideration.

**Hierarchical Bayesian bandits** (Bastani et al., 2019; Kveton et al., 2021; Basu et al., 2021; Simchowitz et al., 2021; Wan et al., 2021; Hong et al., 2022b; Peleg et al., 2022; Wan et al., 2022; Aouali et al., 2023b) applied TS to simple graphical models, wherein action parameters are generally sampled from a Gaussian distribution centered at a single latent parameter. These works mostly span meta- and multi-task learning for multi-armed bandits, except in cases such as Aouali et al. (2023b); Hong et al. (2022a) that consider the contextual bandit setting. Precisely, Aouali et al. (2023b) assume that action parameters are sampled from a Gaussian distribution centered at a linear mixture of multiple latent parameters. On the other hand, Hong et al. (2022a) applied TS to a graphical model represented by a tree. Our work can be seen as an extension of all these works to much more complex graphical models, for which both theoretical and algorithmic foundations are developed. Note that the settings

in most of these works can be recovered with specific choices of the diffusion depth $L$ and functions $f_\ell$. This attests to the modeling power of dTS.

**Approximate Thompson sampling** is a major problem in the Bayesian inference literature. This is because most posterior distributions are intractable, and thus practitioners must resort to sophisticated computational techniques such as Markov chain Monte Carlo (Kruschke, 2010). Prior works (Riquelme et al., 2018; Chapelle & Li, 2012; Kveton et al., 2020) highlight the favorable empirical performance of approximate Thompson sampling. Particularly, (Kveton et al., 2020) provide theoretical guarantees for Thompson sampling when using the Laplace approximation in generalized linear bandits (GLB). In our context, we incorporate approximate sampling when the reward exhibits non-linearity. While our approximation does not come with formal guarantees, it enjoys strong practical performance. An in-depth analysis of this approximation is left as a direction for future works. Similarly, approximating the posterior distribution when the diffusion model is non-linear as well as analyzing it is an interesting direction of future works.

**Bandits with underlying structure** also align with our work, where we assume a structured relationship among actions, captured by a diffusion model. In latent bandits (Maillard & Mannor, 2014; Hong et al., 2020), a single latent variable indexes multiple candidate models. Within structured finite-armed bandits (Lattimore & Munos, 2014; Gupta et al., 2018), each action is linked to a known mean function parameterized by a common latent parameter. This latent parameter is learned. TS was also applied to complex structures (Yu et al., 2020; Gopalan et al., 2014). However, simultaneous computational and statistical efficiencies aren't guaranteed. Meta- and multi-task learning with upper confidence bound (UCB) approaches have a long history in bandits (Azar et al., 2013; Gentile et al., 2014; Deshmukh et al., 2017; Cella et al., 2020). These, however, often adopt a frequentist perspective, analyze a stronger form of regret, and sometimes result in conservative algorithms. In contrast, our approach is Bayesian, with analysis centered on Bayes regret. Remarkably, our algorithm, dTS, performs well as analyzed without necessitating additional tuning. Finally, **Low-rank bandits** (Hu et al., 2021; Cella et al., 2022; Yang et al., 2020) also relate to our linear diffusion model when $L = 1$. Broadly, there exist two key distinctions between these prior works and the special case of our model (linear diffusion model with $L = 1$). First, they assume $\theta_{*,i} = W_1 \psi_{*,1}$, whereas we incorporate additional uncertainty in the covariance $\Sigma_1$ to account for possible misspecification as $\theta_{*,i} = \mathcal{N}(W_1 \psi_{*,1}, \Sigma_1)$. Consequently, these algorithms might suffer linear regret due to model misalignment. Second, we assume that the mixing matrix $W_1$ is available and pre-learned offline, whereas they learn it online. While this is more general, it leads to computationally expensive methods that are difficult to employ in a real-world online setting.

**Large action spaces.** Roughly speaking, the regret bound of dTS scales with $K\sigma_1^2$ rather than $K \sum_\ell \sigma_\ell^2$. This is particularly beneficial when $\sigma_1$ is small, a common scenario in diffusion models with decreasing variances. A notable case is when $\sigma_1 = 0$, where the regret becomes independent of $K$. Also, our analysis (Section 4.1) indicates that the gap in performance between dTS and LinTS becomes more pronounced when the number of action increases, highlighting dTS's suitability for large action spaces. Note that some prior works (Foster et al., 2020; Xu & Zeevi, 2020; Zhu et al., 2022) proposed bandit algorithms that do not scale with $K$. However, our setting differs significantly from theirs, explaining our inherent dependency on $K$ when $\sigma_1 > 0$. Precisely, they assume a reward function of $r(x, i) = \phi(x, i)^\top \theta_*$, with a shared $\theta_* \in \mathbb{R}^d$ across actions and a known mapping $\phi$. In contrast, we consider $r(x, i) = x^\top \theta_{*,i}$, requiring the learning of $K$ separate $d$-dimensional action parameters. In their setting, with the availability of $\phi$, the regret of dTS would similarly be independent of $K$. However, obtaining such a mapping $\phi$ can be challenging as it needs to encapsulate complex context-action dependencies. Notably, our setting reflects a common practical scenario, such as in recommendation systems where each product is often represented by its embedding. In summary, the dependency on $K$ is more related to our setting than the method itself, and dTS would scale with $d$ only in their setting. Note that dTS is both computationally and statistically efficient (Section 4.1). This becomes particularly notable in large action spaces. Our empirical results in Fig. 2, notably with $K = 10^4$, demonstrate that dTS significantly outperforms the baselines. More importantly, the performance gap between dTS and these baselines is larger when the number of actions ($K$) increases, highlighting the improved scalability of dTS to large action spaces.

# B POSTERIOR DERIVATIONS FOR LINEAR DIFFUSION MODELS

## B.1 LINEAR DIFFUSION MODEL

Here, we assume the link functions $f_\ell$ are linear such as $f_\ell(\psi_{*,\ell}) = W_\ell \psi_{*,\ell}$ for $\ell \in [L]$, where $W_\ell \in \mathbb{R}^{d \times d}$ are *known mixing matrices*. Then, Eq. (1) becomes a linear Gaussian system (LGS) (Bishop, 2006) and can be summarized as follows

$$\psi_{*,L} \sim \mathcal{N}(0, \Sigma_{L+1}), \tag{16}$$
$$\psi_{*,\ell-1} \mid \psi_{*,\ell} \sim \mathcal{N}(W_\ell \psi_{*,\ell}, \Sigma_\ell), \qquad \forall \ell \in [L]/\{1\},$$
$$\theta_{*,i} \mid \psi_{*,1} \sim \mathcal{N}(W_1 \psi_{*,1}, \Sigma_1), \qquad \forall i \in [K],$$
$$Y_t \mid X_t, \theta_{*,A_t} \sim P(\cdot \mid X_t; \theta_{*,A_t}), \qquad \forall t \in [n].$$

This model is important, both in theory and practice. For theory, it leads to closed-form posteriors when the reward distribution is linear-Gaussian as $P(\cdot \mid x; \theta_{*,i}) = \mathcal{N}(\cdot; x^\top \theta_{*,i}, \sigma^2)$. This allows bounding the Bayes regret of dTS. For practice, the posterior expressions are used to motivate efficient approximations for the general case in Eq. (1) as we show in Section 3.1.

In this section, we derive the $K + L$ posteriors $P_{t,i}$ and $Q_{t,\ell}$, for which we provide the full expressions in Appendix B.2. In our proofs, $p(x) \propto f(x)$ means that the probability density $p$ satisfies $p(x) = \frac{f(x)}{Z}$ for any $x \in \mathbb{R}^d$, where $Z$ is a normalization constant. In particular, we extensively use that if $p(x) \propto \exp[-\frac{1}{2}x^\top \Lambda x + x^\top m]$, where $\Lambda$ is positive definite. Then $p$ is the multivariate Gaussian density with covariance $\Sigma = \Lambda^{-1}$ and mean $\mu = \Sigma m$. These are standard notations and techniques to manipulate Gaussian distributions (Koller & Friedman, 2009, Chapter 7).

## B.2 POSTERIOR EXPRESSIONS FOR LINEAR DIFFUSION MODELS

In this section, we consider the linear link function case in Appendix B.1, and the proofs are provided in Appendices B.3 and B.4. Recall that we posit that the reward distribution is parameterized as a generalized linear model (GLM) (McCullagh & Nelder, 1989), allowing for non-linear rewards. As a result, despite linearity in link functions, the non-linearity in rewards makes it challenging to obtain closed-form posteriors. However, since this non-linearity arises solely from the reward distribution, we approximate it using a Gaussian distribution. This leads to efficient posterior approximations that are exact in cases where the reward function is indeed Gaussian (a special case of the GLM model). Precisely, the reward distribution $P(\cdot \mid x; \theta)$ is an exponential-family distribution. Therefore, the log-likelihoods write $\log \mathbb{P}(H_{t,i} \mid \theta_{*,i} = \theta) = \sum_{k \in S_{t,i}} Y_k X_k^\top \theta - A(X_k^\top \theta) + C(Y_k)$, where $C$ is a real function, and $A$ is a twice continuously differentiable function whose derivative is the mean function, $\dot{A} = g$. Now we let $\hat{B}_{t,i}$ and $\hat{G}_{t,i}$ be the maximum likelihood estimate (MLE) and the Hessian of the negative log-likelihood, respectively, defined as

$$\hat{B}_{t,i} = \arg\max_{\theta \in \mathbb{R}^d} \log \mathbb{P}(H_{t,i} \mid \theta_{*,i} = \theta), \qquad \hat{G}_{t,i} = \sum_{k \in S_{t,i}} \dot{g}(X_k^\top \hat{B}_{t,i}) X_k X_k^\top. \tag{17}$$

where $S_{t,i} = \{\ell \in [t-1] : A_\ell = i\}$ are the rounds where the agent takes action $i$ up to round $t$. Then we approximation the respective likelihood as $\mathbb{P}(H_{t,i} \mid \theta_{*,i} = \theta) \approx \mathcal{N}(\theta; \hat{B}_{t,i}, \hat{G}_{t,i}^{-1})$. This approximation makes all posteriors Gaussian. First, the conditional action-posterior reads $P_{t,i}(\cdot \mid \psi_1) = \mathcal{N}(\cdot; \hat{\mu}_{t,i}, \hat{\Sigma}_{t,i})$,

$$\hat{\Sigma}_{t,i}^{-1} = \Sigma_1^{-1} + \hat{G}_{t,i} \qquad \hat{\mu}_{t,i} = \hat{\Sigma}_{t,i}(\Sigma_1^{-1} W_1 \psi_1 + \hat{G}_{t,i} \hat{B}_{t,i}). \tag{18}$$

For $\ell \in [L]/\{1\}$, the $\ell - 1$-th conditional latent-posterior is $Q_{t,\ell-1}(\cdot \mid \psi_\ell) = \mathcal{N}(\bar{\mu}_{t,\ell-1}, \bar{\Sigma}_{t,\ell-1})$,

$$\bar{\Sigma}_{t,\ell-1}^{-1} = \Sigma_\ell^{-1} + \bar{G}_{t,\ell-1}, \qquad \bar{\mu}_{t,\ell-1} = \bar{\Sigma}_{t,\ell-1}(\Sigma_\ell^{-1} W_\ell \psi_\ell + \bar{B}_{t,\ell-1}), \tag{19}$$

and the $L$-th latent-posterior is $Q_{t,L}(\cdot) = \mathcal{N}(\bar{\mu}_{t,L}, \bar{\Sigma}_{t,L})$,

$$\bar{\Sigma}_{t,L}^{-1} = \Sigma_{L+1}^{-1} + \bar{G}_{t,L}, \qquad \bar{\mu}_{t,L} = \bar{\Sigma}_{t,L} \bar{B}_{t,L}. \tag{20}$$

Finally, $\bar{G}_{t,\ell}$ and $\bar{B}_{t,\ell}$ for $\ell \in [L]$ are computed recursively. The basis of the recursion are

$$\bar{G}_{t,1} = W_1^\top \sum_{i=1}^K (\Sigma_1^{-1} - \Sigma_1^{-1} \hat{\Sigma}_{t,i} \Sigma_1^{-1}) W_1, \qquad \bar{B}_{t,1} = W_1^\top \Sigma_1^{-1} \sum_{i=1}^K \hat{\Sigma}_{t,i} \hat{G}_{t,i} \hat{B}_{t,i}. \tag{21}$$

Then, the recursive step for $\ell \in [L]/\{1\}$ is,

$$\bar{G}_{t,\ell} = W_\ell^\top \left( \Sigma_\ell^{-1} - \Sigma_\ell^{-1} \bar{\Sigma}_{t,\ell-1} \Sigma_\ell^{-1} \right) W_\ell , \qquad \bar{B}_{t,\ell} = W_\ell^\top \Sigma_\ell^{-1} \bar{\Sigma}_{t,\ell-1} \bar{B}_{t,\ell-1} . \quad (22)$$

This concludes the derivation of our posterior approximation. Note that these approximations are exact when the reward distribution follows a linear-Gaussian model, $P(\cdot \mid x; \theta_{*,a}) = \mathcal{N}(\cdot; x^\top \theta_{*,a}, \sigma^2)$.

## B.3 DERIVATION OF ACTION-POSTERIORS FOR LINEAR DIFFUSION MODELS

To simplify derivations, we consider the case where the reward distribution is indeed linear-Gaussian as $P(\cdot \mid X_t; \theta_{*,A_t}) = \mathcal{N}\left( X_t^\top \theta_{*,A_t}, \sigma^2 \right)$, but the same derivations can be applied when the rewards are non-linear. In this case, the likelihood approximation in Eq. (17) becomes exact as we have that $\mathbb{P}(H_{t,i} \mid \theta_{*,i} = \theta) \propto \mathcal{N}(\theta; \hat{B}_{t,i}, \hat{G}_{t,i}^{-1})$, where $\hat{B}_{t,i}$ is the corresponding MLE and $\hat{G}_{t,i} = \sigma^{-2} \sum_{k \in S_{t,i}} X_k X_k^\top$ in this case. Our derivations rely on the fact that the MLE $\hat{B}_{t,i}$ in this linear-Gaussian case satisfies: $\hat{G}_{t,i} \hat{B}_{t,i} = v \sum_{k \in S_{t,i}} X_k Y_k^\top$.

**Proposition B.1.** *Consider the following model, which corresponds to the last two layers in Eq. (16)*

$$\theta_{*,i} \mid \psi_{*,1} \sim \mathcal{N}\left( W_1 \psi_{*,1}, \Sigma_1 \right) ,$$
$$Y_t \mid X_t, \theta_{*,A_t} \sim \mathcal{N}\left( X_t^\top \theta_{*,A_t}, \sigma^2 \right) , \qquad \forall t \in [n] .$$

*Then we have that for any $t \in [n]$ and $i \in [K]$, $P_{t,i}(\theta \mid \psi_1) = \mathbb{P}(\theta_{*,i} = \theta \mid \psi_{*,1} = \psi_1, H_{t,i}) = \mathcal{N}(\theta; \hat{\mu}_{t,i}, \hat{\Sigma}_{t,i})$, where*

$$\hat{\Sigma}_{t,i}^{-1} = \hat{G}_{t,i} + \Sigma_1^{-1} , \qquad \hat{\mu}_{t,i} = \hat{\Sigma}_{t,i} \left( \hat{G}_{t,i} \hat{B}_{t,i} + \Sigma_1^{-1} W_1 \psi_1 \right) .$$

*Proof.* Let $v = \sigma^{-2} , \quad \Lambda_1 = \Sigma_1^{-1}$. Then the action-posterior decomposes as

$$
\begin{aligned}
P_{t,i}(\theta \mid \psi_1) &= \mathbb{P}(\theta_{*,i} = \theta \mid \psi_{*,1} = \psi_1, H_{t,i}) , \\
&\propto \mathbb{P}(H_{t,i} \mid \psi_{*,1} = \psi_1, \theta_{*,i} = \theta) \mathbb{P}(\theta_{*,i} = \theta \mid \psi_{*,1} = \psi_1) , \quad \text{(Bayes rule)} \\
&= \mathbb{P}(H_{t,i} \mid \theta_{*,i} = \theta) \mathbb{P}(\theta_{*,i} = \theta \mid \psi_{*,1} = \psi_1) , \text{(given } \theta_{*,i}, H_{t,i} \text{ is independent of } \psi_{*,1}) \\
&= \prod_{k \in S_{t,i}} \mathcal{N}(Y_k; X_k^\top \theta, \sigma^2) \mathcal{N}(\theta; W_1 \psi_1, \Sigma_1) , \\
&= \exp \left[ -\frac{1}{2} \left( v \sum_{k \in S_{t,i}} (Y_k^2 - 2Y_k X_k^\top \theta + (X_k^\top \theta)^2) + \theta^\top \Lambda_1 \theta - 2\theta^\top \Lambda_1 W_1 \psi_1 \right. \right. \\
&\qquad\qquad \left. \left. + \left( W_1 \psi_1 \right)^\top \Lambda_1 \left( W_1 \psi_1 \right) \right) \right] , \\
&\propto \exp \left[ -\frac{1}{2} \left( \theta^\top (v \sum_{k \in S_{t,i}} X_k X_k^\top + \Lambda_1) \theta - 2\theta^\top \left( v \sum_{k \in S_{t,i}} X_k Y_k + \Lambda_1 W_1 \psi_1 \right) \right) \right] , \\
&\propto \mathcal{N}\left( \theta; \hat{\mu}_{t,i}, \hat{\Lambda}_{t,i}^{-1} \right) ,
\end{aligned}
$$

with $\hat{\Lambda}_{t,i} = v \sum_{k \in S_{t,i}} X_k X_k^\top + \Lambda_1 , \hat{\Lambda}_{t,i} \hat{\mu}_{t,i} = v \sum_{k \in S_{t,i}} X_k Y_k + \Lambda_1 W_1 \psi_1$. Using that, in this linear-Gaussian case, $\hat{G}_{t,i} = v \sum_{k \in S_{t,i}} X_k X_k^\top$ and $\hat{G}_{t,i} \hat{B}_{t,i} = v \sum_{k \in S_{t,i}} X_k Y_k$ concludes the proof. $\square$

The same proof applies when the reward distribution is not linear-Gaussian, with the approximation $\mathbb{P}(H_{t,i} \mid \theta_{*,i} = \theta) \approx \mathcal{N}\left( \theta; \hat{B}_{t,i}, \hat{G}_{t,i}^{-1} \right)$. Using this approximation in the derivations above leads to the same results.

## B.4 DERIVATION OF RECURSIVE LATENT-POSTERIORS FOR LINEAR DIFFUSION MODELS

Again, to simplify derivations, we consider the case where the reward distribution is indeed linear-Gaussian as $P(\cdot \mid X_t; \theta_{*,A_t}) = \mathcal{N}\left( X_t^\top \theta_{*,A_t}, \sigma^2 \right)$, but the same derivations can be applied when the rewards are non-linear.

**Proposition B.2.** *For any $\ell \in [L]/\{1\}$, the $\ell - 1$-th conditional latent-posterior reads $Q_{t,\ell-1}(\cdot \mid \psi_\ell) = \mathcal{N}(\bar{\mu}_{t,\ell-1}, \bar{\Sigma}_{t,\ell-1})$, with*

$$\bar{\Sigma}_{t,\ell-1}^{-1} = \Sigma_\ell^{-1} + \bar{G}_{t,\ell-1}, \qquad \bar{\mu}_{t,\ell-1} = \bar{\Sigma}_{t,\ell-1}\big(\Sigma_\ell^{-1}W_\ell\psi_\ell + \bar{B}_{t,\ell-1}\big), \qquad (23)$$

*and the $L$-th latent-posterior reads $Q_{t,L}(\cdot) = \mathcal{N}(\bar{\mu}_{t,L}, \bar{\Sigma}_{t,L})$, with*

$$\bar{\Sigma}_{t,L}^{-1} = \Sigma_{L+1}^{-1} + \bar{G}_{t,L}, \qquad \bar{\mu}_{t,L} = \bar{\Sigma}_{t,L}\bar{B}_{t,L}. \qquad (24)$$

*Proof.* Let $\ell \in [L]/\{1\}$. Then, Bayes rule yields that

$$Q_{t,\ell-1}(\psi_{\ell-1} \mid \psi_\ell) \propto \mathbb{P}\left(H_t \mid \psi_{*,\ell-1} = \psi_{\ell-1}\right)\mathcal{N}(\psi_{\ell-1}, W_\ell\psi_\ell, \Sigma_\ell),$$

But from [Lemma B.3](#), we know that

$$\mathbb{P}\left(H_t \mid \psi_{*,\ell-1} = \psi_{\ell-1}\right) \propto \exp\left[-\frac{1}{2}\psi_{\ell-1}^\top \bar{G}_{t,\ell-1}\psi_{\ell-1} + \psi_{\ell-1}^\top \bar{B}_{t,\ell-1}\right].$$

Therefore,

$$
\begin{aligned}
Q_{t,\ell-1}(\psi_{\ell-1} \mid \psi_\ell) &\propto \exp\left[-\frac{1}{2}\psi_{\ell-1}^\top \bar{G}_{t,\ell-1}\psi_{\ell-1} + \psi_{\ell-1}^\top \bar{B}_{t,\ell-1}\right]\mathcal{N}(\psi_{\ell-1}, W_\ell\psi_\ell, \Sigma_\ell), \\
&\propto \exp\Big[-\frac{1}{2}\psi_{\ell-1}^\top \bar{G}_{t,\ell-1}\psi_{\ell-1} + \psi_{\ell-1}^\top \bar{B}_{t,\ell-1} \\
&\qquad\qquad -\frac{1}{2}(\psi_{\ell-1} - W_\ell\psi_\ell)^\top \Sigma_\ell^{-1}(\psi_{\ell-1} - W_\ell\psi_\ell))\Big], \\
&\stackrel{(i)}{\propto} \exp\left[-\frac{1}{2}\psi_{\ell-1}^\top(\bar{G}_{t,\ell-1} + \Sigma_\ell^{-1})\psi_{\ell-1} + \psi_{\ell-1}^\top(\bar{B}_{t,\ell-1} + \Sigma_\ell^{-1}W_\ell\psi_\ell)\right], \\
&\stackrel{(ii)}{\propto} \mathcal{N}(\psi_{\ell-1}; \bar{\mu}_{t,\ell-1}, \bar{\Sigma}_{t,\ell-1}),
\end{aligned}
$$

with $\bar{\Sigma}_{t,\ell-1}^{-1} = \Sigma_\ell^{-1} + \bar{G}_{t,\ell-1}$ and $\bar{\mu}_{t,\ell-1} = \bar{\Sigma}_{t,\ell-1}\big(\Sigma_\ell^{-1}W_\ell\psi_\ell + \bar{B}_{t,\ell-1}\big)$. In $(i)$, we omit terms that are constant in $\psi_{\ell-1}$. In $(ii)$, we complete the square. This concludes the proof for $\ell \in [L]/\{1\}$. For $Q_{t,L}$, we use Bayes rule to get

$$Q_{t,L}(\psi_L) \propto \mathbb{P}\left(H_t \mid \psi_{*,L} = \psi_L\right)\mathcal{N}(\psi_L, 0, \Sigma_{L+1}).$$

Then from [Lemma B.3](#), we know that

$$\mathbb{P}\left(H_t \mid \psi_{*,L} = \psi_L\right) \propto \exp\left[-\frac{1}{2}\psi_L^\top \bar{G}_{t,L}\psi_L + \psi_L^\top \bar{B}_{t,L}\right],$$

We then use the same derivations above to compute the product $\exp\left[-\frac{1}{2}\psi_L^\top \bar{G}_{t,L}\psi_L + \psi_L^\top \bar{B}_{t,L}\right] \times \mathcal{N}(\psi_L, 0, \Sigma_{L+1})$, which concludes the proof. $\qquad\square$

**Lemma B.3.** *The following holds for any $t \in [n]$ and $\ell \in [L]$,*

$$\mathbb{P}\left(H_t \mid \psi_{*,\ell} = \psi_\ell\right) \propto \exp\left[-\frac{1}{2}\psi_\ell^\top \bar{G}_{t,\ell}\psi_\ell + \psi_\ell^\top \bar{B}_{t,\ell}\right],$$

*where $\bar{G}_{t,\ell}$ and $\bar{B}_{t,\ell}$ are defined by recursion in [Appendix B.2](#).*

*Proof.* We prove this result by induction. To reduce clutter, we let $v = \sigma^{-2}$, and $\Lambda_1 = \Sigma_1^{-1}$. We start with the base case of the induction when $\ell = 1$.

**(I) Base case.** Here we want to show that $\mathbb{P}\left(H_t \,|\, \psi_{*,1} = \psi_1\right) \propto \exp\left[-\frac{1}{2}\psi_1^\top \bar{G}_{t,1}\psi_1 + \psi_1^\top \bar{B}_{t,1}\right)\right]$, where $\bar{G}_{t,1}$ and $\bar{B}_{t,1}$ are given in Eq. (21). First, we have that

$$\mathbb{P}\left(H_t \,|\, \psi_{*,1} = \psi_1\right) \overset{(i)}{=} \prod_{i\in[K]} \mathbb{P}\left(H_{t,i} \,|\, \psi_{*,1} = \psi_1\right) = \prod_{i\in[K]} \int_\theta \mathbb{P}\left(H_{t,i}, \theta_{*,i} = \theta \,|\, \psi_{*,1} = \psi_1\right) \mathrm{d}\theta,$$

$$= \prod_{i\in[K]} \int_\theta \mathbb{P}\left(H_{t,i} \,|\, \theta_{*,i} = \theta\right) \mathcal{N}\left(\theta; \mathrm{W}_1\psi_1, \Sigma_1\right) \mathrm{d}\theta,$$

$$= \prod_{i\in[K]} \underbrace{\int_\theta \Big(\prod_{k\in S_{t,i}} \mathcal{N}(Y_k; X_k^\top\theta, \sigma^2)\Big) \mathcal{N}\left(\theta; \mathrm{W}_1\psi_1, \Sigma_1\right) \mathrm{d}\theta}_{h_i(\psi_1)},$$

$$= \prod_{i\in[K]} h_i(\psi_1), \tag{25}$$

where $(i)$ follows from the fact that $\theta_{*,i}$ for $i \in [K]$ are conditionally independent given $\psi_{*,1} = \psi_1$ and that given $\theta_{*,i}$, $H_{t,i}$ is independent of $\psi_{*,1}$. Now we compute $h_i(\psi_1) = \int_\theta \left(\prod_{k\in S_{t,i}} \mathcal{N}(Y_k; X_k^\top\theta, \sigma^2)\right) \mathcal{N}\left(\theta; \mathrm{W}_1\psi_1, \Sigma_1\right) \mathrm{d}\theta$ as

$$h_i(\psi_1) = \int_\theta \Big(\prod_{k\in S_{t,i}} \mathcal{N}(Y_k; X_k^\top\theta, \sigma^2)\Big) \mathcal{N}\left(\theta; \mathrm{W}_1\psi_1, \Sigma_1\right) \mathrm{d}\theta,$$

$$\propto \int_\theta \exp\Big[-\frac{1}{2}v\sum_{k\in S_{t,i}}(Y_k - X_k^\top\theta)^2 - \frac{1}{2}(\theta - \mathrm{W}_1\psi_1)^\top\Lambda_1(\theta - \mathrm{W}_1\psi_1)\Big]\mathrm{d}\theta,$$

$$= \int_\theta \exp\Big[-\frac{1}{2}\Big(v\sum_{k\in S_{t,i}}(Y_k^2 - 2Y_k\theta^\top X_k + (\theta^\top X_k)^2) + \theta^\top\Lambda_1\theta - 2\theta^\top\Lambda_1\mathrm{W}_1\psi_1$$

$$+ (\mathrm{W}_1\psi_1)^\top\Lambda_1(\mathrm{W}_1\psi_1)\Big)\Big]\mathrm{d}\theta,$$

$$\propto \int_\theta \exp\Big[-\frac{1}{2}\Big(\theta^\top\Big(v\sum_{k\in S_{t,i}}X_kX_k^\top + \Lambda_1\Big)\theta - 2\theta^\top\Big(v\sum_{k\in S_{t,i}}Y_kX_k$$

$$+ \Lambda_1\mathrm{W}_1\psi_1\Big) + (\mathrm{W}_1\psi_1)^\top\Lambda_1(\mathrm{W}_1\psi_1)\Big)\Big]\mathrm{d}\theta.$$

But we know that $\hat{G}_{t,i} = v\sum_{k\in S_{t,i}}X_kX_k^\top$, and $\hat{G}_{t,i}\hat{B}_{t,i} = v\sum_{k\in S_{t,i}}Y_kX_k$ (because we assumed linear-Gaussian likelihood). To further simplify expressions, we also let

$$V = \left(\hat{G}_{t,i} + \Lambda_1\right)^{-1}, \quad U = V^{-1}, \quad \beta = V\left(\hat{G}_{t,i}\hat{B}_{t,i} + \Lambda_1\mathrm{W}_1\psi_1\right).$$

We have that $UV = VU = I_d$, and thus

$$h_i(\psi_1) \propto \int_\theta \exp\left[-\frac{1}{2}\left(\theta^\top U\theta - 2\theta^\top UV\left(\hat{G}_{t,i}\hat{B}_{t,i} + \Lambda_1\mathrm{W}_1\psi_1\right) + (\mathrm{W}_1\psi_1)^\top\Lambda_1(\mathrm{W}_1\psi_1)\right)\right]\mathrm{d}\theta,$$

$$= \int_\theta \exp\left[-\frac{1}{2}\left(\theta^\top U\theta - 2\theta^\top U\beta + (\mathrm{W}_1\psi_1)^\top\Lambda_1(\mathrm{W}_1\psi_1)\right)\right]\mathrm{d}\theta,$$

$$= \int_\theta \exp\left[-\frac{1}{2}\left((\theta - \beta)^\top U(\theta - \beta) - \beta^\top U\beta + (\mathrm{W}_1\psi_1)^\top\Lambda_1(\mathrm{W}_1\psi_1)\right)\right]\mathrm{d}\theta,$$

$$\propto \exp\left[-\frac{1}{2}\left(-\beta^\top U\beta + (\mathrm{W}_1\psi_1)^\top\Lambda_1(\mathrm{W}_1\psi_1)\right)\right],$$

$$= \exp\left[-\frac{1}{2}\left(-\left(\hat{G}_{t,i}\hat{B}_{t,i} + \Lambda_1\mathrm{W}_1\psi_1\right)^\top V\left(\hat{G}_{t,i}\hat{B}_{t,i} + \Lambda_1\mathrm{W}_1\psi_1\right) + (\mathrm{W}_1\psi_1)^\top\Lambda_1(\mathrm{W}_1\psi_1)\right)\right],$$

$$\propto \exp\left[-\frac{1}{2}\left(\psi_1^\top\mathrm{W}_1^\top\left(\Lambda_1 - \Lambda_1 V\Lambda_1\right)\mathrm{W}_1\psi_1 - 2\psi_1^\top\left(\mathrm{W}_1^\top\Lambda_1 V\hat{G}_{t,i}\hat{B}_{t,i}\right)\right)\right],$$

$$= \exp\left[-\frac{1}{2}\psi_1^\top\Omega_i\psi_1 + \psi_1^\top m_i\right],$$

where

$$\Omega_i = W_1^\top \left(\Lambda_1 - \Lambda_1 V \Lambda_1\right) W_1 = W_1^\top \left(\Lambda_1 - \Lambda_1 (\hat{G}_{t,i} + \Lambda_1)^{-1} \Lambda_1\right) W_1 \,,$$

$$m_i = W_1^\top \Lambda_1 V \hat{G}_{t,i} \hat{B}_{t,i} = W_1^\top \Lambda_1 (\hat{G}_{t,i} + \Lambda_1)^{-1} \hat{G}_{t,i} \hat{B}_{t,i} \,. \tag{26}$$

But notice that $V = (\hat{G}_{t,i} + \Lambda_1)^{-1} = \hat{\Sigma}_{t,i}$ and thus

$$\Omega_i = W_1^\top \left(\Lambda_1 - \Lambda_1 \hat{\Sigma}_{t,i} \Lambda_1\right) W_1 \,, \qquad\qquad m_i = W_1^\top \Lambda_1 \hat{\Sigma}_{t,i} \hat{G}_{t,i} \hat{B}_{t,i} \,. \tag{27}$$

Finally, we plug this result in Eq. (25) to get

$$\mathbb{P}\left(H_t \mid \psi_{*,1} = \psi_1\right) = \prod_{i \in [K]} h_i(\psi_1) \propto \prod_{i \in [K]} \exp\left[-\frac{1}{2}\psi_1^\top \Omega_i \psi_1 + \psi_1^\top m_i\right] \,,$$

$$= \exp\left[-\frac{1}{2}\psi_1^\top \sum_{i \in [K]} \Omega_i \psi_1 + \psi_1^\top \sum_{i \in [K]} m_i\right] \,,$$

$$= \exp\left[-\frac{1}{2}\psi_1^\top \bar{G}_{t,1} \psi_1 + \psi_1^\top \bar{B}_{t,1}\right] \,,$$

where

$$\bar{G}_{t,1} = \sum_{i=1}^K \Omega_i = \sum_{i=1}^K W_1^\top \left(\Lambda_1 - \Lambda_1 \hat{\Sigma}_{t,i} \Lambda_1\right) W_1 = W_1^\top \sum_{i=1}^K \left(\Sigma_1^{-1} - \Sigma_1^{-1} \hat{\Sigma}_{t,i} \Sigma_1^{-1}\right) W_1 \,,$$

$$\bar{B}_{t,1} = \sum_{i=1}^K m_i = \sum_{i=1}^K \hat{\Sigma}_{t,i} \hat{G}_{t,i} \hat{B}_{t,i} = W_1^\top \Sigma_1^{-1} \sum_{i=1}^K \hat{\Sigma}_{t,i} \hat{G}_{t,i} \hat{B}_{t,i} \,.$$

This concludes the proof of the base case.

**(II) Induction step.** Let $\ell \in [L]/\{1\}$. Suppose that

$$\mathbb{P}\left(H_t \mid \psi_{*,\ell-1} = \psi_{\ell-1}\right) \propto \exp\left[-\frac{1}{2}\psi_{\ell-1}^\top \bar{G}_{t,\ell-1} \psi_{\ell-1} + \psi_{\ell-1}^\top \bar{B}_{t,\ell-1}\right] \,. \tag{28}$$

Then we want to show that

$$\mathbb{P}\left(H_t \mid \psi_{*,\ell} = \psi_\ell\right) \propto \exp\left[-\frac{1}{2}\psi_\ell^\top \bar{G}_{t,\ell} \psi_\ell + \psi_\ell^\top \bar{B}_{t,\ell}\right] \,,$$

where

$$\bar{G}_{t,\ell} = W_\ell^\top \left(\Sigma_\ell^{-1} - \Sigma_\ell^{-1} \bar{\Sigma}_{t,\ell-1} \Sigma_\ell^{-1}\right) W_\ell = W_\ell^\top \left(\Sigma_\ell^{-1} - \Sigma_\ell^{-1}(\Sigma_\ell^{-1} + \bar{G}_{t,\ell-1})^{-1} \Sigma_\ell^{-1}\right) W_\ell \,,$$

$$\bar{B}_{t,\ell} = W_\ell^\top \Sigma_\ell^{-1} \bar{\Sigma}_{t,\ell-1} \bar{B}_{t,\ell-1} = W_\ell^\top \Sigma_\ell^{-1}(\Sigma_\ell^{-1} + \bar{G}_{t,\ell-1})^{-1} \bar{B}_{t,\ell-1} \,.$$

To achieve this, we start by expressing $\mathbb{P}\left(H_t \mid \psi_{*,\ell} = \psi_\ell\right)$ in terms of $\mathbb{P}\left(H_t \mid \psi_{*,\ell-1} = \psi_{\ell-1}\right)$ as

$$\mathbb{P}\left(H_t \mid \psi_{*,\ell} = \psi_\ell\right) = \int_{\psi_{\ell-1}} \mathbb{P}\left(H_t, \psi_{*,\ell-1} = \psi_{\ell-1} \mid \psi_{*,\ell} = \psi_\ell\right) \mathrm{d}\psi_{\ell-1} \,,$$

$$= \int_{\psi_{\ell-1}} \mathbb{P}\left(H_t \mid \psi_{*,\ell-1} = \psi_{\ell-1}, \psi_{*,\ell} = \psi_\ell\right) \mathcal{N}(\psi_{\ell-1}; W_\ell \psi_\ell, \Sigma_\ell) \, \mathrm{d}\psi_{\ell-1} \,,$$

$$= \int_{\psi_{\ell-1}} \mathbb{P}\left(H_t \mid \psi_{*,\ell-1} = \psi_{\ell-1}\right) \mathcal{N}(\psi_{\ell-1}; W_\ell \psi_\ell, \Sigma_\ell) \, \mathrm{d}\psi_{\ell-1} \,,$$

$$\propto \int_{\psi_{\ell-1}} \exp\left[-\frac{1}{2}\psi_{\ell-1}^\top \bar{G}_{t,\ell-1} \psi_{\ell-1} + \psi_{\ell-1}^\top \bar{B}_{t,\ell-1}\right] \mathcal{N}(\psi_{\ell-1}; W_\ell \psi_\ell, \Sigma_\ell) \, \mathrm{d}\psi_{\ell-1} \,,$$

$$\propto \int_{\psi_{\ell-1}} \exp\left[-\frac{1}{2}\psi_{\ell-1}^\top \bar{G}_{t,\ell-1} \psi_{\ell-1} + \psi_{\ell-1}^\top \bar{B}_{t,\ell-1}\right.$$
$$\left. + (\psi_{\ell-1} - W_\ell \psi_\ell)^\top \Lambda_\ell (\psi_{\ell-1} - W_\ell \psi_\ell)\right] \mathrm{d}\psi_{\ell-1} \,.$$

Now let $S = \bar{G}_{t,\ell-1} + \Lambda_\ell$ and $V = \bar{B}_{t,\ell-1} + \Lambda_\ell \mathrm{W}_\ell \psi_\ell$. Then we have that,

$$
\mathbb{P}\left(H_t \mid \psi_{*,\ell} = \psi_\ell\right)
$$

$$
\propto \int_{\psi_{\ell-1}} \exp\Big[ -\frac{1}{2}\psi_{\ell-1}^\top \bar{G}_{t,\ell-1}\psi_{\ell-1} + \psi_{\ell-1}^\top \bar{B}_{t,\ell-1}
$$

$$
+ (\psi_{\ell-1} - \mathrm{W}_\ell\psi_\ell)^\top \Lambda_\ell(\psi_{\ell-1} - \mathrm{W}_\ell\psi_\ell)\Big] \, \mathrm{d}\psi_{\ell-1} \,,
$$

$$
\propto \int_{\psi_{\ell-1}} \exp\Big[ -\frac{1}{2}\Big( \psi_{\ell-1}^\top S\psi_{\ell-1} - 2\psi_{\ell-1}^\top \left(\bar{B}_{t,\ell-1} + \Lambda_\ell \mathrm{W}_\ell\psi_\ell\right) + \psi_\ell^\top \mathrm{W}_\ell^\top \Lambda_\ell \mathrm{W}_\ell\psi_\ell \Big)\Big] \, \mathrm{d}\psi_{\ell-1} \,,
$$

$$
= \int_{\psi_{\ell-1}} \exp\Big[ -\frac{1}{2}\Big( \psi_{\ell-1}^\top S(\psi_{\ell-1} - 2S^{-1}V) + \psi_\ell^\top \mathrm{W}_\ell^\top \Lambda_\ell \mathrm{W}_\ell\psi_\ell \Big)\Big] \, \mathrm{d}\psi_{\ell-1} \,,
$$

$$
= \int_{\psi_{\ell-1}} \exp\Big[ -\frac{1}{2}\Big( (\psi_{\ell-1} - S^{-1}V)^\top S(\psi_{\ell-1} - S^{-1}V)
$$

$$
+ \psi_\ell^\top \mathrm{W}_\ell^\top \Lambda_\ell \mathrm{W}_\ell\psi_\ell - V^\top S^{-1}V \Big)\Big] \, \mathrm{d}\psi_{\ell-1}.
$$

In the second step, we omit constants in $\psi_\ell$ and $\psi_{\ell-1}$. Thus

$$
\mathbb{P}\left(H_t \mid \psi_{*,\ell} = \psi_\ell\right)
$$

$$
\propto \int_{\psi_{\ell-1}} \exp\left[ -\frac{1}{2}\left( (\psi_{\ell-1} - S^{-1}V)^\top S(\psi_{\ell-1} - S^{-1}V) + \psi_\ell^\top \mathrm{W}_\ell^\top \Lambda_\ell \mathrm{W}_\ell\psi_\ell - V^\top S^{-1}V \right)\right] \, \mathrm{d}\psi_{\ell-1},
$$

$$
\propto \exp\left[ -\frac{1}{2}\left( \psi_\ell^\top \mathrm{W}_\ell^\top \Lambda_\ell \mathrm{W}_\ell\psi_\ell - V^\top S^{-1}V \right)\right] \,.
$$

It follows that

$$
\mathbb{P}\left(H_t \mid \psi_{*,\ell} = \psi_\ell\right)
$$

$$
\propto \exp\left[ -\frac{1}{2}\left( \psi_\ell^\top \mathrm{W}_\ell^\top \Lambda_\ell \mathrm{W}_\ell\psi_\ell - V^\top S^{-1}V \right)\right] \,,
$$

$$
= \exp\left[ -\frac{1}{2}\left( \psi_\ell^\top \mathrm{W}_\ell^\top \Lambda_\ell \mathrm{W}_\ell\psi_\ell - \left(\bar{B}_{t,\ell-1} + \Lambda_\ell \mathrm{W}_\ell\psi_\ell\right)^\top S^{-1} \left(\bar{B}_{t,\ell-1} + \Lambda_\ell \mathrm{W}_\ell\psi_\ell\right) \right)\right]
$$

$$
\propto \exp\left[ -\frac{1}{2}\left( \psi_\ell^\top \left( \mathrm{W}_\ell^\top \Lambda_\ell \mathrm{W}_\ell - \mathrm{W}_\ell^\top \Lambda_\ell S^{-1} \Lambda_\ell \mathrm{W}_\ell \right)\psi_\ell - 2\psi_\ell^\top \mathrm{W}_\ell^\top \Lambda_\ell S^{-1}\bar{B}_{t,\ell-1}\right)\right] \,,
$$

$$
= \exp\left[ -\frac{1}{2}\psi_\ell^\top \bar{G}_{t,\ell}\psi_\ell + \psi_\ell^\top \bar{B}_{t,\ell}\right] \,.
$$

In the last step, we omit constants in $\psi_\ell$ and we set

$$
\bar{G}_{t,\ell} = \mathrm{W}_\ell^\top \left( \Lambda_\ell - \Lambda_\ell S^{-1}\Lambda_\ell \right) \mathrm{W}_\ell = \mathrm{W}_\ell^\top \left( \Lambda_\ell - \Lambda_\ell(\Lambda_\ell + \bar{G}_{t,\ell-1})^{-1}\Sigma_\ell^{-1}\Lambda_\ell \right) \mathrm{W}_\ell \,,
$$

$$
\bar{B}_{t,\ell} = \mathrm{W}_\ell^\top \Lambda_\ell S^{-1}\bar{B}_{t,\ell-1} = \mathrm{W}_\ell^\top \Lambda_\ell(\Lambda_\ell + \bar{G}_{t,\ell-1})^{-1}\bar{B}_{t,\ell-1} \,.
$$

This completes the proof. $\qquad\square$

Similarly, this same proof applies when the reward distribution is not linear-Gaussian, with the approximation $\mathbb{P}\left(H_{t,i} \mid \theta_{*,i} = \theta\right) \approx \mathcal{N}\left(\theta; \hat{B}_{t,i}, \hat{G}_{t,i}^{-1}\right)$. Using this approximation in the derivations above leads to the same results.

## C  POSTERIOR DERIVATIONS FOR NON-LINEAR DIFFUSION MODELS

After deriving the exact posteriors in the case where the link functions $f_\ell$ are linear (Appendix B.2), we now get back to the general case with any link functions $f_\ell$ that can be non-linear. Approximation is needed since both the link functions and rewards can be non-linear. To avoid any computational challenges, we use a simple and intuitive approximation, where all posteriors $P_{t,i}$ and $Q_{t,\ell}$ are approximated by the Gaussian distributions in Appendix B.2, with few changes. First, the terms $\mathrm{W}_\ell\psi_\ell$ in Eq. (19) are replaced by $f_\ell(\psi_\ell)$. This accounts for the fact that the prior mean is now $f_\ell(\psi_\ell)$

rather than $W_\ell \psi_\ell$, and this is the main difference between the linear diffusion model in Eq. (16) and the general, potentially non-linear, diffusion model in Eq. (1). Second, the matrix multiplications that involve the matrices $W_\ell$ in Eq. (21) and Eq. (22) are simply removed. Despite being simple, this approximation is efficient and avoids the computational burden of heavy approximate sampling algorithms required for each latent parameter. This is why deriving the exact posterior for linear link functions was key beyond enabling theoretical analyses. Moreover, this approximation retains some key attributes of exact posteriors. Specifically, in the absence of data, it recovers precisely the prior in Eq. (1), and as more data is accumulated, the influence of the prior diminishes.

# D  REGRET PROOF AND ADDITIONAL DISCUSSIONS

## D.1  SKETCH OF THE PROOF

We start with the following standard lemma upon which we build our analysis (Aouali et al., 2023b).

**Lemma D.1.** *Assume that* $\mathbb{P}(\theta_{*,i} = \theta \mid H_t) = \mathcal{N}(\theta; \check{\mu}_{t,i}, \check{\Sigma}_{t,i})$ *for any* $i \in [K]$, *then for any* $\delta \in (0,1)$,

$$\mathcal{BR}(n) \leq \sqrt{2n \log(1/\delta)} \sqrt{\mathbb{E}\left[\sum_{t=1}^n \|X_t\|^2_{\check{\Sigma}_{t,A_t}}\right]} + cn\delta, \qquad \text{where } c > 0 \text{ is a constant}. \quad (29)$$

Applying Lemma D.1 requires proving that the *marginal* action-posteriors $\mathbb{P}(\theta_{*,i} = \theta \mid H_t)$ in Eq. (3) are Gaussian and computing their covariances, while we only know the *conditional* action-posteriors $P_{t,i}$ and latent-posteriors $Q_{t,\ell}$. This is achieved by leveraging the preservation properties of the family of Gaussian distributions (Koller & Friedman, 2009) and the total covariance decomposition (Weiss, 2005) which leads to the next lemma.

**Lemma D.2.** *Let* $t \in [n]$ *and* $i \in [K]$, *then the marginal covariance matrix* $\check{\Sigma}_{t,i}$ *reads*

$$\check{\Sigma}_{t,i} = \hat{\Sigma}_{t,i} + \sum_{\ell \in [L]} P_{i,\ell} \bar{\Sigma}_{t,\ell} P_{i,\ell}^\top, \quad \text{where } P_{i,\ell} = \hat{\Sigma}_{t,i} \Sigma_1^{-1} W_1 \prod_{k=1}^{\ell-1} \bar{\Sigma}_{t,k} \Sigma_{k+1}^{-1} W_{k+1}. \quad (30)$$

The marginal covariance matrix $\check{\Sigma}_{t,i}$ in Eq. (30) decomposes into $L+1$ terms. The first term corresponds to the posterior uncertainty of $\theta_{*,i} \mid \psi_{*,1}$. The remaining $L$ terms capture the posterior uncertainties of $\psi_{*,L}$ and $\psi_{*,\ell-1} \mid \psi_{*,\ell}$ for $\ell \in [L]/\{1\}$. These are then used to quantify the posterior information gain of latent parameters after one round as follows.

**Lemma D.3** (Posterior information gain). *Let* $t \in [n]$ *and* $\ell \in [L]$, *then*

$$\bar{\Sigma}_{t+1,\ell}^{-1} - \bar{\Sigma}_{t,\ell}^{-1} \succeq \sigma^{-2} \sigma_{\text{MAX}}^{-2\ell} P_{A_t,\ell}^\top X_t X_t^\top P_{A_t,\ell}, \quad \text{where } \sigma_{\text{MAX}}^2 = \max_{\ell \in [L+1]} 1 + \frac{\sigma_\ell^2}{\sigma^2}. \quad (31)$$

Finally, Lemma D.2 is used to decompose $\|X_t\|^2_{\check{\Sigma}_{t,A_t}}$ in Eq. (29) into $L+1$ terms. Each term is bounded thanks to Lemma D.3. This results in the Bayes regret bound in Theorem 4.1.

## D.2  TECHNICAL CONTRIBUTIONS

Our main technical contributions are the following.

**Lemma D.2.** In `dTS`, sampling is done hierarchically, meaning the marginal posterior distribution of $\theta_{*,i} \mid H_t$ is not explicitly defined. Instead, we use the conditional posterior distribution of $\theta_{*,i} \mid H_t, \psi_{*,1}$. The first contribution was deriving $\theta_{*,i} \mid H_t$ using the total covariance decomposition combined with an induction proof, as our posteriors in Appendix B.2 were derived recursively. Unlike in Bayes regret analysis for standard Thompson sampling, where the posterior distribution of $\theta_{*,i} \mid H_t$ is predetermined due to the absence of latent parameters, our method necessitates this recursive total covariance decomposition, marking a first difference from the standard Bayesian proofs of Thompson sampling. Note that `HierTS`, which is developed for multi-task linear bandits, also employs total covariance decomposition, but it does so under the assumption of a single latent parameter; on which action parameters are centered. Our extension significantly differs as it is tailored for contextual bandits with multiple, successive levels of latent parameters, moving away from `HierTS`'s assumption of a 1-level structure. Roughly speaking, `HierTS` when applied to contextual would consider a single-level hierarchy, where $\theta_{*,i} \mid \psi_{*,1} \sim \mathcal{N}(\psi_{*,1}, \Sigma_1)$ with $L = 1$. In contrast, our model proposes a

multi-level hierarchy, where the first level is $\theta_{*,i}|\psi_{*,1} \sim \mathcal{N}(W_1\psi_{*,1}, \Sigma_1)$. This also introduces a new aspect to our approach – the use of a linear function $W_1\psi_{*,1}$, as opposed to `HierTS`'s assumption where action parameters are centered directly on the latent parameter. Thus, while `HierTS` also uses the total covariance decomposition, our generalize it to multi-level hierarchies under $L$ linear functions $W_\ell\psi_{*,\ell}$, instead of a single-level hierarchy under a single identity function $\psi_{*,1}$.

**Lemma D.3.** In Bayes regret proofs for standard Thompson sampling, we often quantify the posterior information gain. This is achieved by monitoring the increase in posterior precision for the action taken $A_t$ in each round $t \in [n]$. However, in `dTS`, our analysis extends beyond this. We not only quantify the posterior information gain for the taken action but also for every latent parameter, since they are also learned. This lemma addresses this aspect. To elaborate, we use the recursive formulas in Appendix B.2 that connect the posterior covariance of each latent parameter $\psi_{*,\ell}$ with the covariance of the posterior action parameters $\theta_{*,i}$. This allows us to propagate the information gain associated with the action taken in round $A_t$ to all latent parameters $\psi_{*,\ell}$, for $\ell \in [L]$ by induction. This is a novel contribution, as it is not a feature of Bayes regret analyses in standard Thompson sampling.

**Proposition 4.2.** Building upon the insights of Theorem 4.1, we introduce the sparsity assumption (**A3**). Under this assumption, we demonstrate that the Bayes regret outlined in Theorem 4.1 can be significantly refined. Specifically, the regret becomes contingent on dimensions $d_\ell \leq d$, as opposed to relying on the entire dimension $d$. The underlying principle of this sparsity assumption is straightforward: the Bayes regret is influenced by the quantity of parameters that require learning. With the sparsity assumption, this number is reduced to less than $d$ for each latent parameter. To substantiate this claim, we revisit the proof of Theorem 4.1 and modify a crucial equality. This adjustment results in a more precise representation by partitioning the covariance matrix of each latent parameter $\psi_{*,\ell}$ into blocks. These blocks comprise a $d_\ell \times d_\ell$ segment corresponding to the learnable $d_\ell$ parameters of $\psi_{*,\ell}$, and another block of size $(d-d_\ell) \times (d-d_\ell)$ that does not necessitate learning. This decomposition allows us to conclude that the final regret is solely dependent on $d_\ell$, marking a significant refinement from the original theorem.

## D.3 PROOF OF LEMMA D.2

In this proof, we heavily rely on the total covariance decomposition (Weiss, 2005). Also, refer to (Hong et al., 2022b, Section 5.2) for a brief introduction to this decomposition. Now, from Eq. (18), we have that

$$\text{cov}\left[\theta_{*,i} \mid H_t, \psi_{*,1}\right] = \hat{\Sigma}_{t,i} = \left(\hat{G}_{t,i} + \Sigma_1^{-1}\right)^{-1},$$

$$\mathbb{E}\left[\theta_{*,i} \mid H_t, \psi_{*,1}\right] = \hat{\mu}_{t,i} = \hat{\Sigma}_{t,i}\left(\hat{G}_{t,i}\hat{B}_{t,i} + \Sigma_1^{-1}W_1\psi_{*,1}\right).$$

First, given $H_t$, $\text{cov}\left[\theta_{*,i} \mid H_t, \psi_{*,1}\right] = \left(\hat{G}_{t,i} + \Sigma_1^{-1}\right)^{-1}$ is constant. Thus

$$\mathbb{E}\left[\text{cov}\left[\theta_{*,i} \mid H_t, \psi_{*,1}\right] \mid H_t\right] = \text{cov}\left[\theta_{*,i} \mid H_t, \psi_{*,1}\right] = \left(\hat{G}_{t,i} + \Sigma_1^{-1}\right)^{-1} = \hat{\Sigma}_{t,i}.$$

In addition, given $H_t$, $\hat{\Sigma}_{t,i}$, $\hat{G}_{t,i}$ and $\hat{B}_{t,i}$ are constant. Thus

$$\begin{aligned}
\text{cov}\left[\mathbb{E}\left[\theta_{*,i} \mid H_t, \psi_{*,1}\right] \mid H_t\right] &= \text{cov}\left[\hat{\Sigma}_{t,i}\left(\hat{G}_{t,i}\hat{B}_{t,i} + \Sigma_1^{-1}W_1\psi_{*,1}\right) \mid H_t\right], \\
&= \text{cov}\left[\hat{\Sigma}_{t,i}\Sigma_1^{-1}W_1\psi_{*,1} \mid H_t\right], \\
&= \hat{\Sigma}_{t,i}\Sigma_1^{-1}W_1\text{cov}\left[\psi_{*,1} \mid H_t\right]W_1^\top\Sigma_1^{-1}\hat{\Sigma}_{t,i}, \\
&= \hat{\Sigma}_{t,i}\Sigma_1^{-1}W_1\bar{\bar{\Sigma}}_{t,1}W_1^\top\Sigma_1^{-1}\hat{\Sigma}_{t,i},
\end{aligned}$$

where $\bar{\bar{\Sigma}}_{t,1} = \text{cov}\left[\psi_{*,1} \mid H_t\right]$ is the marginal posterior covariance of $\psi_{*,1}$. Finally, the total covariance decomposition (Weiss, 2005; Hong et al., 2022b) yields that

$$\begin{aligned}
\check{\Sigma}_{t,i} = \text{cov}\left[\theta_{*,i} \mid H_t\right] &= \mathbb{E}\left[\text{cov}\left[\theta_{*,i} \mid H_t, \psi_{*,1}\right] \mid H_t\right] + \text{cov}\left[\mathbb{E}\left[\theta_{*,i} \mid H_t, \psi_{*,1}\right] \mid H_t\right], \\
&= \hat{\Sigma}_{t,i} + \hat{\Sigma}_{t,i}\Sigma_1^{-1}W_1\bar{\bar{\Sigma}}_{t,1}W_1^\top\Sigma_1^{-1}\hat{\Sigma}_{t,i},
\end{aligned} \tag{32}$$

However, $\bar{\bar{\Sigma}}_{t,1} = \mathrm{cov}\left[\psi_{*,1} \mid H_t\right]$ is different from $\bar{\Sigma}_{t,1} = \mathrm{cov}\left[\psi_{*,1} \mid H_t, \psi_{*,2}\right]$ that we already derived in Eq. (19). Thus we do not know the expression of $\bar{\bar{\Sigma}}_{t,1}$. But we can use the same total covariance decomposition trick to find it. Precisely, let $\bar{\bar{\Sigma}}_{t,\ell} = \mathrm{cov}\left[\psi_{*,\ell} \mid H_t\right]$ for any $\ell \in [L]$. Then we have that

$$\bar{\Sigma}_{t,1} = \mathrm{cov}\left[\psi_{*,1} \mid H_t, \psi_{*,2}\right] = \left(\Sigma_2^{-1} + \bar{G}_{t,1}\right)^{-1},$$
$$\bar{\mu}_{t,1} = \mathbb{E}\left[\psi_{*,1} \mid H_t, \psi_{*,2}\right] = \bar{\Sigma}_{t,1}\left(\Sigma_2^{-1}\mathrm{W}_2\psi_{*,2} + \bar{B}_{t,1}\right).$$

First, given $H_t$, $\mathrm{cov}\left[\psi_{*,1} \mid H_t, \psi_{*,2}\right] = \left(\Sigma_2^{-1} + \bar{G}_{t,1}\right)^{-1}$ is constant. Thus

$$\mathbb{E}\left[\mathrm{cov}\left[\psi_{*,1} \mid H_t, \psi_{*,2}\right] \mid H_t\right] = \mathrm{cov}\left[\psi_{*,1} \mid H_t, \psi_{*,2}\right] = \bar{\Sigma}_{t,1}.$$

In addition, given $H_t$, $\bar{\Sigma}_{t,1}$, $\tilde{\Sigma}_{t,1}$ and $\bar{B}_{t,1}$ are constant. Thus

$$\begin{aligned}
\mathrm{cov}\left[\mathbb{E}\left[\psi_{*,1} \mid H_t, \psi_{*,2}\right] \mid H_t\right] &= \mathrm{cov}\left[\bar{\Sigma}_{t,1}\left(\Sigma_2^{-1}\mathrm{W}_2\psi_{*,2} + \bar{B}_{t,1}\right) \mid H_t\right], \\
&= \mathrm{cov}\left[\bar{\Sigma}_{t,1}\Sigma_2^{-1}\mathrm{W}_2\psi_{*,2} \mid H_t\right], \\
&= \bar{\Sigma}_{t,1}\Sigma_2^{-1}\mathrm{W}_2\mathrm{cov}\left[\psi_{*,2} \mid H_t\right]\mathrm{W}_2^\top\Sigma_2^{-1}\bar{\Sigma}_{t,1}, \\
&= \bar{\Sigma}_{t,1}\Sigma_2^{-1}\mathrm{W}_2\bar{\bar{\Sigma}}_{t,2}\mathrm{W}_2^\top\Sigma_2^{-1}\bar{\Sigma}_{t,1}.
\end{aligned}$$

Finally, total covariance decomposition (Weiss, 2005; Hong et al., 2022b) leads to

$$\begin{aligned}
\bar{\bar{\Sigma}}_{t,1} = \mathrm{cov}\left[\psi_{*,1} \mid H_t\right] &= \mathbb{E}\left[\mathrm{cov}\left[\psi_{*,1} \mid H_t, \psi_{*,2}\right] \mid H_t\right] + \mathrm{cov}\left[\mathbb{E}\left[\psi_{*,1} \mid H_t, \psi_{*,2}\right] \mid H_t\right], \\
&= \bar{\Sigma}_{t,1} + \bar{\Sigma}_{t,1}\Sigma_2^{-1}\mathrm{W}_2\bar{\bar{\Sigma}}_{t,2}\mathrm{W}_2^\top\Sigma_2^{-1}\bar{\Sigma}_{t,1}.
\end{aligned}$$

Now using the techniques, this can be generalized using the same technique as above to

$$\bar{\bar{\Sigma}}_{t,\ell} = \bar{\Sigma}_{t,\ell} + \bar{\Sigma}_{t,\ell}\Sigma_{\ell+1}^{-1}\mathrm{W}_{\ell+1}\bar{\bar{\Sigma}}_{t,\ell+1}\mathrm{W}_{\ell+1}^\top\Sigma_{\ell+1}^{-1}\bar{\Sigma}_{t,\ell}, \qquad \forall \ell \in [L-1].$$

Then, by induction, we get that

$$\bar{\bar{\Sigma}}_{t,1} = \sum_{\ell \in [L]} \bar{\mathrm{P}}_\ell\bar{\Sigma}_{t,\ell}\bar{\mathrm{P}}_\ell^\top, \qquad \forall \ell \in [L-1],$$

where we use that by definition $\bar{\bar{\Sigma}}_{t,L} = \mathrm{cov}\left[\psi_{*,L} \mid H_t\right] = \bar{\Sigma}_{t,L}$ and set $\bar{\mathrm{P}}_1 = I_d$ and $\bar{\mathrm{P}}_\ell = \prod_{k=1}^{\ell-1}\bar{\Sigma}_{t,k}\Sigma_{k+1}^{-1}\mathrm{W}_{k+1}$ for any $\ell \in [L]/\{1\}$. Plugging this in Eq. (32) leads to

$$\begin{aligned}
\check{\Sigma}_{t,i} &= \hat{\Sigma}_{t,i} + \sum_{\ell \in [L]} \hat{\Sigma}_{t,i}\Sigma_1^{-1}\mathrm{W}_1\bar{\mathrm{P}}_\ell\bar{\Sigma}_{t,\ell}\bar{\mathrm{P}}_\ell^\top\mathrm{W}_1^\top\Sigma_1^{-1}\hat{\Sigma}_{t,i}, \\
&= \hat{\Sigma}_{t,i} + \sum_{\ell \in [L]} \hat{\Sigma}_{t,i}\Sigma_1^{-1}\mathrm{W}_1\bar{\mathrm{P}}_\ell\bar{\Sigma}_{t,\ell}(\hat{\Sigma}_{t,i}\Sigma_1^{-1}\mathrm{W}_1)^\top, \\
&= \hat{\Sigma}_{t,i} + \sum_{\ell \in [L]} \mathrm{P}_{i,\ell}\bar{\Sigma}_{t,\ell}\mathrm{P}_{i,\ell}^\top,
\end{aligned}$$

where $\mathrm{P}_{i,\ell} = \hat{\Sigma}_{t,i}\Sigma_1^{-1}\mathrm{W}_1\bar{\mathrm{P}}_\ell = \hat{\Sigma}_{t,i}\Sigma_1^{-1}\mathrm{W}_1\prod_{k=1}^{\ell-1}\bar{\Sigma}_{t,k}\Sigma_{k+1}^{-1}\mathrm{W}_{k+1}$.

### D.4 PROOF OF LEMMA D.3

We prove this result by induction. We start with the base case when $\ell = 1$.

**(I) Base case.** Let $u = \sigma^{-1}\hat{\Sigma}_{t,A_t}^{\frac{1}{2}}X_t$ From the expression of $\bar{\Sigma}_{t,1}$ in Eq. (19), we have that

$$\bar{\Sigma}_{t+1,1}^{-1} - \bar{\Sigma}_{t,1}^{-1} = W_1^\top \left( \Sigma_1^{-1} - \Sigma_1^{-1}(\hat{\Sigma}_{t,A_t}^{-1} + \sigma^{-2}X_tX_t^\top)^{-1}\Sigma_1^{-1} - (\Sigma_1^{-1} - \Sigma_1^{-1}\hat{\Sigma}_{t,A_t}\Sigma_1^{-1}) \right) W_1 \,,$$

$$= W_1^\top \left( \Sigma_1^{-1}(\hat{\Sigma}_{t,A_t} - (\hat{\Sigma}_{t,A_t}^{-1} + \sigma^{-2}X_tX_t^\top)^{-1})\Sigma_1^{-1} \right) W_1 \,,$$

$$= W_1^\top \left( \Sigma_1^{-1}\hat{\Sigma}_{t,A_t}^{\frac{1}{2}}(I_d - (I_d + \sigma^{-2}\hat{\Sigma}_{t,A_t}^{\frac{1}{2}}X_tX_t^\top\hat{\Sigma}_{t,A_t}^{\frac{1}{2}})^{-1})\hat{\Sigma}_{t,A_t}^{\frac{1}{2}}\Sigma_1^{-1} \right) W_1 \,,$$

$$= W_1^\top \left( \Sigma_1^{-1}\hat{\Sigma}_{t,A_t}^{\frac{1}{2}}(I_d - (I_d + uu^\top)^{-1})\hat{\Sigma}_{t,A_t}^{\frac{1}{2}}\Sigma_1^{-1} \right) W_1 \,,$$

$$\overset{(i)}{=} W_1^\top \left( \Sigma_1^{-1}\hat{\Sigma}_{t,A_t}^{\frac{1}{2}}\frac{uu^\top}{1 + u^\top u}\hat{\Sigma}_{t,A_t}^{\frac{1}{2}}\Sigma_1^{-1} \right) W_1 \,,$$

$$\overset{(ii)}{=} \sigma^{-2}W_1^\top\Sigma_1^{-1}\hat{\Sigma}_{t,A_t}\frac{X_tX_t^\top}{1 + u^\top u}\hat{\Sigma}_{t,A_t}\Sigma_1^{-1}W_1 \,. \tag{33}$$

In $(i)$ we use the Sherman-Morrison formula. Note that $(ii)$ says that $\bar{\Sigma}_{t+1,1}^{-1} - \bar{\Sigma}_{t,1}^{-1}$ is one-rank which we will also need in induction step. Now, we have that $\|X_t\|^2 = 1$. Therefore,

$$1 + u^\top u = 1 + \sigma^{-2}X_t^\top\hat{\Sigma}_{t,A_t}X_t \leq 1 + \sigma^{-2}\lambda_1(\Sigma_1)\|X_t\|^2 = 1 + \sigma^{-2}\sigma_1^2 \leq \sigma_{\text{MAX}}^2 \,,$$

where we use that by definition of $\sigma_{\text{MAX}}^2$ in Lemma D.3, we have that $\sigma_{\text{MAX}}^2 \geq 1 + \sigma^{-2}\sigma_1^2$. Therefore, by taking the inverse, we get that $\frac{1}{1+u^\top u} \geq \sigma_{\text{MAX}}^{-2}$. Combining this with Eq. (33) leads to

$$\bar{\Sigma}_{t+1,1}^{-1} - \bar{\Sigma}_{t,1}^{-1} \succeq \sigma^{-2}\sigma_{\text{MAX}}^{-2}W_1^\top\Sigma_1^{-1}\hat{\Sigma}_{t,A_t}X_tX_t^\top\hat{\Sigma}_{t,A_t}\Sigma_1^{-1}W_1$$

Noticing that $\mathrm{P}_{A_t,1} = \hat{\Sigma}_{t,A_t}\Sigma_1^{-1}W_1$ concludes the proof of the base case when $\ell = 1$.

**(II) Induction step.** Let $\ell \in [L]/\{1\}$ and suppose that $\bar{\Sigma}_{t+1,\ell-1}^{-1} - \bar{\Sigma}_{t,\ell-1}^{-1}$ is one-rank and that it holds for $\ell - 1$ that

$$\bar{\Sigma}_{t+1,\ell-1}^{-1} - \bar{\Sigma}_{t,\ell-1}^{-1} \succeq \sigma^{-2}\sigma_{\text{MAX}}^{-2(\ell-1)}\mathrm{P}_{A_t,\ell-1}^\top X_tX_t^\top\mathrm{P}_{A_t,\ell-1} \,, \quad \text{where } \sigma_{\text{MAX}}^{-2} = \max_{\ell\in[L]}1 + \sigma^{-2}\sigma_\ell^2 \,.$$

Then, we want to show that $\bar{\Sigma}_{t+1,\ell}^{-1} - \bar{\Sigma}_{t,\ell}^{-1}$ is also one-rank and that it holds that

$$\bar{\Sigma}_{t+1,\ell}^{-1} - \bar{\Sigma}_{t,\ell}^{-1} \succeq \sigma^{-2}\sigma_{\text{MAX}}^{-2\ell}\mathrm{P}_{A_t,\ell}^\top X_tX_t^\top\mathrm{P}_{A_t,\ell} \,, \qquad \text{where } \sigma_{\text{MAX}}^{-2} = \max_{\ell\in[L]}1 + \sigma^{-2}\sigma_\ell^2 \,.$$

This is achieved as follows. First, we notice that by the induction hypothesis, we have that $\tilde{\Sigma}_{t+1,\ell-1}^{-1} - \bar{G}_{t,\ell-1} = \bar{\Sigma}_{t+1,\ell-1}^{-1} - \bar{\Sigma}_{t,\ell-1}^{-1}$ is one-rank. In addition, the matrix is positive semi-definite. Thus we can write it as $\tilde{\Sigma}_{t+1,\ell-1}^{-1} - \bar{G}_{t,\ell-1} = uu^\top$ where $u \in \mathbb{R}^d$. Then, similarly to the base case, we have

$$\bar{\Sigma}_{t+1,\ell}^{-1} - \bar{\Sigma}_{t,\ell}^{-1} = \tilde{\Sigma}_{t+1,\ell}^{-1} - \tilde{\Sigma}_{t,\ell}^{-1} \,,$$

$$= W_\ell^\top\left(\Sigma_\ell + \tilde{\Sigma}_{t+1,\ell-1}\right)^{-1}W_\ell - W_\ell^\top\left(\Sigma_\ell + \tilde{\Sigma}_{t,\ell-1}\right)^{-1}W_\ell \,,$$

$$= W_\ell^\top\left[\left(\Sigma_\ell + \tilde{\Sigma}_{t+1,\ell-1}\right)^{-1} - \left(\Sigma_\ell + \tilde{\Sigma}_{t,\ell-1}\right)^{-1}\right]W_\ell \,,$$

$$= W_\ell^\top\Sigma_\ell^{-1}\left[\left(\Sigma_\ell^{-1} + \bar{G}_{t,\ell-1}\right)^{-1} - \left(\Sigma_\ell^{-1} + \tilde{\Sigma}_{t+1,\ell-1}^{-1}\right)^{-1}\right]\Sigma_\ell^{-1}W_\ell \,,$$

$$= W_\ell^\top\Sigma_\ell^{-1}\left[\left(\Sigma_\ell^{-1} + \bar{G}_{t,\ell-1}\right)^{-1} - \left(\Sigma_\ell^{-1} + \bar{G}_{t,\ell-1} + \tilde{\Sigma}_{t+1,\ell-1}^{-1} - \bar{G}_{t,\ell-1}\right)^{-1}\right]\Sigma_\ell^{-1}W_\ell \,,$$

$$= W_\ell^\top\Sigma_\ell^{-1}\left[\left(\Sigma_\ell^{-1} + \bar{G}_{t,\ell-1}\right)^{-1} - \left(\Sigma_\ell^{-1} + \bar{G}_{t,\ell-1} + uu^\top\right)^{-1}\right]\Sigma_\ell^{-1}W_\ell \,,$$

$$= W_\ell^\top\Sigma_\ell^{-1}\left[\bar{\Sigma}_{t,\ell-1} - \left(\bar{\Sigma}_{t,\ell-1}^{-1} + uu^\top\right)^{-1}\right]\Sigma_\ell^{-1}W_\ell \,,$$

$$= W_\ell^\top\Sigma_\ell^{-1}\left[\bar{\Sigma}_{t,\ell-1}\frac{uu^\top}{1 + u^\top\bar{\Sigma}_{t,\ell-1}u}\bar{\Sigma}_{t,\ell-1}\right]\Sigma_\ell^{-1}W_\ell \,,$$

$$= W_\ell^\top\Sigma_\ell^{-1}\bar{\Sigma}_{t,\ell-1}\frac{uu^\top}{1 + u^\top\bar{\Sigma}_{t,\ell-1}u}\bar{\Sigma}_{t,\ell-1}\Sigma_\ell^{-1}W_\ell$$

However, we it follows from the induction hypothesis that $uu^\top = \tilde{\Sigma}_{t+1,\ell-1}^{-1} - \bar{G}_{t,\ell-1} = \bar{\Sigma}_{t+1,\ell-1}^{-1} - \bar{\Sigma}_{t,\ell-1}^{-1} \succeq \sigma^{-2}\sigma_{\mathrm{MAX}}^{-2(\ell-1)}\mathrm{P}_{A_t,\ell-1}^\top X_t X_t^\top \mathrm{P}_{A_t,\ell-1}$. Therefore,

$$\bar{\Sigma}_{t+1,\ell}^{-1} - \bar{\Sigma}_{t,\ell}^{-1} = \mathrm{W}_\ell^\top \Sigma_\ell^{-1}\bar{\Sigma}_{t,\ell-1}\frac{uu^\top}{1 + u^\top\bar{\Sigma}_{t,\ell-1}u}\bar{\Sigma}_{t,\ell-1}\Sigma_\ell^{-1}\mathrm{W}_\ell\,,$$

$$\succeq \mathrm{W}_\ell^\top \Sigma_\ell^{-1}\bar{\Sigma}_{t,\ell-1}\frac{\sigma^{-2}\sigma_{\mathrm{MAX}}^{-2(\ell-1)}\mathrm{P}_{A_t,\ell-1}^\top X_t X_t^\top \mathrm{P}_{A_t,\ell-1}}{1 + u^\top\bar{\Sigma}_{t,\ell-1}u}\bar{\Sigma}_{t,\ell-1}\Sigma_\ell^{-1}\mathrm{W}_\ell\,,$$

$$= \frac{\sigma^{-2}\sigma_{\mathrm{MAX}}^{-2(\ell-1)}}{1 + u^\top\bar{\Sigma}_{t,\ell-1}u}\mathrm{W}_\ell^\top \Sigma_\ell^{-1}\bar{\Sigma}_{t,\ell-1}\mathrm{P}_{A_t,\ell-1}^\top X_t X_t^\top \mathrm{P}_{A_t,\ell-1}\bar{\Sigma}_{t,\ell-1}\Sigma_\ell^{-1}\mathrm{W}_\ell\,,$$

$$= \frac{\sigma^{-2}\sigma_{\mathrm{MAX}}^{-2(\ell-1)}}{1 + u^\top\bar{\Sigma}_{t,\ell-1}u}\mathrm{P}_{A_t,\ell}^\top X_t X_t^\top \mathrm{P}_{A_t,\ell}\,.$$

Finally, we use that $1 + u^\top\bar{\Sigma}_{t,\ell-1}u \leq 1 + \|u\|_2\lambda_1(\bar{\Sigma}_{t,\ell-1}) \leq 1 + \sigma^{-2}\sigma_\ell^2$. Here we use that $\|u\|_2 \leq \sigma^{-2}$, which can also be proven by induction, and that $\lambda_1(\bar{\Sigma}_{t,\ell-1}) \leq \sigma_\ell^2$, which follows from the expression of $\bar{\Sigma}_{t,\ell-1}$ in Appendix B.2. Therefore, we have that

$$\bar{\Sigma}_{t+1,\ell}^{-1} - \bar{\Sigma}_{t,\ell}^{-1} \succeq \frac{\sigma^{-2}\sigma_{\mathrm{MAX}}^{-2(\ell-1)}}{1 + u^\top\bar{\Sigma}_{t,\ell-1}u}\mathrm{P}_{A_t,\ell}^\top X_t X_t^\top \mathrm{P}_{A_t,\ell}\,,$$

$$\succeq \frac{\sigma^{-2}\sigma_{\mathrm{MAX}}^{-2(\ell-1)}}{1 + \sigma^{-2}\sigma_\ell^2}\mathrm{P}_{A_t,\ell}^\top X_t X_t^\top \mathrm{P}_{A_t,\ell}\,,$$

$$\succeq \sigma^{-2}\sigma_{\mathrm{MAX}}^{-2\ell}\mathrm{P}_{A_t,\ell}^\top X_t X_t^\top \mathrm{P}_{A_t,\ell}\,,$$

where the last inequality follows from the definition of $\sigma_{\mathrm{MAX}}^2 = \max_{\ell\in[L]} 1 + \sigma^{-2}\sigma_\ell^2$. This concludes the proof.

## D.5 PROOF OF THEOREM 4.1

We start with the following standard result which we borrow from (Hong et al., 2022a; Aouali et al., 2023b),

$$\mathcal{BR}(n) \leq \sqrt{2n\log(1/\delta)}\sqrt{\mathbb{E}\left[\sum_{t=1}^{n}\|X_t\|_{\check{\Sigma}_{t,A_t}}^2\right]} + cn\delta\,, \qquad \text{where } c > 0 \text{ is a constant}\,. \tag{34}$$

Then we use Lemma D.2 and express the marginal covariance $\check{\Sigma}_{t,A_t}$ as

$$\check{\Sigma}_{t,i} = \hat{\Sigma}_{t,i} + \sum_{\ell\in[L]}\mathrm{P}_{i,\ell}\bar{\Sigma}_{t,\ell}\mathrm{P}_{i,\ell}^\top\,, \qquad \text{where } \mathrm{P}_{i,\ell} = \hat{\Sigma}_{t,i}\Sigma_1^{-1}\mathrm{W}_1\prod_{k=1}^{\ell-1}\bar{\Sigma}_{t,k}\Sigma_{k+1}^{-1}\mathrm{W}_{k+1}\,. \tag{35}$$

Therefore, we can decompose $\|X_t\|_{\check{\Sigma}_{t,A_t}}^2$ as

$$\|X_t\|_{\check{\Sigma}_{t,A_t}}^2 = \sigma^2\frac{X_t^\top\check{\Sigma}_{t,A_t}X_t}{\sigma^2} \stackrel{(i)}{=} \sigma^2\left(\sigma^{-2}X_t^\top\hat{\Sigma}_{t,A_t}X_t + \sigma^{-2}\sum_{\ell\in[L]}X_t^\top\mathrm{P}_{A_t,\ell}\bar{\Sigma}_{t,\ell}\mathrm{P}_{A_t,\ell}^\top X_t\right)\,,$$

$$\stackrel{(ii)}{\leq} c_0\log(1 + \sigma^{-2}X_t^\top\hat{\Sigma}_{t,A_t}X_t) + \sum_{\ell\in[L]}c_\ell\log(1 + \sigma^{-2}X_t^\top\mathrm{P}_{A_t,\ell}\bar{\Sigma}_{t,\ell}\mathrm{P}_{A_t,\ell}^\top X_t)\,, \tag{36}$$

where $(i)$ follows from Eq. (35), and we use the following inequality in $(ii)$

$$x = \frac{x}{\log(1+x)}\log(1+x) \leq \left(\max_{x\in[0,u]}\frac{x}{\log(1+x)}\right)\log(1+x) = \frac{u}{\log(1+u)}\log(1+x)\,,$$

which holds for any $x \in [0, u]$, where constants $c_0$ and $c_\ell$ are derived as

$$c_0 = \frac{\sigma_1^2}{\log(1 + \frac{\sigma_1^2}{\sigma^2})}\,, \quad c_\ell = \frac{\sigma_{\ell+1}^2}{\log(1 + \frac{\sigma_{\ell+1}^2}{\sigma^2})}\,, \text{with the convention that } \sigma_{L+1} = 1\,.$$

The derivation of $c_0$ uses that
$$X_t^\top \hat{\Sigma}_{t,A_t} X_t \leq \lambda_1(\hat{\Sigma}_{t,A_t}) \|X_t\|^2 \leq \lambda_d^{-1}(\Sigma_1^{-1} + G_{t,A_t}) \leq \lambda_d^{-1}(\Sigma_1^{-1}) = \lambda_1(\Sigma_1) = \sigma_1^2 \,.$$
The derivation of $c_\ell$ follows from
$$X_t^\top P_{A_t,\ell} \bar{\Sigma}_{t,\ell} P_{A_t,\ell}^\top X_t \leq \lambda_1(P_{A_t,\ell} P_{A_t,\ell}^\top) \lambda_1(\bar{\Sigma}_{t,\ell}) \|X_t\|^2 \leq \sigma_{\ell+1}^2 \,.$$
Therefore, from Eq. (36) and Eq. (34), we get that

$$\mathcal{BR}(n) \leq \sqrt{2n\log(1/\delta)} \Big( \mathbb{E}\Big[ c_0 \sum_{t=1}^n \log(1 + \sigma^{-2} X_t^\top \hat{\Sigma}_{t,A_t} X_t)$$

$$+ \sum_{\ell \in [L]} c_\ell \sum_{t=1}^n \log(1 + \sigma^{-2} X_t^\top P_{A_t,\ell} \bar{\Sigma}_{t,\ell} P_{A_t,\ell}^\top X_t) \Big] \Big)^{\frac{1}{2}} + cn\delta \qquad (37)$$

Now we focus on bounding the logarithmic terms in Eq. (37).

**(I) First term in Eq. (37)** We first rewrite this term as

$$\log(1 + \sigma^{-2} X_t^\top \hat{\Sigma}_{t,A_t} X_t) \overset{(i)}{=} \log\det(I_d + \sigma^{-2} \hat{\Sigma}_{t,A_t}^{\frac{1}{2}} X_t X_t^\top \hat{\Sigma}_{t,A_t}^{\frac{1}{2}}) \,,$$
$$= \log\det(\hat{\Sigma}_{t,A_t}^{-1} + \sigma^{-2} X_t X_t^\top) - \log\det(\hat{\Sigma}_{t,A_t}^{-1}) = \log\det(\hat{\Sigma}_{t+1,A_t}^{-1}) - \log\det(\hat{\Sigma}_{t,A_t}^{-1}) \,,$$

where $(i)$ follows from the Weinstein–Aronszajn identity. Then we sum over all rounds $t \in [n]$, and get a telescoping

$$\sum_{t=1}^n \log\det(I_d + \sigma^{-2} \hat{\Sigma}_{t,A_t}^{\frac{1}{2}} X_t X_t^\top \hat{\Sigma}_{t,A_t}^{\frac{1}{2}}) = \sum_{t=1}^n \log\det(\hat{\Sigma}_{t+1,A_t}^{-1}) - \log\det(\hat{\Sigma}_{t,A_t}^{-1}) \,,$$

$$= \sum_{t=1}^n \sum_{i=1}^K \log\det(\hat{\Sigma}_{t+1,i}^{-1}) - \log\det(\hat{\Sigma}_{t,i}^{-1}) = \sum_{i=1}^K \sum_{t=1}^n \log\det(\hat{\Sigma}_{t+1,i}^{-1}) - \log\det(\hat{\Sigma}_{t,i}^{-1}) \,,$$

$$= \sum_{i=1}^K \log\det(\hat{\Sigma}_{n+1,i}^{-1}) - \log\det(\hat{\Sigma}_{1,i}^{-1}) \overset{(i)}{=} \sum_{i=1}^K \log\det(\Sigma_1^{\frac{1}{2}} \hat{\Sigma}_{n+1,i}^{-1} \Sigma_1^{\frac{1}{2}}) \,,$$

where $(i)$ follows from the fact that $\hat{\Sigma}_{1,i} = \Sigma_1$. Now we use the inequality of arithmetic and geometric means and get

$$\sum_{t=1}^n \log\det(I_d + \sigma^{-2} \hat{\Sigma}_{t,A_t}^{\frac{1}{2}} X_t X_t^\top \hat{\Sigma}_{t,A_t}^{\frac{1}{2}}) = \sum_{i=1}^K \log\det(\Sigma_1^{\frac{1}{2}} \hat{\Sigma}_{n+1,i}^{-1} \Sigma_1^{\frac{1}{2}}) \,,$$

$$\leq \sum_{i=1}^K d\log\left( \frac{1}{d} \text{Tr}(\Sigma_1^{\frac{1}{2}} \hat{\Sigma}_{n+1,i}^{-1} \Sigma_1^{\frac{1}{2}}) \right) \,, \qquad (38)$$

$$\leq \sum_{i=1}^K d\log\left( 1 + \frac{n}{d}\frac{\sigma_1^2}{\sigma^2} \right) = Kd\log\left( 1 + \frac{n}{d}\frac{\sigma_1^2}{\sigma^2} \right) \,.$$

**(II) Remaining terms in Eq. (37)** Let $\ell \in [L]$. Then we have that

$$\log(1 + \sigma^{-2} X_t^\top P_{A_t,\ell} \bar{\Sigma}_{t,\ell} P_{A_t,\ell}^\top X_t) = \sigma_{\text{MAX}}^{2\ell} \sigma_{\text{MAX}}^{-2\ell} \log(1 + \sigma^{-2} X_t^\top P_{A_t,\ell} \bar{\Sigma}_{t,\ell} P_{A_t,\ell}^\top X_t) \,,$$

$$\leq \sigma_{\text{MAX}}^{2\ell} \log(1 + \sigma^{-2} \sigma_{\text{MAX}}^{-2\ell} X_t^\top P_{A_t,\ell} \bar{\Sigma}_{t,\ell} P_{A_t,\ell}^\top X_t) \,,$$

$$\overset{(i)}{=} \sigma_{\text{MAX}}^{2\ell} \log\det(I_d + \sigma^{-2} \sigma_{\text{MAX}}^{-2\ell} \bar{\Sigma}_{t,\ell}^{\frac{1}{2}} P_{A_t,\ell}^\top X_t X_t^\top P_{A_t,\ell} \bar{\Sigma}_{t,\ell}^{\frac{1}{2}}) \,,$$

$$= \sigma_{\text{MAX}}^{2\ell} \Big( \log\det(\bar{\Sigma}_{t,\ell}^{-1} + \sigma^{-2} \sigma_{\text{MAX}}^{-2\ell} P_{A_t,\ell}^\top X_t X_t^\top P_{A_t,\ell}) - \log\det(\bar{\Sigma}_{t,\ell}^{-1}) \Big) \,,$$

where we use the Weinstein–Aronszajn identity in $(i)$. Now we know from Lemma D.3 that the following inequality holds $\sigma^{-2} \sigma_{\text{MAX}}^{-2\ell} P_{A_t,\ell}^\top X_t X_t^\top P_{A_t,\ell} \preceq \bar{\Sigma}_{t+1,\ell}^{-1} - \bar{\Sigma}_{t,\ell}^{-1}$. As a result, we get that $\bar{\Sigma}_{t,\ell}^{-1} + \sigma^{-2} \sigma_{\text{MAX}}^{-2\ell} P_{A_t,\ell}^\top X_t X_t^\top P_{A_t,\ell} \preceq \bar{\Sigma}_{t+1,\ell}^{-1}$. Thus,

$$\log(1 + \sigma^{-2} X_t^\top P_{A_t,\ell} \bar{\Sigma}_{t,\ell} P_{A_t,\ell}^\top X_t) \leq \sigma_{\text{MAX}}^{2\ell} \Big( \log\det(\bar{\Sigma}_{t+1,\ell}^{-1}) - \log\det(\bar{\Sigma}_{t,\ell}^{-1}) \Big) \,,$$

Then we sum over all rounds $t \in [n]$, and get a telescoping

$$\sum_{t=1}^{n} \log(1 + \sigma^{-2} X_t^\top P_{A_t,\ell} \bar{\Sigma}_{t,\ell} P_{A_t,\ell}^\top X_t) \leq \sigma_{\text{MAX}}^{2\ell} \sum_{t=1}^{n} \log \det(\bar{\Sigma}_{t+1,\ell}^{-1}) - \log \det(\bar{\Sigma}_{t,\ell}^{-1}),$$

$$= \sigma_{\text{MAX}}^{2\ell} \left( \log \det(\bar{\Sigma}_{n+1,\ell}^{-1}) - \log \det(\bar{\Sigma}_{1,\ell}^{-1}) \right),$$

$$\overset{(i)}{=} \sigma_{\text{MAX}}^{2\ell} \left( \log \det(\bar{\Sigma}_{n+1,\ell}^{-1}) - \log \det(\Sigma_{\ell+1}^{-1}) \right),$$

$$= \sigma_{\text{MAX}}^{2\ell} \left( \log \det(\Sigma_{\ell+1}^{\frac{1}{2}} \bar{\Sigma}_{n+1,\ell}^{-1} \Sigma_{\ell+1}^{\frac{1}{2}}) \right),$$

where we use that $\bar{\Sigma}_{1,\ell} = \Sigma_{\ell+1}$ in $(i)$. Finally, we use the inequality of arithmetic and geometric means and get that

$$\sum_{t=1}^{n} \log(1 + \sigma^{-2} X_t^\top P_{A_t,\ell} \bar{\Sigma}_{t,\ell} P_{A_t,\ell}^\top X_t) \leq \sigma_{\text{MAX}}^{2\ell} \left( \log \det(\Sigma_{\ell+1}^{\frac{1}{2}} \bar{\Sigma}_{n+1,\ell}^{-1} \Sigma_{\ell+1}^{\frac{1}{2}}) \right),$$

$$\leq d\sigma_{\text{MAX}}^{2\ell} \log \left( \frac{1}{d} \text{Tr}(\Sigma_{\ell+1}^{\frac{1}{2}} \bar{\Sigma}_{n+1,\ell}^{-1} \Sigma_{\ell+1}^{\frac{1}{2}}) \right), \quad (39)$$

$$\leq d\sigma_{\text{MAX}}^{2\ell} \log \left( 1 + \frac{\sigma_{\ell+1}^2}{\sigma_\ell^2} \right),$$

The last inequality follows from the expression of $\bar{\Sigma}_{n+1,\ell}^{-1}$ in Eq. (19) that leads to

$$\Sigma_{\ell+1}^{\frac{1}{2}} \bar{\Sigma}_{n+1,\ell}^{-1} \Sigma_{\ell+1}^{\frac{1}{2}} = I_d + \Sigma_{\ell+1}^{\frac{1}{2}} \bar{G}_{t,\ell} \Sigma_{\ell+1}^{\frac{1}{2}},$$

$$= I_d + \Sigma_{\ell+1}^{\frac{1}{2}} W_\ell^\top \left( \Sigma_\ell^{-1} - \Sigma_\ell^{-1} \bar{\Sigma}_{t,\ell-1} \Sigma_\ell^{-1} \right) W_\ell \Sigma_{\ell+1}^{\frac{1}{2}}, \quad (40)$$

since $\bar{G}_{t,\ell} = W_\ell^\top \left( \Sigma_\ell^{-1} - \Sigma_\ell^{-1} \bar{\Sigma}_{t,\ell-1} \Sigma_\ell^{-1} \right) W_\ell$. This allows us to bound $\frac{1}{d} \text{Tr}(\Sigma_{\ell+1}^{\frac{1}{2}} \bar{\Sigma}_{n+1,\ell}^{-1} \Sigma_{\ell+1}^{\frac{1}{2}})$ as

$$\frac{1}{d} \text{Tr}(\Sigma_{\ell+1}^{\frac{1}{2}} \bar{\Sigma}_{n+1,\ell}^{-1} \Sigma_{\ell+1}^{\frac{1}{2}}) = \frac{1}{d} \text{Tr}(I_d + \Sigma_{\ell+1}^{\frac{1}{2}} W_\ell^\top \left( \Sigma_\ell^{-1} - \Sigma_\ell^{-1} \bar{\Sigma}_{t,\ell-1} \Sigma_\ell^{-1} \right) W_\ell \Sigma_{\ell+1}^{\frac{1}{2}}),$$

$$= \frac{1}{d}(d + \text{Tr}(\Sigma_{\ell+1}^{\frac{1}{2}} W_\ell^\top \left( \Sigma_\ell^{-1} - \Sigma_\ell^{-1} \bar{\Sigma}_{t,\ell-1} \Sigma_\ell^{-1} \right) W_\ell \Sigma_{\ell+1}^{\frac{1}{2}}),$$

$$\leq 1 + \frac{1}{d} \sum_{k=1}^{d} \lambda_1(\Sigma_{\ell+1}^{\frac{1}{2}} W_\ell^\top \left( \Sigma_\ell^{-1} - \Sigma_\ell^{-1} \bar{\Sigma}_{t,\ell-1} \Sigma_\ell^{-1} \right) W_\ell \Sigma_{\ell+1}^{\frac{1}{2}}),$$

$$\leq 1 + \frac{1}{d} \sum_{k=1}^{d} \lambda_1(\Sigma_{\ell+1}) \lambda_1(W_\ell^\top W_\ell) \lambda_1 \left( \Sigma_\ell^{-1} - \Sigma_\ell^{-1} \bar{\Sigma}_{t,\ell-1} \Sigma_\ell^{-1} \right),$$

$$\leq 1 + \frac{1}{d} \sum_{k=1}^{d} \lambda_1(\Sigma_{\ell+1}) \lambda_1(W_\ell^\top W_\ell) \lambda_1 \left( \Sigma_\ell^{-1} \right),$$

$$\leq 1 + \frac{1}{d} \sum_{k=1}^{d} \frac{\sigma_{\ell+1}^2}{\sigma_\ell^2} = 1 + \frac{\sigma_{\ell+1}^2}{\sigma_\ell^2}, \quad (41)$$

where we use the assumption that $\lambda_1(W_\ell^\top W_\ell) = 1$ **(A2)** and that $\lambda_1(\Sigma_{\ell+1}) = \sigma_{\ell+1}^2$ and $\lambda_1(\Sigma_\ell^{-1}) = 1/\sigma_\ell^2$. This is because $\Sigma_\ell = \sigma_\ell^2 I_d$ for any $\ell \in [L+1]$. Finally, plugging Eqs. (38) and (39) in Eq. (37) concludes the proof.

### D.6 PROOF OF PROPOSITION 4.2

We use exactly the same proof in Appendix D.5, with one change to account for the sparsity assumption **(A3)**. The change corresponds to Eq. (39). First, recall that Eq. (39) writes

$$\sum_{t=1}^{n} \log(1 + \sigma^{-2} X_t^\top P_{A_t,\ell} \bar{\Sigma}_{t,\ell} P_{A_t,\ell}^\top X_t) \leq \sigma_{\text{MAX}}^{2\ell} \left( \log \det(\Sigma_{\ell+1}^{\frac{1}{2}} \bar{\Sigma}_{n+1,\ell}^{-1} \Sigma_{\ell+1}^{\frac{1}{2}}) \right),$$

where

$$\Sigma_{\ell+1}^{\frac{1}{2}}\bar{\Sigma}_{n+1,\ell}^{-1}\Sigma_{\ell+1}^{\frac{1}{2}} = I_d + \Sigma_{\ell+1}^{\frac{1}{2}}W_\ell^\top\big(\Sigma_\ell^{-1} - \Sigma_\ell^{-1}\bar{\Sigma}_{t,\ell-1}\Sigma_\ell^{-1}\big)W_\ell\Sigma_{\ell+1}^{\frac{1}{2}}\,,$$

$$= I_d + \sigma_{\ell+1}^2 W_\ell^\top\big(\Sigma_\ell^{-1} - \Sigma_\ell^{-1}\bar{\Sigma}_{t,\ell-1}\Sigma_\ell^{-1}\big)W_\ell\,, \qquad (42)$$

where the second equality follows from the assumption that $\Sigma_{\ell+1} = \sigma_{\ell+1}^2 I_d$. But notice that in our assumption, **(A3)**, we assume that $W_\ell = (\bar{W}_\ell, 0_{d,d-d_\ell})$, where $\bar{W}_\ell \in \mathbb{R}^{d\times d_\ell}$ for any $\ell \in [L]$. Therefore, we have that for any $d \times d$ matrix $B \in \mathbb{R}^{dd\times d}$, the following holds, $W_\ell^\top BW_\ell = \begin{pmatrix} \bar{W}_\ell^\top B\bar{W}_\ell & 0_{d_\ell,d-d_\ell} \\ 0_{d-d_\ell,d_\ell} & 0_{d-d_\ell,d-d_\ell} \end{pmatrix}$. In particular, we have that

$$W_\ell^\top\big(\Sigma_\ell^{-1} - \Sigma_\ell^{-1}\bar{\Sigma}_{t,\ell-1}\Sigma_\ell^{-1}\big)W_\ell = \begin{pmatrix} \bar{W}_\ell^\top\big(\Sigma_\ell^{-1} - \Sigma_\ell^{-1}\bar{\Sigma}_{t,\ell-1}\Sigma_\ell^{-1}\big)\bar{W}_\ell & 0_{d_\ell,d-d_\ell} \\ 0_{d-d_\ell,d_\ell} & 0_{d-d_\ell,d-d_\ell} \end{pmatrix}. \quad (43)$$

Therefore, plugging this in Eq. (42) yields that

$$\Sigma_{\ell+1}^{\frac{1}{2}}\bar{\Sigma}_{n+1,\ell}^{-1}\Sigma_{\ell+1}^{\frac{1}{2}} = \begin{pmatrix} I_{d_\ell} + \sigma_{\ell+1}^2\bar{W}_\ell^\top\big(\Sigma_\ell^{-1} - \Sigma_\ell^{-1}\bar{\Sigma}_{t,\ell-1}\Sigma_\ell^{-1}\big)\bar{W}_\ell & 0_{d_\ell,d-d_\ell} \\ 0_{d-d_\ell,d_\ell} & I_{d-d_\ell} \end{pmatrix}. \quad (44)$$

As a result, $\det(\Sigma_{\ell+1}^{\frac{1}{2}}\bar{\Sigma}_{n+1,\ell}^{-1}\Sigma_{\ell+1}^{\frac{1}{2}}) = \det(I_{d_\ell} + \sigma_{\ell+1}^2\bar{W}_\ell^\top\big(\Sigma_\ell^{-1} - \Sigma_\ell^{-1}\bar{\Sigma}_{t,\ell-1}\Sigma_\ell^{-1}\big)\bar{W}_\ell)$. This allows us to move the problem from a $d$-dimensional one to a $d_\ell$-dimensional one. Then we use the inequality of arithmetic and geometric means and get that

$$\sum_{t=1}^n \log(1 + \sigma^{-2}X_t^\top P_{A_t,\ell}\bar{\Sigma}_{t,\ell}P_{A_t,\ell}^\top X_t) \leq \sigma_{\text{MAX}}^{2\ell}\Big(\log\det(\Sigma_{\ell+1}^{\frac{1}{2}}\bar{\Sigma}_{n+1,\ell}^{-1}\Sigma_{\ell+1}^{\frac{1}{2}})\Big)\,,$$

$$= \sigma_{\text{MAX}}^{2\ell}\log\det(I_{d_\ell} + \sigma_{\ell+1}^2\bar{W}_\ell^\top\big(\Sigma_\ell^{-1} - \Sigma_\ell^{-1}\bar{\Sigma}_{t,\ell-1}\Sigma_\ell^{-1}\big)\bar{W}_\ell)\,,$$

$$\leq d_\ell\sigma_{\text{MAX}}^{2\ell}\log\left(\frac{1}{d_\ell}\text{Tr}(I_{d_\ell} + \sigma_{\ell+1}^2\bar{W}_\ell^\top\big(\Sigma_\ell^{-1} - \Sigma_\ell^{-1}\bar{\Sigma}_{t,\ell-1}\Sigma_\ell^{-1}\big)\bar{W}_\ell)\right)\,,$$

$$\leq d_\ell\sigma_{\text{MAX}}^{2\ell}\log\left(1 + \frac{\sigma_{\ell+1}^2}{\sigma_\ell^2}\right)\,. \qquad (45)$$

To get the last inequality, we use derivations similar to the ones we used in Eq. (41). Finally, the desired result in obtained by replacing Eq. (39) by Eq. (45) in the previous proof in Appendix D.5.

## D.7 ADDITIONAL DISCUSSION: LINK TO TWO-LEVEL HIERARCHIES

The linear diffusion Eq. (16) can be marginalized into a 2-level hierarchy using two different strategies. The first one yields,

$$\psi_{*,L} \sim \mathcal{N}(0, \sigma_{L+1}^2 B_L B_L^\top)\,, \qquad (46)$$

$$\theta_{*,i} \mid \psi_{*,L} \sim \mathcal{N}(\psi_{*,L}, \Omega_1)\,, \qquad\qquad \forall i \in [K]\,,$$

with $\Omega_1 = \sigma_1^2 I_d + \sum_{\ell=1}^{L-1}\sigma_{\ell+1}^2 B_\ell B_\ell^\top$ and $B_\ell = \prod_{k=1}^\ell W_k$. The second strategy yields,

$$\psi_{*,1} \sim \mathcal{N}(0, \Omega_2)\,, \qquad (47)$$

$$\theta_{*,i} \mid \psi_{*,1} \sim \mathcal{N}(\psi_{*,1}, \sigma_1^2 I_d)\,, \qquad\qquad \forall i \in [K]\,,$$

where $\Omega_2 = \sum_{\ell=1}^L\sigma_{\ell+1}^2 B_\ell B_\ell^\top$. Recently, `HierTS` (Hong et al., 2022b) was developed for such two-level graphical models, and we call `HierTS` under Eq. (46) by `HierTS-1` and `HierTS` under Eq. (47) by `HierTS-2`. Then, we start by highlighting the differences between these two variants of `HierTS`. First, their regret bounds scale as

$$\texttt{HierTS-1}: \tilde{\mathcal{O}}\big(\sqrt{nd(K\textstyle\sum_{\ell=1}^L\sigma_\ell^2 + L\sigma_{L+1}^2)}\big)\,, \quad \texttt{HierTS-2}: \tilde{\mathcal{O}}\big(\sqrt{nd(K\sigma_1^2 + \textstyle\sum_{\ell=1}^L\sigma_{\ell+1}^2)}\big)\,.$$

When $K \approx L$, the regret bounds of `HierTS-1` and `HierTS-2` are similar. However, when $K > L$, `HierTS-2` outperforms `HierTS-1`. This is because `HierTS-2` puts more uncertainty on a single $d$-dimensional latent parameter $\psi_{*,1}$, rather than $K$ individual $d$-dimensional action parameters $\theta_{*,i}$. More importantly, `HierTS-1` implicitly assumes that action parameters $\theta_{*,i}$ are conditionally

independent given $\psi_{*,L}$, which is not true. Consequently, `HierTS-2` outperforms `HierTS-1`. Note that, under the linear diffusion model Eq. (16), `dTS` and `HierTS-2` have roughly similar regret bounds. Specifically, their regret bounds dependency on $K$ is identical, where both methods involve multiplying $K$ by $\sigma_1^2$, and both enjoy improved performance compared to `HierTS-1`. That said, note that Theorem 4.1 and Proposition 4.2 provide an understanding of how `dTS`'s regret scales under linear link functions $f_\ell$, and do not say that using `dTS` is better than using `HierTS` when the link functions $f_\ell$ are linear since the latter can be obtained by a proper marginalization of latent parameters (i.e., `HierTS-2` instead of `HierTS-1`). While such a comparison is not the goal of this work, we still provide it for completeness next.

When the mixing matrices $W_\ell$ are dense (i.e., assumption **(A3)** is not applicable), `dTS` and `HierTS-2` have comparable regret bounds and computational efficiency. However, under the sparsity assumption **(A3)** and with mixing matrices that allow for conditional independence of $\psi_{*,1}$ coordinates given $\psi_{*,2}$, `dTS` enjoys a computational advantage over `HierTS-2`. This advantage explains why works focusing on multi-level hierarchies typically benchmark their algorithms against two-level structures akin to `HierTS-1`, rather than the more competitive `HierTS-2`. This is also consistent with prior works in Bayesian bandits using multi-level hierarchies, such as Tree-based priors (Hong et al., 2022a), which compared their method to `HierTS-1`. In line with this, we also compared `dTS` with `HierTS-1` in our experiments. But this is only given for completeness as this is not the aim of Theorem 4.1 and Proposition 4.2. More importantly, `HierTS` is inapplicable in the general case in Eq. (1) with non-linear link functions since the latent parameters cannot be analytically marginalized.

## E    ADDITIONAL EXPERIMENTAL DETAILS

### E.1    SWISS ROLL DATA

Fig. 5 shows samples from the Swiss roll data and samples from generated by the pre-trained diffusion model for different pre-training sample sizes.

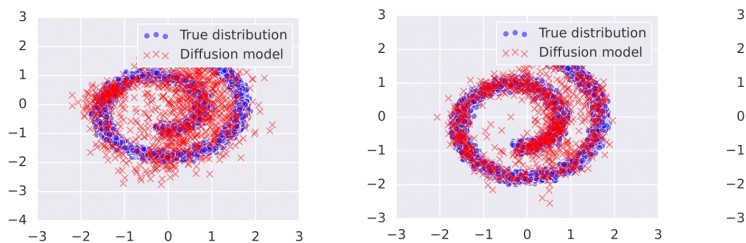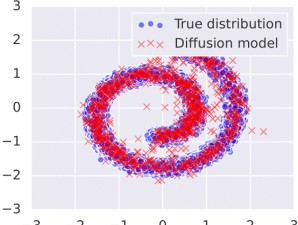

(a) Diffusion pre-trained on 50 samples from the Swiss roll dataset.    (b) Diffusion pre-trained on $10^3$ samples from the Swiss roll dataset.    (c) Diffusion pre-trained on $10^4$ samples from the Swiss roll dataset.

Figure 5: True distribution of action parameters (blue) vs. distribution of pre-trained diffusion model (red).

### E.2    DIFFUSION MODELS PRE-TRAINING

We used JAX for diffusion model pre-training, summarized as follows:

- **Parameterization:** Functions $f_\ell$ are parameterized with a fully connected 2-layer neural network (NN) with ReLU activation. The step $\ell$ is provided as input to capture the current sampling stage. Covariances are fixed (not learned) as $\Sigma_\ell = \sigma_\ell^2 I_d$ with $\sigma_\ell$ increasing with $\ell$.
- **Loss:** Offline data samples are progressively noised over steps $\ell \in [L]$, creating increasingly noisy versions of the data following a predefined noise schedule (Ho et al., 2020). The NN is trained to reverse this noise (i.e., denoise) by predicting the noise added at each step. The loss function measures the $L_2$ norm difference between the predicted and actual noise at each step, as explained in Ho et al. (2020).

- **Optimization:** Adam optimizer with a $10^{-3}$ learning rate was used. The NN was trained for 20,000 epochs with a batch size of min(2048, pre-training sample size). We used CPUs for pre-training, which was efficient enough to conduct multiple ablation studies.

- **After pre-training:** The pre-trained diffusion model is used as a prior for dTS and compared to LinTS as the reference baseline. In our ablation study, we plot the cumulative regret of LinTS in the last round divided by that of dTS. A ratio greater than 1 indicates that dTS outperforms LinTS, with higher values representing a larger performance gap.

### E.3 QUALITY OF OUR POSTERIOR APPROXIMATION

To assess the quality of our posterior approximation, we consider the scenario where the true distribution of action parameters is $\mathcal{N}(0_d, I_d)$ with $d = 2$ and rewards are linear. We pre-train a diffusion model using samples drawn from $\mathcal{N}(0_d, I_d)$. We then consider two priors: the true prior $\mathcal{N}(0_d, I_d)$ and the pre-trained diffusion model prior. This yields two posteriors:

- $P_1$ : Uses $\mathcal{N}(0_d, I_d)$ as the prior. $P_1$ is an exact posterior since the prior is Gaussian and rewards are linear-Gaussian.

- $P_2$ : Uses the pre-trained diffusion model as the prior. $P_2$ is our approximate posterior.

The learned diffusion model prior matches the true Gaussian prior (as seen in Fig. 7a). Thus, if our approximation is accurate, their posteriors $P_1$ and $P_2$ should also be similar. This is observed in Fig. 7b where the approximate posterior $P_2$ nearly matches the exact posterior $P_1$.

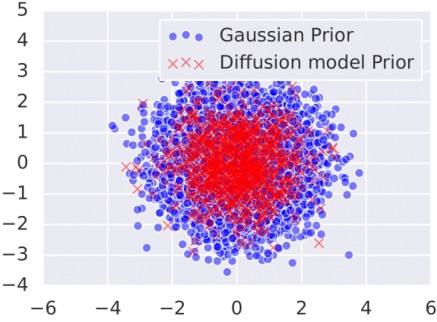
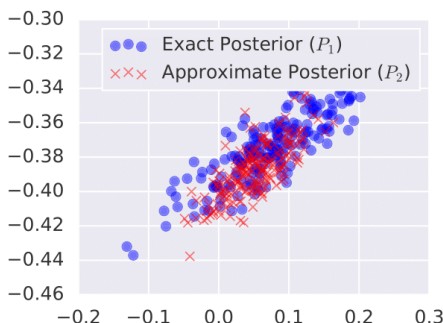

(a) Gaussian distribution vs. diffusion model pre-trained on $10^3$ samples drawn from it.

(b) Exact posterior $P_1$ vs. approximate posterior $P_2$ after $n = 100$ rounds of interactions.

Figure 6: Assessing the quality of our posterior approximation.

### E.4 BOUND COMPARISON

Here, we compare our bound in Theorem 4.1 to bounds of LinTS from the literature.

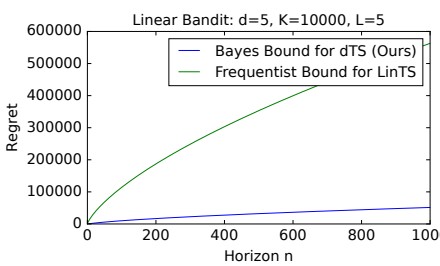 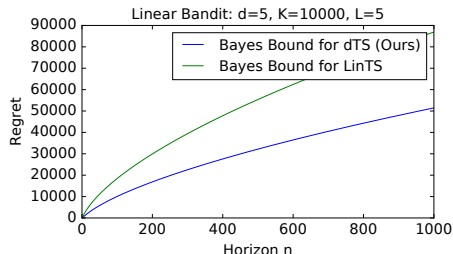

(a) Comparing our bound to the frequentist bound of `LinTS` in Abeille & Lazaric (2017).

(b) Comparing our bound to the standard Bayesian bound of `LinTS`.

Figure 7: Comparing our bound to the frequentist and Bayesian bounds of `LinTS`.

## F BROADER IMPACT

This work contributes to the development and analysis of practical algorithms for online learning to act under uncertainty. While our generic setting and algorithms have broad potential applications, the specific downstream social impacts are inherently dependent on the chosen application domain. Nevertheless, we acknowledge the crucial need to consider potential biases that may be present in pre-trained diffusion models, given that our method relies on them.

## G LIMITATIONS AND FUTURE RESEARCH

We designed diffusion Thompson sampling (`dTS`); for which we developed both theoretical and algorithmic foundations in numerous practical settings. We identified several directions for future work. Exploring other approximations for non-linear diffusion models, both empirically and theoretically. For theory, future research could explore the advantages of non-linear diffusion models by deriving their Bayes regret bounds, akin to our analysis in Section 4. Empirically, investigating our and other approximations in complex tasks would be interesting. Additionally, exploring the extension of this work to offline (or off-policy) learning in contextual bandits (Swaminathan & Joachims, 2015; Aouali et al., 2023a) represents a promising avenue for future research. Our work focused on contextual bandits, laying the groundwork for future exploration into reinforcement learning. This exploration can also be done from both practical (empirical) and theoretical angles. Finally, while our method, which approximates rewards using a Gaussian distribution, worked well for linear rewards and those following a generalized linear model, its effectiveness in real-world, complex scenarios needs further testing.

## H AMOUNT OF COMPUTATION REQUIRED

Our experiments were conducted on internal machines with 30 CPUs and thus they required a moderate amount of computation. These experiments are also reproducible with minimal computational resources.

