# OpenReview forum: "Diffusion Models Meet Contextual Bandits"
_ICLR.cc/2025/Conference — Submitted to ICLR 2025_

### Official Review · Reviewer_QyxM · 2024-11-01

**Soundness:** 3
**Presentation:** 3
**Contribution:** 2
**Rating:** 5
**Confidence:** 3

**Summary:**

The paper considers the classical contextual bandit problem from a Bayesian perspective, particularly through the use of a diffusion model. The arms are correlated via a shared distribution for their parameters. The paper proposes a new Thompson sampling algorithm that incorporates a diffusion model prior. The estimation of the posterior distribution and efficient sampling method are applicable to other Thompson sampling problems. The effectiveness of the algorithm is demonstrated by the regret bound in a linear case, where the regret is of order $\sqrt{KT(\log{T})^2}$. Additionally, the paper presents a numerical study to validate the algorithm, showing improvement over existing benchmarks in cases where the true prior is a diffusion model as well as in cases where the true prior is not a diffusion model.

**Strengths:**

1. The paper proposes a new algorithm for contextual bandits with dependent arms, using diffusion models.
2. The paper is relatively clearly written and easy to follow.
3. The paper includes a variety of interpretations of the results, both theoretical and numerical.
4. The theoretical statements and numerical experiments are solid, as they are clearly presented and explained. The regret bound is nearly optimal in that it is approximately of order $\sqrt{T}$, when the $\log{T}$ term is negligible.

**Weaknesses:**

1. The actual role of using a diffusion model remains unclear. If the prior is not a diffusion model, will the algorithm still perform effectively in this scenario?

2. The theoretical statements provided are only valid for the linear case. It is also unclear whether there would be a significant gap in performance between linear and non-linear cases, which could impact the applicability of the algorithm in more complex settings.

3. The paper addresses both the contextual bandit and bandits with dependent arms. However, this combined focus makes it challenging to isolate the specific impact of using a diffusion model. It remains unclear whether the diffusion model could be effectively applied in the traditional contextual bandit problem without dependent arms, as dependent arms generally allow for more informative decision-making.

4. Lastly, I am curious whether a diffusion model is truly necessary in this setting, especially if it is already pre-trained, to handle the contextual bandit scenario with dependent arms.

**Questions:**

I would refer to the weaknesses section. Any clarification or illustration would be very helpful.

---

> ### Author Response · Authors · 2024-11-22
>
> We thank the reviewer for their thoughtful feedback and recognition of the novelty, clarity, and rigor of our work. Below, we address the key concerns.
>
> ---
>
> ### Concern 1: When the true prior is not a diffusion model
>
> **Reviewer’s Comment:**
> The reviewer questions whether the algorithm performs effectively when the true prior is not a diffusion model.
>
> **Response:**
> dTS performs strongly even when the true prior is not a diffusion model. As shown in our experiments (Section 5, Figure 4), dTS consistently outperforms LinTS although the pre-trained diffusion model is only an apprixmation of the true prior distribution which is not a diffusion model. Specifically, Figures 4(a, b) show that the regret ratio of LinTS to dTS exceeds 1 across all tested diffusion model depths ($L \in [2, 100]$) and pre-training sample sizes ($[10, 50000]$), and in all these scenarios the true prior distribution is not a diffusion model. Similarly, dTS outperforms LinTS in the MovieLens experiments, where the true prior is also not a diffusion model.
>
> ---
>
> ### Concern 2: Theoretical analysis limited to the linear case
>
> **Reviewer’s Comment:**
> The theoretical statements are valid only for linear link functions, and it is unclear whether performance gaps between linear and non-linear link functions exist, potentially impacting applicability.
>
> **Response:**
> The linear case provides a tractable framework for establishing theoretical guarantees. Extending the analysis to non-linear link functions is challenging and remains future work. However, our empirical results (Section 5) demonstrate that dTS performs strongly in non-linear settings, consistently outperforming baselines, which supports its broader applicability.
>
> ---
>
> ### Concern 3: The combined focus on contextual bandits and dependent arms
>
> **Reviewer’s Comment:**
> The reviewer finds it challenging to isolate the impact of the diffusion model due to the combined focus on contextual bandits and bandits with dependent arms.
>
> **Response:**
> Theoretically, we show improvements in settings with dependent arms, as our analysis assumes linear link functions and leverages these dependencies. In practice, however, diffusion models are beneficial even when arms are independent, as they capture the complex distribution of action parameters.
>
> ---
>
> ### Concern 4: The necessity of using a diffusion model
>
> **Reviewer’s Comment:**
> The reviewer questions whether a diffusion model is necessary, particularly when it is already pre-trained, for handling dependent arms in contextual bandits.
>
> **Response:**
> The diffusion model is essential for capturing complex action parameter distributions that standard priors (e.g., Gaussians) cannot handle. While the pre-trained diffusion model serves as a strong starting point, combining it with Thompson Sampling (TS) enables adaptive exploration of high-reward regions. Efficient posterior updates and sampling fine-tunes the diffusion model, leading to better decision-making. This integration is critical; without it, performance deteriorates, as evidenced by dTS’s initial increase in cumulative regret (when relying solely on the pre-trained diffusion model). Over time, cumulative regret stabilizes and converges as TS updates the model.
>
> ---
>
> We’re grateful for the detailed feedback, which allowed us to strengthen our work through additional clarifications and experiments. If we have adequately addressed the reviewer’s concerns, we kindly request a re-evaluation of our work and a potential increase in score/confidence. For any unresolved issues, we’re happy to provide further clarifications. Thank you very much!

---

> > ### Author Response · Authors · 2024-12-02
> >
> > Dear Reviewer QyxM,
> >
> > We would be very grateful to hear your thoughts on our response. We are happy to provide further clarifications or details to address any remaining concerns. Thank you again for your thoughtful involvement.

---

### Official Review · Reviewer_KvQk · 2024-11-02

**Soundness:** 2
**Presentation:** 2
**Contribution:** 2
**Rating:** 5
**Confidence:** 3

**Summary:**

This paper studies contextual bandits where the feature vectors of the actions are hidden and generated by a known diffusion model. The authors propose an algorithm based on posterior sampling and show that the posterior update can be performed efficiently, using an approximation inspired by the linear case. The authors also theoretically demonstrate that the algorithm achieves a sublinear regret bound when the link functions are linear. Lastly, they present experiments to illustrate the practical performance of the algorithm.

**Strengths:**

1. Combining bandit algorithms with diffusion models is an innovative approach.

2. The paper introduces an efficient approximation for posterior updates in diffusion models, which I found interesting. The provided experiments offer valuable insights into the performance of this approximation.

**Weaknesses:**

1. I am unsure if the setting makes sense. Specifically, since $\theta_{\*,i}$ are fixed across episodes, one can interpret their concatenation as the hidden vector. This reduces the problem to a generalized linear contextual bandit with dimension $dK$, where the feature vector for each action $i \in [K]$ is $[0, \dots, 0, X_t, 0, \dots]$, activating only the entries corresponding to $\theta_{\*,i}$. According to my calculations, this should result in an algorithm with regret $\tilde{O}(\sqrt{dKT})$. Therefore, I question the advantage of introducing the posterior sampling over diffusion model in this context.

2. The theoretical analysis assumes linear link functions. However, since a combination of linear functions remains linear, the diffusion model may effectively reduce to a single linear function, potentially trivializing the problem.

3. Certain parts of the paper are unclear. For instance, it is not specified whether the covariance matrix $\Sigma_l$ or the link function $g$ are revealed to the agent.

**Questions:**

Please refer to the Weaknesses section above.

---

> ### Author Response · Authors · 2024-11-22
>
> We thank the reviewer for their detailed feedback and for highlighting the novelty and practical insights of our work. Below, we address the key concerns.
>
> ---
>
> ### Concern 1: Interpreting the setting as a generalized linear contextual bandit
>
> **Reviewer’s Comment:**
> The reviewer suggests that concatenating action parameters reduces the problem to a generalized linear contextual bandit of dimension $Kd$, with regret $\tilde{O}(\sqrt{Kdn})$.
>
> **Response:**
> This observation overlooks the critical aspect of computational efficiency. As discussed in Section 4.1, treating the problem this way results in computational complexity $\mathcal{O}(K^3d^3)$ and spatial complexity $\mathcal{O}(K^2d^2)$, which are prohibitively expensive for large $K$. In contrast, dTS achieves $\mathcal{O}((K+L)d^3)$ and $\mathcal{O}((K+L)d^2)$, where $L$ is the diffusion model depth. Since $K \gg L$ in practice, dTS offers significantly better scalability and efficiency.
>
> Beyond computational advantages, dTS leverages the rich prior distribution captured by (non-linear) diffusion models, resulting in improved performance over standard generalized linear contextual bandit algorithms, as demonstrated in our experiments.
>
> ---
>
> ### Concern 2: Theoretical analysis assuming linear link functions
>
> **Reviewer’s Comment:**
> The reviewer posits that linear link functions reduce the problem to a trivial case, as combinations of linear functions remain linear.
>
> **Response:**
> We agree that linear link functions simplify the analysis. However, even in the linear case, dTS outperforms standard LinTS (see Section 4.1). For linear link functions, similar results to “dTS with any $L$” can be achieved using “dTS with $L=1$” by marginalizing all latent parameters $\psi_\ell$ for $\ell \neq 1$ (except $\psi_1$). This marginalization, however, is only feasible in the linear case and becomes intractable for non-linear link functions.
>
> We stress however that Section 4 only complements the paper by analyzing a simplified instance of dTS to provide theoretical insights. The broader contributions of the paper extend beyond this setting. While extending the theoretical analysis to non-linear link functions is challenging and remains future work, dTS is designed to handle any (non-linear) link function. Our empirical results confirm its superior performance over baselines in both linear and non-linear link functions.
>
> ---
>
> ### Concern 3: Clarity on dTS inputs
>
> **Reviewer’s Comment:**
> The reviewer found certain parts unclear, such as whether the covariance matrices $\Sigma_\ell$ or the link function $f_\ell$ are revealed to the agent.
>
> **Response:**
> Algorithm 1 specifies that both the covariance $\Sigma_\ell$ and link function $f_\ell$ are provided as inputs and known to the agent. This is also clarified in the text (starting from line 090). This assumption is mild, as these parameters are typically accessible in pre-trained diffusion models.
>
> ---
>
> We’re grateful for the detailed feedback, which allowed us to strengthen our work through additional clarifications and experiments. If we have adequately addressed the reviewer’s concerns, we kindly request a re-evaluation of our work and a potential increase in score/confidence. For any unresolved issues, we’re happy to provide further clarifications. Thank you very much!

---

> > ### Comment · Reviewer_KvQk · 2024-11-25
> >
> > Thank you for your clarification.
> >
> > However, there is one aspect regarding efficiency that you are missing. When treating the instance as a large linear bandit, the covariance matrix is a matrix of dimension $dK$, with $K$ blocks of size $d \times d$ along the diagonal. Therefore, the time complexity of computing the inverse (or performing matrix multiplication) is $Kd^3$ rather than $K^3d^3$, while the space complexity is $Kd^2$.
> >
> > Furthermore, this algorithm should provide a worst-case regret bound of the same order in general, rather than the Bayesian regret bound under the linearity assumption as presented in the paper. Therefore, I still cannot see the necessity of introducing a diffusion model to this setting.

---

> > > ### Author Response · Authors · 2024-11-25
> > >
> > > We thank the reviewer for the follow-up questions and for seeking further clarification. However, we respectfully disagree. As explained in Section 4.1 (first paragraph), there are two possible versions of linear Thompson sampling. The first version maintains a joint posterior over $(\theta_i)_{i \in [K]} \mid H_t$. However, this approach is computationally inefficient because the covariance matrix of the joint distribution is full (non-diagonal), reflecting correlations between action parameters $\theta_i$ for $i \in [K]$. Thus we disagree that in this case, the covariance will be diagonal, it's the opposite due to action parameter correlations.
> > >
> > > The second version maintains $K$ individual posteriors, each corresponding to $\theta_i \mid H_t$. While this version is computationally efficient and comparable to dTS (with dTS being slightly more expensive due to the additional $L$ layers), it aligns with the implementation of LinTS in our experiments. Importantly, dTS consistently outperforms LinTS because dTS captures correlations between action parameters, whereas LinTS does not. Additionally, dTS has a tighter Bayesian regret bound than LinTS, as demonstrated in Section 4.1 and Appendix E.4 (Figure 7(b)).
> > >
> > > While we agree that LinTS and dTS have similar asymptotic worst-case bounds, dTS’s bound is significantly tighter when constants are considered, as shown in Appendix E.4 (Figure 7(a)).
> > >
> > > We hope this response addresses your questions and clarifies our results. Please let us know if further clarifications are needed. Thank you for your feedback and re-iteration! We greatly appreciate it.

---

> > > > ### Author Response · Authors · 2024-12-02
> > > >
> > > > Reviewer KvQk,
> > > >
> > > > We hope our response regarding the computational complexity of the proposed alternative was clear. Specifically, the high computational cost makes the proposed alternative infeasible in practice (see our response above).
> > > >
> > > > We just realized we did not answer your question regarding the derivation of frequentist (worst-case) regret. In fact, it is common to analyze _Thompson Sampling (TS) with informative priors_ in the Bayesian setting, as Bayes regret captures the benefits of using informative priors. In contrast, frequentist regret does not account for these benefits. Prior studies that derived Bayes regret typically did not derive frequentist regret, as it is standard practice to focus on one or the other. Both metrics provide valuable insights: frequentist regret reflects worst-case performance, while Bayes regret evaluates average performance over many instances sampled from the prior.
> > > >
> > > > The benefits of using diffusion models are evident from both our experimental results and the theoretical discussion in Section 4.1. The experimental section is extensive and clearly demonstrates the advantages, with dTS outperforming baselines in various scenarios.
> > > >
> > > > We would be very grateful to hear your thoughts on this. We are happy to provide further clarifications or details to address any remaining concerns. Thank you again for your thoughtful involvement.

---

> > > > > ### Comment · Reviewer_KvQk · 2024-12-02
> > > > >
> > > > > Given the reduction to linear bandits and the corresponding sparsity in the covariance matrix, the problem can be solved using the LinUCB algorithm in $\mathcal{O}(Kd^3)$ time with a regret bound of $\mathcal{O}(\sqrt{dKT})$. Thus, the proposed algorithm does not offer a significant advantage compared to this straightforward approach.
> > > > >
> > > > > That said, I agree that the paper proposes a prior-sampling-based algorithm, which might not be a fair comparison to the UCB approach. However, I still have two main concerns:
> > > > >
> > > > > 1. The analysis is valid only for linear link functions, where the diffusion model collapses to a linear model. While the authors attempt to demonstrate potential advantages through experimental results, one could always combine deep learning with an existing problem to replace some linear mappings in the current algorithm. They can then present theoretical analysis in the linear case "to provide theoretical insights" and conduct experiments comparing their algorithm with existing non-deep-learning approaches to demonstrate performance. I do not find this compelling, as it may not even constitute a fair comparison.
> > > > >
> > > > > 2. The theoretical results provide only a Bayesian regret. However, a worst-case regret bound for Thompson sampling-based algorithms is achievable. [Agrawal et al., 2013] establish a $\mathcal{O}(\sqrt{d^3T})$ worst-case regret bound for Thompson sampling on linear bandits. Therefore, I do not believe the theoretical results in this paper are robust.
> > > > >
> > > > > I acknowledge that the paper may contain some contributions, so I have decided to increase my score but still lean toward rejecting it.
> > > > >
> > > > > Agrawal, Shipra, and Navin Goyal. "Thompson sampling for contextual bandits with linear payoffs." International Conference on Machine Learning. PMLR, 2013.

---

### Official Review · Reviewer_jdND · 2024-11-04

**Soundness:** 3
**Presentation:** 3
**Contribution:** 2
**Rating:** 6
**Confidence:** 4

**Summary:**

The paper introduces Diffusion Thompson Sampling (dTS), an algorithm designed to handle the exploration-exploitation trade-off in contextual bandits with large action spaces. The key idea is to leverage pre-trained diffusion models to capture correlations between actions, which can guide more efficient exploration.

The authors propose using diffusion models as priors in a Thompson sampling framework, allowing the algorithm to explore the action space more strategically by exploiting the underlying structure of the action set. The paper provides a theoretical analysis of the algorithm, including an upper bound on the Bayes regret for linear diffusion and reward structures.

In addition to the theoretical results, the paper presents empirical evaluations that demonstrate the performance of the proposed method across various settings, showing that it outperforms traditional approaches in contexts with both linear and non-linear reward and diffusion models.

**Strengths:**

- The paper leverages *pre-trained diffusion models* to capture correlations in large action spaces, which provides a more structured and efficient approach to exploration compared to traditional methods in contextual bandits.

- The paper provides a Bayes regret bound for diffusion Thompson sampling (dTS) in the linear case, offering some theoretical backing to the proposed algorithm, though the assumptions could be more thoroughly justified (see the Weakness section).

- Comprehensive Empirical Results: The paper presents empirical evaluations that show dTS outperforming standard methods in several settings, particularly in large-scale action spaces, which lends support to the algorithm’s practical effectiveness.

**Weaknesses:**

- The unclear presentation of the assumptions, along with the lack of thorough discussion or justification, makes it difficult to fully evaluate the soundness of the work (see more details below).
- In terms of numbering, the assumptions should begin from **1** instead of 0. For example, (A0), (A1), (A2) should be renamed to (A1), (A2), (A3), etc., for consistency and clarity.
- Assumption (A0) should be separated into two distinct parts:
  (i) The assumption that **$W_\ell$ is known** for all $\ell \in [L]$, and
  (ii) The assumption of **Gaussian reward noise**, which is stronger than the sub-Gaussian noise typically assumed in the contextual bandit literature.
- Regarding **Assumption (A2)**, I encourage the authors to justify why such an assumption is necessary (rather than simply referring to "milder assumptions"). This assumption is non-standard compared to existing literature, and further elaboration is needed to position the analysis on firm ground. Providing a more explicit explanation would allow future work to build upon these assumptions, even if they seem restrictive.
- A fundamental concern lies in how the **$O(\sqrt{d(K+L)T})$ Bayes regret** in Theorem 4.1 compares with the well-established **$O(\sqrt{KT})$ Bayes regret** for the non-contextual multi-armed bandit (as in Russo and Van Roy, 2014). The bound of $O(\sqrt{d(K+L)T})$ in Theorem 4.1 is worse than both the non-contextual result in Bayes regret and the **worst-case regret bound of $O(\sqrt{KT})$** from Agrawal and Goyal (2023b) without prior. Why should practitioners consider this method in the linear case when the theory suggests it’s more effective to treat arms independently and use the non-contextual MAB version of Thompson sampling?
- The section “**Why the bound increases with L?**” (Line 268) is confusing. The theoretical result in the paper shows that increasing $L$ worsens the upper bound, implying that in the linear case, one should use **$L=1$**. The current explanation seems contrary to the theoretical findings for the linear case, and the authors should clarify this discrepancy.
- A **table of notations** is necessary, at least at the beginning of the appendix, to improve clarity and assist readers in following the paper’s mathematical presentation.


### **Minor Comments**:

- **Line 1151-1152**: "This sparsity assumption is both a novel and a key technical contribution to our work."
  It is debatable whether adding a structural assumption qualifies as novel unless it offers new insights or introduces milder assumptions compared to existing literature.

**Questions:**

Theorem 4.1 presents a **$O(\sqrt{d(K+L)T})$ Bayes regret** bound. Compared to Russo and Van Roy’s (2014) **$O(d\sqrt{T})$ Bayes regret** for linear Thompson sampling, I am curious about how your bound handles the **$O(\sqrt{d})$** factor. While I understand that the additional **K** factor stems from a different setting (with shared parameters across arms in Russo and Van Roy’s work), I would appreciate further elaboration on how your method results in a **$O(\sqrt{d})$** regret bound. Could you explain this in more detail?

---

> ### Author Response · Authors · 2024-11-22
>
> We thank the reviewer for their detailed feedback and recognition of the novelty and practical effectiveness of our proposed algorithm. Below, we address the key concerns and suggestions.
>
> ---
>
> ### Concern 1: Presentation and justification of assumptions
>
> **Reviewer’s Comment:**
> The presentation of assumptions is unclear, and certain assumptions (e.g., (A0) and (A2)) require better justification. The numbering should start from 1 instead of 0, and Assumption (A0) should be separated into two distinct parts.
>
> **Response:**
> We appreciate this suggestion and have revised the manuscript accordingly. Assumptions now start from (A1), and Assumption (A1) (previously (A0)) has been separated into two parts for clarity. Additionally, Assumption (A2) is now explicitly justified: when a pre-trained diffusion model (with linear link functions) is used as a prior, (A2) is automatically satisfied. This has been clarified in the revised manuscript.
>
> ---
>
> ### Concern 2: Comparison of Bayes regret bounds with prior work
>
> **Reviewer’s Comment:**
> The Bayes regret bound in Theorem 4.1 compared to existing bounds.
>
> **Response:**
> The regret bound in Theorem 4.1 is tighter than previous bounds. Specifically, Bayes regret bounds scale as $\mathcal{O}(\sqrt{dKn})$ for finite action spaces and $\mathcal{O}(d\sqrt{n})$ for infinite action spaces. This explains the $\mathcal{O}(\sqrt{d})$ improvement over Russo and Van Roy (2014), who assumed an infinite action space. But, Theorem 4.1 has smaller constants and provides a tighter bound compared to Russo and Van Roy (2014). It is also tighter than the frequentist bound for linear Thompson sampling (e.g., [1]). This is due to the fact that the constants are smaller in our case compared to prior studies. Moreover, in the non-contextual setting ($d=1$), our bound scales as $\mathcal{O}(\sqrt{K})$, matching the rate by Agrawal and Goyal (2023b) as reported in the review (although we could not identify the exact paper referenced). In general, Bayes regret bounds are inherently tighter than frequentist bounds, which aligns with the expectation that Theorem 4.1 outperforms worst-case bounds.
>
> ---
>
> ### Concern 3: Impact of $L$
>
> **Reviewer’s Comment:**
> The section on “Why the bound increases with $L$?” (Line 268) is unclear, and the explanation seems contradictory to the theoretical results.
>
> **Response:**
>
> The theoretical results predict that regret increases with $L$ when the true prior distribution matches a diffusion model of depth $L$, as increasing $L$ reflects a more complex action parameter distribution and hence a more complex bandit problem. This behavior is captured by our regret bound.
>
> In practice, when $L$ is small, the pre-trained diffusion model may be too simple to capture the true prior distribution, violating the assumptions of our theory. Increasing $L$ improves the pre-trained model’s quality, reducing regret. Once $L$ is large enough and the pre-trained model adequately captures the true prior distribution, the theoretical assumptions hold, and regret begins to increase with $L$, as predicted. This is validated empirically in Figure 4(b).
>
> ---
>
> ### Concern 4: Table of notations
>
> **Reviewer’s Comment:**
> A table of notations is necessary to improve clarity.
>
> **Response:**
> We have added a table of notations at the beginning of the appendix to enhance readability and assist readers in following the mathematical presentation.
>
> ---
>
> ### Concern 5: Novelty of sparsity assumption
>
> **Reviewer’s Comment:**
> It is debatable whether adding a sparsity assumption qualifies as novel unless it introduces new insights or milder conditions.
>
> **Response:**
> We agree that sparsity assumptions are not inherently novel unless they provide new insights. Our insight is that regret scales only with $d_\ell$ (the non-sparse dimensions of the mixing matrices) rather than the full dimension $d$. We have rephrased the relevant statement in the revised manuscript to emphasize this contribution.
>
> ---
>
> We’re grateful for the detailed feedback, which allowed us to strengthen our work through additional clarifications and experiments. If we have adequately addressed the reviewer’s concerns, we kindly request a re-evaluation of our work and a potential increase in score. For any unresolved issues, we’re happy to provide further clarifications. Thank you very much!
>
> [1] Abeille, Marc, and Alessandro Lazaric. "Linear thompson sampling revisited." Artificial Intelligence and Statistics. PMLR, 2017.

---

> > ### Comment · Reviewer_jdND · 2024-11-24
> >
> > Thank you for your response. I appreciate your efforts in addressing my concerns.
> >
> > Perhaps my previous questions were not sufficiently clear. I'd like to ask some clarifying questions to better understand your results:
> >
> > When $d = 1$, isn't your Bayes regret bound still $\tilde{O}(\sqrt{(K + L)n})$? This appears to be worse than the standard $\tilde{O}(\sqrt{Kn})$ regret bound for non-contextual multi-armed bandits, such as those presented in Agrawal and Goyal (2013b), which you cited in your paper. There is a well-known $\tilde{O}(\sqrt{Kn})$ worst-case regret bound (and Bayes regret of the same order) for such settings. Could you clarify how your bound compares in this case?
> >
> > When $d > 1$ but the context vectors are fixed over time (i.e., $X_t = X$ for all $t$), can't we treat each arm independently and apply standard MAB algorithms to achieve a regret of $\tilde{O}(\sqrt{Kn})$? In contrast, your approach seems to incur a regret of $\tilde{O}(\sqrt{d(K + L)n})$. Even if we set aside the dependence on $d$, along with the question in the previous paragraph, I'm more concerned about the additional $\sqrt{Ln}$ term in your bound. So, in this case, do you still have motivation to solve with your dTS?
> >
> > Regarding your comment that "increasing $L$ improves the pre-trained model's quality, reducing regret," along with your empirical results in Figure 4(b): Are you suggesting that there is a some sweet spot of $L$?
> > Should practitioners choose $L$ in practice to balance model quality and regret?
> >
> > I hope these questions help clarify my concerns. I look forward to your detailed response.

---

> ### Author Response · Authors · 2024-11-25
>
> We thank the reviewer for the follow-up questions and for seeking further clarification. Below, we address each question in detail.
>
> ---
>
> ### Question 1: Regret bound comparison in the case $d = 1$
>
> **Reviewer’s Question:**
> When $d = 1$, isn't your Bayes regret bound still $\mathcal{O}(\sqrt{(K+L)n})$? This appears worse than the standard $\mathcal{O}(\sqrt{Kn})$ bound for non-contextual multi-armed bandits (MABs), such as those presented in Agrawal and Goyal (2013b).
>
> **Response:**
> For $d=1$, our Bayes regret bound is $\mathcal{O}((K+L)n)$, but constants are ignored in this form. In Bayes regret, constants are crucial as they showcase the benefits of using informative priors (e.g., small $\sigma_\ell^2$ in our case). It is hard to compare tightness theoretically, but we can plot the bounds in their explicit expressions. Agrawal and Goyal (2013b) do not provide explicit constants in their results. Therefore, we compared our bound to existing frequentist bounds for Thompson sampling in linear bandits (e.g., [1]) in Appendix E.4 (Figure 7(a)) and found that our bound is much tighter. Similar behavior is expected for the MAB setting. Furthermore, our bound is tighter than standard Bayes regret bounds for LinTS, as illustrated in Appendix E.4 (Figure 7(b)). These comparisons highlight the advantages of incorporating structured priors into dTS.
>
> **Important Note:** While these comparisons are insightful, regret bounds derived for linear bandits are generally less tight when applied to MAB settings compared to bounds specifically derived for MABs. This applies to Question 2 as well.
>
> ---
>
> ### Question 2: Fixed contexts over time and motivation for dTS
>
> **Reviewer’s Question:**
> When $d > 1$ but the context vectors are fixed over time, can't we treat each arm independently and apply standard MAB algorithms to achieve $\mathcal{O}(\sqrt{Kn})$ regret? In contrast, your approach seems to incur $\mathcal{O}(\sqrt{(K+L)dn})$ regret. Is there still motivation to use dTS in this case?
>
> **Response:**
> We agree that in this specific setting, the $d$ term remains in our bound. This occurs because our regret bounds are derived for linear bandits, and translating them to fixed-context scenarios does not eliminate the context dimension $d$. But this is true for regret bounds in the literature that were derived in linear bandits; translating these to the specific fixed-context case doesn’t make the context dimension $d$ disappear
>
> However, by adapting our proof techniques for the MAB setting, we can derive bounds independent of $d$. In such cases, posterior variances replace posterior covariances in the analysis, effectively removing the dependence on $d$. We did not do this since we focused on generalized linear bandits.
>
> For both questions Q1 and Q2, in practice, using dTS is more beneficial in high-dimensional problems with complex dependencies. Thus, in simple MAB problems, one might use simpler models.
>
> ---
>
> ### Question 3: Sweet spot for $L$ and practical recommendations
>
> **Reviewer’s Question:**
> Regarding your comment that "increasing $L$ improves the pre-trained model's quality, reducing regret," and the empirical results in Figure 4(b): Is there a sweet spot for $L$, and how should practitioners choose $L$?
>
> **Response:**
> Yes, as shown in Figure 4(b), performance improves with $L$ until approximately $L=40$, after which it declines. The optimal value of $L$ depends on the application. Practitioners can empirically tune $L$ by selecting the smallest value that accurately captures the offline data distribution. This tuning process can be performed offline before integrating the diffusion model prior into dTS, ensuring both computational efficiency and improved performance.
>
> ---
>
> We hope this response addresses your questions and clarifies our results. Please let us know if further clarifications are needed. Thank you for your thoughtful feedback and prompt response! We greatly appreciate it.
>
> [1] Abeille, Marc, and Alessandro Lazaric. "Linear thompson sampling revisited." Artificial Intelligence and Statistics. PMLR, 2017.

---

> ### Comment · Reviewer_jdND · 2024-11-30
>
> Thank you for your response.
>
> However, I feel that my primary concern remains unresolved. As I stated in my earlier comment: *“Even if we set aside the dependence on $d$, along with the question in the previous paragraph, I’m more **concerned about the additional $\sqrt{Ln}$ term in your bound**”* My main question was and is about the extra $\sqrt{Ln}$ dependence in your proposed method’s regret bound.
>
> From your response, aside from the acknowledgment that *“in simple MAB problems, one might use simpler models,”* it is unclear how the additional dependence on $L$ aligns with the practical utility of dTS in such settings. The cases with $d = 1$ and $d > 1$ with fixed contexts—both of which are special cases of your problem—highlight that your method incurs sub-optimal regret due to the additional $L$-dependent term. Simply stating that *“using dTS is more beneficial in high-dimensional problems with complex dependencies”* feels vague and lacks specificity. It is critical to clearly define the regimes where your method is advantageous and where it is not.
>
> I acknowledge that as $L$ increases in the prior, the problem inherently becomes more complex, and the regret bound naturally increases. However, your argument in the section *“Why the bound increases with $L$?”*—and even in your response above—that there is an optimal $L$ in practice does not align with your theoretical result, where the regret monotonically increases with $L$. This statement is confusing and risks misrepresenting the theoretical findings. I strongly suggest removing or rephrasing the statement, *“Increasing $L$ improves the pre-trained model’s quality, reducing regret,”* as your definition of Bayes regret is defined with respect to a known prior (hence "Increasing $L$ ... reducing regret" does not make sense). If you wish to discuss the practical behavior of $L$ (when $L$ is not known?), such discussion belongs in the experimental section and should not be conflated with the theoretical analysis.
>
> Finally, regarding practical applications, let’s consider a scenario where you are solving a contextual bandit problem with $K$ parameters from scratch, with no pre-existing data (a typical bandit setting). How would you recommend choosing $L$ in such a case? In many practical scenarios, your exact prior or $L$ may not be known in advance.
>
> I look forward to further clarifications.

---

> ### Author Response · Authors · 2024-11-30
>
> We would like to thank you for engaging in this discussion; we greatly appreciate your thoughtful feedback and the opportunity to clarify our approach.
>
> ### **(I) Dependence on $L$ Does Not Make Bayes Regret Suboptimal**
>
> ---
>
> #### **1) When Comparing Bayes Regret Bounds with Each Other**
>
> Let us start by clarifying that Bayes regret should be understood as follows: when the true prior is $P_0$, the Bayes regret is $\mathcal{O}(f(n, K, d, P_0))$, where the bound $f(n, K, d, P_0)$ depends on the horizon $n$, problem parameters $K, d$, and the true prior $P_0$ (more precisely, it depends on the variance of the true prior $P_0$). Therefore, when the true prior $P_0$ has high variance, the problem becomes more challenging, and the bound increases naturally. The $L$ term is directly related to the variance of the diffusion model prior, and this is why it appears. However, this is not problematic, as we explain next.
>
> For dTS, the true prior $P_0$ is a diffusion model, and the regret bound is:
>
> $$
> \mathcal{O}\left(\sqrt{nd \left(K \sigma_1^2 +  \sum_{\ell=1}^L \sigma_{\ell+1}^2\right)}\right),
> $$
>
> where we have dropped dependence on $\sigma_{max}$ to simplify the explanation. Indeed, we observe an additional $L$ term, but this arises because the overall variance of this prior is roughly $\sum_{\ell=1}^{L+1} \sigma_{\ell+1}^2$.
>
> For LinTS with a Gaussian prior that marginalizes over all latent parameters, the prior variance is roughly $\sum_{\ell=1}^{L+1} \sigma_\ell^2$, and the Bayes regret becomes:
>
> $$
> \mathcal{O}\left(\sqrt{ndK \sum_{\ell=1}^{L+1} \sigma_\ell^2}\right).
> $$
>
> Hence, the $L$ term also appears in the LinTS regret bound. In fact, dTS has a better dependency on $L$ because it only multiplies $K$ by $\sigma_1^2$, whereas in LinTS, $K$ is multiplied by the much higher constant $\sum_{\ell=1}^{L+1} \sigma_\ell^2$.
>
> The $L$ dependency is thus a consequence of the prior variance in a diffusion model and will also appear for LinTS Bayes bounds.
>
> Note that comparing Bayes regret bounds with different prior variances is not meaningful; for example, comparing LinTS with a small prior variance $v$ to dTS with a much larger prior variance is not meaningful. This is because we could reduce the variances in the diffusion model prior so that their sum $\sum_{\ell=0}^L \sigma_\ell^2$ is smaller than the variance used in LinTS. Actually, the dependence on $L$ can disappear with proper choices of $\sigma_\ell$. This is why to compare Bayes regret bounds, we compare methods whose true priors have the same variance (e.g., we compare methods in the same problem difficulty). A rigorous comparison of our bound with that of LinTS is given in Section 4.1.
>
> ---
>
> #### **2) When Comparing Bayes Regret Bounds with Frequentist Bounds**
>
> First, we do not believe it is rigorous to compare Bayes regret bounds with frequentist ones, as they operate in very different settings, and it is common practice not to compare them.
>
> However, if we make such a comparison and include $L$ in it, then all other constants (including variances) must also be included for a rigorous analysis. When this is done, our bound is much tighter (Appendix E.4). Of course, if we only care about asymptotic behavior and include $L$, the presence of $L$ may make it appear as if the bound is loose. However, as shown in our results (Appendix E.4), our bound is significantly smaller than the frequentist bound for LinTS and the Bayes regret bound for LinTS.
>
> ---
>
> ### **(II) Practical Use of Diffusion Models**
>
> If no offline data or pre-trained diffusion model is available, simpler models may be more appropriate. We will clarify that dTS relies on a pre-trained diffusion model, either from an existing source or trained on offline data (as done in our MovieLens experiment). However, this is not a strong assumption. In practical applications, offline data is often available, enabling us to pre-train a diffusion model. There are also numerous pre-trained diffusion models available for domains similar to the bandit problem (e.g., pre-trained models on product images for recommendation problems). The choice of $L$ depends on the problem, and we select the smallest $L$ that achieves the best empirical results on the offline data. This process is independent of our work and follows standard guidelines for pre-training diffusion models.
>
> To address potential confusion, we will move the discussion of the practical behavior of $L$ to the experiments section. Many thanks for this suggestion.
>
> ---
>
> We have attempted to explain this from first principles regarding how Bayes regret bounds should be compared (and that we should account for constants in our comparisons). We hope this explanation is clear, and we are keen to elaborate further if needed. Thank you again for engaging in this discussion; we greatly appreciate it.

---

> ### Comment · Reviewer_jdND · 2024-11-30
>
> Thank you for your response. However, I believe there is a misunderstanding regarding my original question, as evident in the authors’ response **(I) Dependence on $L$ Does Not Make Bayes Regret Suboptimal**. My concern was never about the comparison between LinTS and your method (or frequentist vs. Bayes regrets). Instead, my focus was on the apparent sub-optimality of your proposed method in special cases, such as $d = 1$ or $d > 1$ with fixed context.
>
> To clarify, throughout the discussion, I emphasized that in special cases (e.g., $d = 1$ or $d > 1$ with fixed context), the problem appears to admit a $\sqrt{Kn}$ **Bayes regret** bound when solved as a multi-armed bandit (MAB), while your proposed solution incurs a $\sqrt{(K+L)n}$ regret bound for any $L \geq 1$. This creates a gap in such cases, which is the sub-optimality I referred to—not the gap between LinTS and your method (or not the difference between frequentist and Bayes regrets).
>
> To further elaborate, consider a case when $d = 3$ with $\mathcal{X} = $ {$(1,2,0), (0,1,2)$}. Suppose the nature selects $X_t = (1,2,0)$ for odd $t$ and $X_t = (0,1,2)$ for even $t$ (this selection of nature is unknown to the algorithm a priori, but $X_t$ is revealed at each time). Given the diffusion prior as defined in your method, this problem setting satisfies your problem set-up.
> For this, one can treat this as two distinct instances of MABs: when $X_t = (1,2,0)$ it is one instance, and when $X_t = (0,1,2)$ it is the other instance. Each instance can theoretically be solved with a $O(\sqrt{Kn})$ Bayes regret bound (with time horizon effectively halved for either instance, which is just a constant factor, also suppose I do not care about the dependence on $d$ here). This approach avoids any dependence on $L$ whereas your method will still have $L$ dependence. I would like to clarify whether this reasoning is incorrect. Simple “Yes” or “No” would be fine.
>
> ---
>
> **[Minor comments, not directly related to the primary concern]**
> Regarding the comparison between Bayes regret bounds and frequentist bounds, I want to clearly clarify my statement about Theorem 4.1. When I noted that the regret bound is “worse than both the non-contextual result in Bayes regret and the worst-case regret bound,” I meant that Theorem 4.1 is sub-optimal even in terms of Bayes regret.
> The reason I mentioned the frequentist regret bound was to highlight that Bayes regret is typically expected to be sharper than (or at most equal to) the frequentist regret bound for the same problem (same problem but the regret definition differ by the expectation over prior), given the weaker nature of Bayes regret. However, my primary concern remains the sub-optimality of Theorem 4.1 in terms of Bayes regret itself, particularly in the scenarios outlined above. You do not need to answer this part. This is just my clarification, just so that the authors understand the point.
>
> I hope this clarification resolves any misunderstanding and look forward to your thoughts on these points.

---

> > ### Author Response · Authors · 2024-11-30
> >
> > Thanks for your further questions. Below is our response.
> >
> > ### On the Role of $L$
> >
> > Even in the aforementioned special cases, including the diffusion depth $L$ in the comparison while discarding all constants in the standard $\mathcal{O}(\sqrt{Kn})$ bound is not meaningful. $L$ is a constant, and we can easily remove it from our bound by choosing small variances $\sigma_\ell^2$. Additionally, the standard $\mathcal{O}(\sqrt{Kn})$ bound depends on the specific algorithm used. For example:
> >
> > - If the comparison is with a Thompson Sampling (TS) algorithm for Multi-Armed Bandits (MAB), the prior variance will appear in the bound. A similar comparison to what we made with LinTS shows that an $L$ is also embedded in the standard $\mathcal{O}(\sqrt{Kn})$ bound of TS for MAB. Specifically, for a TS algorithm with prior variance $\sum_{\ell=1}^{L+1} \sigma_\ell^2$, the Bayes regret will explicitly contain $L$ because it is part of the prior variance. This assumption is natural since we compare to dTS, whose prior variance is $\sum_{\ell=1}^{L+1} \sigma_\ell^2$. As explained, a fair comparison in Bayes regret requires the prior variances of the algorithms to be the same. For this reason, we must clarify "the standard $\mathcal{O}(\sqrt{Kn})$." When this bound refers to TS for MAB, our bound remains tighter as the dependency on $L$ is hidden in the standard bound as well.
> >
> > We emphasize that $L$ is a constant tied to the prior variance. If we focus on $L$ in our discussion, we must also account for the prior variance used by the algorithm we compare dTS to. For algorithms without priors, the corresponding bound (frequentist) will contain much larger, hidden constants. Thus, we should be cautious when discussing $L$ without considering other constants.
> >
> > For the simplified setting you proposed:
> > - **Yes**, if the MAB algorithm used does not incorporate a prior variance scaling with $L$, but this comparison would be unfair to dTS (as explained above, we should compare algorithms with the same prior variances (aka same problem difficulty)).
> > - **No**, if the MAB setting uses TS with a prior variance of $\sum_{\ell=1}^{L+1} \sigma_\ell^2$, as this ensures a fairer comparison given the same prior variance and problem difficulty.
> >
> > ---
> >
> > ### Additional Comments
> >
> > 1. **Applicability of Bounds in MAB vs. Linear Bandits**:
> >    Our bound was derived for linear bandits. It is well-known that applying linear bandit bounds to the MAB setting might result in loose bounds compared to those specifically derived for MAB settings. However, our theory can be readily adapted to MAB.
> >
> > 2. **Focus and Scope of the Paper**:
> >    Beyond the dependency on $L$, the focus and scope of the paper extend far beyond simplified settings. Our work primarily addresses complex generalized linear bandit settings, provides efficient approximations, and emphasizes high-dimensional (real-world) problems. By high-dimensional, we refer to scenarios with large $d$ and $K$. While simplified settings are interesting, they should not overshadow the broader scope of our work, which focuses on more challenging, high-dimensional problems.
> >
> > We hope this explanation is clear, and we are keen to elaborate further if needed. Thank you again for engaging in this discussion; we greatly appreciate it.

---

> > > ### Comment · Reviewer_jdND · 2024-12-01
> > >
> > > Thank you very much for your answers. I will raise my score.

---

### Official Review · Reviewer_ze97 · 2024-11-04

**Soundness:** 3
**Presentation:** 3
**Contribution:** 2
**Rating:** 6
**Confidence:** 4

**Summary:**

This paper studies the performance of Thompson Sampling for contextual bandit problems with generalized linear model reward distribution. Starting from a prior distribution over the "action-parameter", Thompson Sampling algorithm works by randomly selecting actions according to their posterior probability of being optimal. More specifically, at each time step, it samples an "action-parameter" estimate from the posterior distribution conditioned on the history and selects the action that is optimal for the sampled parameter estimate given the context.

In this work, the authors study the case where the prior is (or can be approximated by) a diffusion model. This idea is inspired by the work of Hsieh et al. (2023), where they showed they could use diffusion model to model the prior together with Thompson Sampling algorithm to solve bandit problems. This paper builds on Hsieh results and extends their idea to contextual bandit problems. They also show how to efficiently compute an approximation of posterior and how to efficiently sample from this approximation, and call the resulting algorithm diffusion Thompson sampling. Under the assumption that the true prior can be written as a linear Gaussian system, their approximate posterior matches the exact posterior and the authors could derive Bayesian regret bounds in $\tilde{O}(\sqrt{n (d K \sigma_{1}^2 + \sum_{l=1}^L d \sigma_{l+1}^2 \sigma_{\text{MAX}}^{2l}})$ where $n$ is the number of time steps, $d$ is the dimension of the problem, $K$ is the number of possible actions, $\sigma$ is the variance of the rewards, $\sigma_1$ is the isotropic variance of the action parameter given the first latent variable, $L$ is the number of latent parameters and $\sigma_{l+1}$ is the isotropic variance of the latent parameter $l$ conditioned on the latent parameter $l+1$ (by construction the $L+1$ latent parameter is zero) and $\sigma^2_{\text{MAX}} = \max_{l\in[L+1]} 1+\frac{\sigma_l^2}{\sigma^2}$.

The authors perform two experiments to demonstrate the performance of the proposed method. The first experiment is performed on synthetic problems where the true prior is a diffusion model. They compare their method with several baselines (LinTS, LinUCB, HierTS, GLM-TS, UCB-GLM) and show that the diffusion Thompson sampling performs better. The second experiment has a prior distribution that is not a diffusion model. In this case, the author first pre-train a diffusion model to approximate the prior distribution before running the algorithm. They show that their method performs better than LinTS.

**Strengths:**

The main strength of the paper is its overall clarity and coherence. The authors introduce the interesting idea of extending the work of Hsieh et al. (2023) to contextual bandit problems, show how to efficiently compute an approximation of the posterior to sample from, derive regret bounds under some specific assumptions, and perform experiments to demonstrate the performance of the proposed method. The different ideas are explained clearly, the code for the experiments is provided and user-friendly, the notations used are rigorous, and the proof techniques are thorough and explicit.

**Weaknesses:**

Although the paper's main ideas are interesting, the experiment section presents some weaknesses. The first experiment, Figure 2, compares the proposed algorithm dTS against several baselines HierTS, LinTS, GLM-TS, UCB-GLM for problems where the true prior is a diffusion model. Unsurprisingly, this setting perfectly fits the proposed algorithm, and it outperforms the baseline algorithms. This first experiments can be understood as a sanity check test that dTS passed. A more interesting experiment, Figure 4, tests the performance of dTS on problems where the true prior is not a diffusion model. However, the performance of dTS in this setting is only compared to LinTS, which is by design not suited to this contextual bandit problem as it cannot capture the correlations among actions. It is, therefore, not surprising that dTS improves the performance of LinTS for this problem, and it is difficult to appreciate its performance. A more fair comparison would have been against algorithms suited to the setting, such as "Vits: Variational Inference Thompson Sampling". Another interesting experiment would have been to compare the performance of dTS against the DiffTS proposed by Hsieh et al. (2023) on bandit problems and verify if dTS can recover the same performance while presenting computational advantages.

**Questions:**

Here is a list of suggestions for the authors.
- In section 4.1, Statistical benefits, the authors mention "The only Bayesian lower bound that we know of is $\Omega(\log^2(n))$. The authors could have mentioned the minimax results from Dani et al. (2008) in $\Omega(d\sqrt{n})$ for the $d$-dimensional linear bandit setting.
- On line 354, the authors mention "we can show that dTS’s regret is independent of K in their setting, assuming the availability of $\phi$". It would be interesting to add this proof in the Appendix.
- As mentioned before, it would be interesting to compare the performance of dTS against fairer baselines for the experiments on MovieLens problem such as the Vits: Variational Inference Thompson Sampling" from Clavier et al. (2023).
- It would also be interesting to compare the performance of dTS against the DiffTS proposed by Hsieh et al. (2023) on bandit problems and verify if dTS can recover the same performance while presenting computational advantages.
- In the Appendix "Extended related work", the authors are pointed to two papers that studied the performance of TS for contextual bandits and derived Bayesian regret bounds: Neu et al. "Lifting the information ratio: An information-theoretic analysis of thompson sampling for contextual bandits" (2022) and Gouverneur et al. "Thompson sampling regret bounds for contextual bandits with sub-Gaussian rewards." (2023).
- For the sake of completeness, the authors are suggested to include in the Appendix the pseudocode of the LinTS algorithm used for the experiments.
- On line 239, the authors mention that their "bound is $\tilde{O}(\sqrt{n})$. A suggestion is to include the non-logarithmic dependencies on $d$ and $K$.
- On line 127, a suggestion to the authors to change "We design Thompson sampling that samples" to "We design a Thompson sampling algorithm that samples".
- On lines 122-123, the authors wrote "The Bayes regret is known to capture the benefits of using informative priors, and hence it is suitable for our problem". The authors are suggested to provide references to support their claim.

---

> ### Author Response · Authors · 2024-11-22
>
> We thank the reviewer for their detailed feedback, recognition of the clarity and coherence of our work, and thoughtful suggestions for improvement. Below, we address the main concerns and suggestions.
>
> ---
>
> ### Concern 1: Experimentation in scenarios with non-diffusion model priors
>
> **Reviewer’s Comment:**
> The performance of dTS in settings where the true prior is not a diffusion model is only compared to LinTS, which cannot capture correlations among actions. A fairer comparison would include algorithms like "Vits: Variational Inference Thompson Sampling" or DiffTS from Hsieh et al. (2023).
>
> **Response:**
> We did not include DiffTS as it focuses on the multi-armed bandit setting and is not directly applicable to the contextual bandit framework. This limitation is acknowledged in their conclusion, where they propose extending DiffTS to contextual bandits as a challenging future direction. As such, neither empirical nor theoretical comparisons are feasible, as DiffTS does not work in our setting and lacks a regret bound. We are currently running MovieLens experiments against  ViTS and will share results upon completion.
>
> ---
>
> ### Concern 2: Additional related work and references
>
> **Reviewer’s Comment:**
> The reviewer points out missing references.
>
> **Response:**
> We have cited Neu et al. (2022) and Gouverneur et al. (2023) and Dani et al. (2008) and included references to support the claim regarding the suitability of Bayes regret for evaluating informative priors.
>
> ---
>
> ### Concern 4: Minor revisions for clarity and consistency
>
> **Reviewer’s Comment:**
> The reviewer suggests minor edits for improved clarity.
>
> **Response:**
> We have incorporated all the suggested edits to improve clarity and consistency in the manuscript.
>
> ---
>
> We’re grateful for the detailed feedback, which allowed us to strengthen our work through additional clarifications and experiments. If we have adequately addressed the reviewer’s concerns, we kindly request a re-evaluation of our work and a potential increase in score. For any unresolved issues, we’re happy to provide further clarifications. Thank you very much!

---

> > ### Author Response · Authors · 2024-11-30
> > **Additional comparison with variational inference/Langevin dynamics**
> >
> > Dear Reviewer ze97,
> >
> > We would like to get back to you after we tried implementing Variational Inference Langevin dynamics to sample from our posterior.
> >
> > ---
> >
> > ### **Challenges with Variational Inference and Langevin Dynamics for Posterior Sampling with Diffusion Priors**
> >
> > We found both VI and Langevin dynamics extremely challenging to apply when the prior is a diffusion model (e.g., as we attempted with viTS [3]). Both approaches rely on the posterior being proportional to $e^{-V(\theta)}$, where $\nabla V(\theta)$ (the gradient of the potential) is assumed to be tractable. The difficulties stem from the nature of $V(\theta)$, which involves the log of an (intractable) integral over all latent parameters $\psi_1, \dots, \psi_L$ (see Eq. (3) in the paper). Computing $\nabla V(\theta)$ is non-trivial. Indeed, methods like [1, 2, 3], which successfully apply VI or Langevin dynamics for Thompson Sampling, rely on simpler priors (e.g., Gaussian) combined with non-linear likelihoods. The approximation was needed because of the non-linearity in the likelihood and not the prior. In contrast, our approach involves both a very complex prior (diffusion model) and a potentially non-linear likelihood, significantly increasing the difficulty.
> >
> > Even if $\nabla V(\theta)$ were tractable (which we could not find a feasible way to achieve), VI would require designing a variational family to match the posterior. Doing so for a diffusion-based prior would demand significant compromises in either expressivity or computational efficiency. Similarly, Langevin dynamics would require fine-tuning for convergence, which becomes tricky in high-dimensional settings with complex priors like ours.
> >
> > ---
> >
> > ### **Advantages of Our Approach**
> >
> > Given these challenges, our efficient approximation offers a practical and effective alternative:
> > - **No Hard Numerical Optimization:** It bypasses the need for tractable gradients or explicit variational families.
> > - **Preserves Posterior Expressiveness:** The updated posterior remains diffusion-based, ensuring no loss of expressivity.
> >
> > ---
> >
> > ### **Summary**
> >
> > In summary, while VI and Langevin dynamics are standard, their application here is severely limited by the intractability of $V(\theta)$ and its gradient. These constraints make our proposed efficient approximation both appealing and well-suited to this context, avoiding the aforementioned difficulties and providing a practical solution for posterior sampling.
> >
> > We hope this response clarifies our approach and highlights its novelty and feasibility.

---

> > > ### Comment · Reviewer_ze97 · 2024-12-01
> > >
> > > Thank you for your response and for addressing some of my comments. I still believe that including a comparison of the performance of dTS against VITS for the MovieLens experiments, as well as a comparison of dTS against DiffTS on (non-contextual) bandit problems, would improve the article. I will maintain my positive score.

---

> > > > ### Author Response · Authors · 2024-12-02
> > > >
> > > > We greatly appreciate your consideration of our rebuttal. One final clarification:
> > > >
> > > > Comparing to viTS in its current form is infeasible when the prior is a diffusion model because it relies on the gradient of the potential function $V(\theta)$, which is intractable (as explained in our response "Additional comparison with variational inference/Langevin dynamics"). Including viTS as a baseline would require modifying viTS through new approximations of the potential function. However, deriving these approximations is neither trivial nor straightforward and constitutes a separate research direction. For this reason, we cannot consider viTS as a baseline in our experiments when the prior is modeled as a diffusion model. A comparison with DiffTS is feasible in the non-contextual case, but we do not focus on the non-contextual setting. We will, however, include it in the appendix of the later version of the paper for completeness.
> > > >
> > > > We are very grateful for your feedback and positive score.

---

### Official Review · Reviewer_eKpC · 2024-11-06

**Soundness:** 3
**Presentation:** 3
**Contribution:** 3
**Rating:** 6
**Confidence:** 4

**Summary:**

This paper introduces Diffusion Thompson Sampling (dTS), a novel algorithm leveraging pre-trained diffusion model priors to optimize exploration in contextual bandits. By capturing correlations among actions, dTS enhances both computational and statistical efficiency. It offers theoretical insights into posterior approximations and Bayes regret bounds, providing a structured and computationally manageable solution for large action spaces in contextual bandit problems.

**Strengths:**

1. The paper thoroughly explains the posterior approximation process for both linear and non-linear diffusion models.

2. The recursive hierarchical sampling in dTS is well-defined, which simplifies the complex diffusion model and supports efficient computational sampling.

**Weaknesses:**

1. The method’s performance is heavily reliant on the accuracy of the pre-trained diffusion model. If the model's prior assumptions are incorrect or misspecified, the effectiveness of the posterior approximations and subsequent regret bounds could be compromised.

2. There is limited discussion on alternative approximation techniques (e.g., variational inference or Monte Carlo methods) that could potentially offer more flexible or accurate approximations, especially for non-linear reward distributions.

**Questions:**

1. How does the choice of the link function $ f_\ell$ impact the posterior approximation and computational efficiency in non-linear scenarios?

2. How would the regret bounds change if the action parameters $\theta^*$ were updated using a non-linear transformation?

3. How does the choice of the number of layers $L$ in the diffusion model affect the convergence rate of the posterior approximations, especially as the action space $K$ increases?

4. What impact does the sparsity assumption on the mixing matrices $W_\ell$ have on the model's computational efficiency and the quality of the posterior approximations?


5. As I have mentioned in (3) of Weaknesses, can you discuss alternative approximation techniques (e.g. Monte Carlo methods [1,2]) that could potentially offer more flexible or accurate approximations, especially for non-linear reward distributions?

References:

[1] Xu, Pan, et al. "Langevin monte carlo for contextual bandits." International Conference on Machine Learning. PMLR, 2022.

[2] Karbasi, Amin, et al. "Langevin thompson sampling with logarithmic communication: bandits and reinforcement learning." International Conference on Machine Learning. PMLR, 2023.

---

> ### Author Response · Authors · 2024-11-22
>
> We thank the reviewer for their thoughtful feedback and recognition of the strengths and contributions of our work. Below, we address the key concerns and questions raised.
>
> ---
>
> ### Concern 1: Reliance on pre-trained diffusion models
>
> **Reviewer’s Comment:**
> The method’s performance depends heavily on the accuracy of the pre-trained diffusion model. Misspecified priors could compromise posterior approximations and regret bounds.
>
> **Response:**
> We agree that dTS relies on the quality of the pre-trained diffusion model. However, dTS mitigates this limitation by combining the diffusion model prior with Thompson Sampling, enabling adaptive updates based on observed rewards. As shown in our experiments, dTS outperforms baselines even when the prior is misspecified, leveraging observed data to refine decision-making.
>
> ---
>
> ### Concern 2: Limited discussion on alternative approximation techniques
>
> **Reviewer’s Comment:**
> There is limited discussion of alternative techniques, such as variational inference or Monte Carlo methods, that could offer more flexible or accurate approximations for non-linear reward distributions.
>
> **Response:**
> We appreciate this suggestion and acknowledge the potential of alternative techniques [1,2]. However, these methods often involve higher computational costs, making them less practical for large-scale action spaces. dTS focuses on efficient posterior updates and sampling. For linear rewards, our method provides a closed-form solution, while for non-linear rewards, only MLE optimization is required.
>
> In contrast, variational inference introduces additional bias by approximating the posterior as Gaussian, whereas dTS preserves the richer structure of diffusion models. Monte Carlo methods, while flexible, incur significant computational overhead, particularly in bandit settings. We are also running ViTS [3] on the MovieLens dataset and will share results upon completion.
>
> ---
>
> ### Response to Questions:
>
> 1. **Impact of the link function on posterior approximation and computational efficiency**
>    The complexity of evaluating the link function affects posterior approximations minimally. For linear link functions, approximations are exact. Quantifying the impact of non-linear link functions on approximation quality is difficult, but empirical results show strong performance, especially as non-linearity increases.
>
> 2. **Changes in regret bounds with non-linear transformations**
>    While our theoretical analysis focuses on linear link functions, regret bounds for non-linear transformations are challenging to derive. Empirical results demonstrate robust performance in non-linear cases, with greater gaps over baselines as non-linearity increases.
>
> 3. **Effect of the number of layers $L$ on convergence**
>    The theoretical results predict that regret increases with $L$ when the true prior matches a diffusion model of depth $L$, reflecting the increased complexity of the action parameter distribution. Empirically, regret initially decreases as $L$ increases due to improved pre-trained diffusion models. Once $L$ is sufficiently large and the pre-trained model captures the true prior, regret begins to increase with $L$, consistent with theoretical predictions (Figure 4b).
>
> 4. **Impact of sparsity assumptions**
>    Sparsity assumptions do not affect computational efficiency or posterior derivations. However, they reduce regret, as dTS’s regret depends only on the non-zero submatrix dimensions rather than the full matrix dimensions.
>
> ---
>
> We thank the reviewer for their detailed feedback, which has strengthened our work through additional clarifications and experiments. If our responses address your concerns, we kindly request a re-evaluation and a potential increase in score. We are happy to provide further clarifications if needed. Thank you!
>
> ---
>
> **References:**
>
> [1] Xu, Pan, et al. "Langevin Monte Carlo for contextual bandits." *International Conference on Machine Learning.* PMLR, 2022.
>
> [2] Karbasi, Amin, et al. "Langevin Thompson Sampling with logarithmic communication: Bandits and reinforcement learning." *International Conference on Machine Learning.* PMLR, 2023.
>
> [3] Clavier, Pierre, Tom Huix, and Alain Durmus. "VITS: Variational Inference Thompson Sampling for contextual bandits." arXiv preprint arXiv:2307.10167 (2023).

---

> ### Author Response · Authors · 2024-11-30
> **Additional comparison with variational inference/Langevin dynamics**
>
> Dear Reviewer eKpC,
>
> We would like to get back to you after we tried implementing Variational Inference Langevin dynamics to sample from our posterior.
>
> ---
>
> ### **Challenges with Variational Inference and Langevin Dynamics for Posterior Sampling with Diffusion Priors**
>
> We found both VI and Langevin dynamics extremely challenging to apply when the prior is a diffusion model (e.g., as we attempted with viTS [3]). Both approaches rely on the posterior being proportional to $e^{-V(\theta)}$, where $\nabla V(\theta)$ (the gradient of the potential) is assumed to be tractable. The difficulties stem from the nature of $V(\theta)$, which involves the log of an (intractable) integral over all latent parameters $\psi_1, \dots, \psi_L$ (see Eq. (3) in the paper). Computing $\nabla V(\theta)$ is non-trivial. Indeed, methods like [1, 2, 3], which successfully apply VI or Langevin dynamics for Thompson Sampling, rely on simpler priors (e.g., Gaussian) combined with non-linear likelihoods. The approximation was needed because of the non-linearity in the likelihood and not the prior. In contrast, our approach involves both a very complex prior (diffusion model) and a potentially non-linear likelihood, significantly increasing the difficulty.
>
> Even if $\nabla V(\theta)$ were tractable (which we could not find a feasible way to achieve), VI would require designing a variational family to match the posterior. Doing so for a diffusion-based prior would demand significant compromises in either expressivity or computational efficiency. Similarly, Langevin dynamics would require fine-tuning for convergence, which becomes tricky in high-dimensional settings with complex priors like ours.
>
> ---
>
> ### **Advantages of Our Approach**
>
> Given these challenges, our efficient approximation offers a practical and effective alternative:
> - **No Hard Numerical Optimization:** It bypasses the need for tractable gradients or explicit variational families.
> - **Preserves Posterior Expressiveness:** The updated posterior remains diffusion-based, ensuring no loss of expressivity.
>
> ---
>
> ### **Summary**
>
> In summary, while VI and Langevin dynamics are standard, their application here is severely limited by the intractability of $V(\theta)$ and its gradient. These constraints make our proposed efficient approximation both appealing and well-suited to this context, avoiding the aforementioned difficulties and providing a practical solution for posterior sampling.
>
> We hope this response clarifies our approach and highlights its novelty and feasibility and increases the reviewer's confidence in our work.

---

### Meta-Review · Area_Chair_ALC6 · 2024-12-21

**Metareview:**

This paper introduces Diffusion Thompson Sampling (dTS), an algorithm that addresses the exploration-exploitation trade-off in contextual bandits with large action spaces. By integrating pre-trained diffusion models, dTS identifies correlations between actions, facilitating more efficient exploration. The algorithm incorporates diffusion models as priors within a Thompson sampling framework, leveraging the underlying structure of the action set to explore the action space more effectively. The theoretical analysis provides upper bounds on the Bayes regret for linear diffusion and reward structures.

The paper is generally well-written, and the proposed algorithm is novel, supported by both solid theoretical results and empirical evaluations. However, during the discussion phase, reviewers identified some weaknesses in the method and its regret analysis. Specifically, the approach seems fundamentally limited when applied to the linear diffusion case. Reviewer KvQk pointed out that linear diffusions reduce to a single Gaussian, making the motivation for using the proposed model in the linear diffusion setting unjustified. As such, the linear assumption appears unreasonable, raising questions about the necessity of a diffusion model in this context. Additionally, the paper does not provide regret analysis for nonlinear diffusions, leaving a significant gap between the theoretical framework and the method’s potential practical applicability. Given these concerns, the reviewers have recommended rejecting the paper.

**Additional Comments On Reviewer Discussion:**

The authors have not adequately addressed or acknowledged critical limitations identified by reviewers, which raises concerns about the method's practical applicability and limitations. This lack of thorough response leaves important questions about the work's scope unanswered.

---

### Decision · Program_Chairs · 2025-01-22

Reject